# An atlas of RNA-dependent proteins in cell division reveals the riboregulation of mitotic protein-protein interactions

Varshni Rajagopal[1,12], Jeanette Seiler[1,12], Isha Nasa[2,3,12], Simona Cantarella[1], Jana Theiss[1], Franziska Herget [1], Bianca Kaifer[1], Melina Klostermann [4,5], Rainer Will [6], Martin Schneider[7], Dominic Helm[7], Julian König[8,9], Kathi Zarnack [4,5], Sven Diederichs [10,11] ✉, Arminja N. Kettenbach [2,3] ✉ & Maïwen Caudron-Herger [1] ✉

Ribonucleoprotein complexes are dynamic assemblies of RNA with RNA-binding proteins, which modulate the fate of RNA. Inversely, RNA ribor-egulates the interactions and functions of the associated proteins. Dysregu-lation of ribonucleoprotein functions is linked to diseases such as cancer and neurological disorders. In dividing cells, RNA and RNA-binding proteins are present in mitotic structures, but their impact on cell division remains unclear. By applying the proteome-wide R-DeeP strategy to cells synchronized in mitosis versus interphase integrated with the RBP2GO knowledge, we pro-vided an atlas of RNA-dependent proteins in cell division, accessible at R-DeeP3.dkfz.de. We uncovered AURKA, KIFC1 and TPX2 as unconventional RNA-binding proteins. KIFC1 was identified as a new substrate of AURKA, and new TPX2-interacting protein. Their pair-wise interactions were RNA depen-dent. In addition, RNA stimulated AURKA kinase activity and stabilized its conformation. In this work, we highlighted riboregulation of major mitotic factors as an additional complexity level of cell division.

From the onset of transcription in the nucleus until their degradation in the cytoplasm, RNA transcripts associate with RNA-binding proteins (RBPs) to form ribonucleoprotein (RNP) complexes[1]. RNPs are dynamic macromolecular assemblies that regulate the fate of RNA molecules by coordinating all aspects of their post-transcriptional maturation and regulation such as splicing, modification, transport, translation and decay[2,3]. Conversely, recent studies pointed to a more RNA-centric view of RNA-protein interactions, termed "riboregula-tion", where RNA modulates RBP localization, conformation, interac-tion and function[4–8].

Due to their central role in various key cellular processes, dysre-gulation of RBP functions is implicated in the initiation and develop-ment of diseases such as neurological disorders, cancer and muscular atrophies[3,9,10]. Therefore, RBPs have attracted increased interest in

[1]Research Group "RNA-Protein Complexes & Cell Proliferation", German Cancer Research Center (DKFZ), Heidelberg, Germany. [2]Department of Biochemistry and Cell Biology, Geisel School of Medicine at Dartmouth, Hanover, NH, USA. [3]Norris Cotton Cancer Center, Geisel School of Medicine at Dartmouth, Lebanon, NH, USA. [4]Buchmann Institute for Molecular Life Sciences, Frankfurt, Germany. [5]Department of Bioinformatics, University of Würzburg, Würzburg, Germany. [6]Cellular Tools Core Facility, German Cancer Research Center (DKFZ), Heidelberg, Germany. [7]Proteomics Core Facility, German Cancer Research Center (DKFZ), Heidelberg, Germany. [8]Institute of Molecular Biology (IMB), Mainz, Germany. [9]Theodor Boveri Institute, Biocenter, University of Würzburg, Würzburg, Germany. [10]Division of Cancer Research, Department of Thoracic Surgery, Medical Center - University of Freiburg, Faculty of Medicine, University of Freiburg, Freiburg, Germany. [11]German Cancer Consortium (DKTK), partner site Freiburg, a partnership between DKFZ and University Medical Center Freiburg, Freiburg, Germany. [12]These authors contributed equally: Varshni Rajagopal, Jeanette Seiler, Isha Nasa. ✉e-mail: s.diederichs@dkfz.de; Arminja.N.Kettenbach@dartmouth.edu; m.caudron@dkfz-heidelberg.de

recent years, leading to the development of multiple strategies to establish comprehensive catalogs of RBPs in humans and in other species[11,12]. Proteome-wide approaches include the identification of the mRNA-bound proteome via UV-crosslinking and subsequent oligo(dT) capture[13–18]. Other techniques based on protease digestion[19,20], modified nucleotides[21,22] or organic phase separation[23–25] have been established to include RBPs that also bind to non-polyadenylated RNAs and orthogonal approaches include bioinformatic-based studies[26,27]. R-DeeP is a complementary experimental strategy based on a sucrose density gradient and ultracentrifugation and thus independent of affinity or property-based protocols[28–30]. R-DeeP adapts the concept of RNA dependence with RNA-dependent proteins being defined as proteins whose interactome depends on RNA and thus comprising proteins interacting directly or indirectly with RNA molecules. Using the R-DeeP strategy, more than 700 new RNA-dependent proteins have been recently identified in human cancer cells[30] and 545 in *Plasmodium falciparum*[31].

All these valuable resources support the view that canonical RBPs can bind to RNA through their structurally well-defined RNA-binding domains (RBDs) such as the RNA recognition motif (RRM) or the K-homology domain (KH)[12,32]. However, they also point to a large number of unconventional RBPs that do not contain any canonical RBDs[5,33,34]. Interestingly, the non-canonical or unconventional RBPs represent the large majority of RBPs detected in humans (76.2% of the RBPs) and other species[12]. Unconventional RBPs usually perform primary functions in a great variety of key cellular pathways which are unrelated to RNA metabolism - and are thus prime candidates for proteins which do not regulate RNA, but are regulated by RNA interactions. Currently, their affinity for RNA and possible physiological relevance remains mostly uncharacterized. A few recent examples include the RNA-dependent stimulation of the transcription co-activator CBP to promote histone acetylation and regulation of enhancer function[35], the RNA-dependent differential regulation of the interferon-inducible protein kinase R (PKR)[6], the importance of the RNA-binding activity of ERα for tumor cell survival and therapeutic response[36] and the riboregulation of the glycolytic enzyme enolase 1, which alters cell metabolism and stem cell differentiation[7].

In the context of cell division, several studies have reported the presence of RNA, *i.e.* protein-coding RNAs (mRNAs)[37–39], non-coding RNAs (ncRNAs)[40,41] and ribosomal RNAs (rRNAs)[39,42], as well as various RBPs[43,44] within structures of the mitotic spindle apparatus such as the centrosomes[45,46], the kinetochores[46] and the spindle microtubules[38,39]. Collectively, these works point to the possible importance of RNA and RBPs for the structural and functional integrity of the mitotic spindle. However, although RNase treatment[43,47] and inhibition of splicing and transcription[40] can disrupt the mitotic spindle apparatus in the *Xenopus laevis* egg extract system, the underlying mechanisms remain unknown. Therefore, we aim here to investigate RNA-dependent proteins involved in mitosis and how they might be affected by RNA.

The available proteome-wide RBP screens provide a highly valuable resource to further investigate the role and relevance of RBPs and RNA in key cellular processes. However, due to the low number of mitotic cells in non-synchronized cell populations, the differential protein expression levels throughout the cell cycle and the dynamics of post-translational modifications that may affect the affinity of proteins for RNA, these screens so far are not adapted to adequately reflect this important phase of the cell cycle. Therefore, we applied the R-DeeP screening strategy[28,29] in both mitotic and interphasic synchronized HeLa cells, to collect specific and quantitative data on RNA-dependent proteins in both cell cycle phases. This resource is now available online in the R-DeeP 3.0 database (https://R-DeeP3.dkfz.de).

Taking advantage of this new resource, we provide an atlas of RNA-dependent proteins in cell division, unraveling that a large number of well-known mitotic factors and key cell division players are linked to RNA, some also supported by the multiple studies that were integrated into the RBP2GO database[12]. In particular, we identify and characterize the RNA dependence of a major mitotic regulator, Aurora kinase A (AURKA). We reveal that AURKA directly binds to RNA and that the presence of RNA is essential for the interaction of AURKA with multiple proteins, including several known targets such as TPX2 and a new target, the kinesin-like protein KIFC1 (also known as XCTK2 or HSET, from the Kinesin-14 family proteins). We show that AURKA phosphorylates KIFC1 at $S^{349}$ and $T^{359}$. Our data reveal that KIFC1, AURKA and TPX2 are unconventional RBPs, *i.e.*, they directly bind to RNA, without containing a known RNA-binding domain. The detailed analysis of the KIFC1-bound RNA reveals that KIFC1 interacts with both rRNA and protein-coding transcripts without apparent sequence specificity. Finally, we demonstrate that RNA stabilizes the conformation of AURKA and stimulates its kinase activity.

Altogether, our findings reveal an essential role for RNA in mediating RNA-dependent mitotic protein-protein interactions and in riboregulating the function of AURKA. Thus, our atlas of RNA-dependent proteins in cell division promises the exciting development of a research field integrating RNA-protein interactions as a new layer of regulation in our understanding of cell division processes.

## Results

### Differential R-DeeP screens in mitosis and interphase provide cell cycle-specific knowledge for RNA-dependent proteins

A gene ontology (GO) analysis of the RNA-dependent proteins identified in unsynchronized HeLa cells[28,29] revealed the significant enrichment of proteins with GO terms related to mitosis (Supplementary Fig. 1a). However, mitotic cells are underrepresented in an unsynchronized cell population. Therefore, to obtain a more comprehensive landscape of the RNA-dependent proteins in mitosis, we established the R-DeeP approach in HeLa cells synchronized in mitosis and in interphase (Fig. 1a, Supplementary Fig. 1b, c).

The migration pattern of the proteins, i.e., their distribution across the different fractions in the control as compared to the RNase-treated gradients revealed the RNA dependence (change in the position – fraction - of the protein in the gradient) or independence (same position/fraction) of the proteins (Fig. 1a). We quantified a total of 7152 proteins in the mitotic gradient and 7060 proteins in the interphasic gradient, which included 6059 proteins that were commonly found in both screens (Fig. 1b).

Using the established bioinformatic pipeline to automatically identify RNA-dependent proteins[28,29], Gaussian-fitted distributions were calculated for each protein profile, and shifts were characterized based on (i) amount of protein shifting in control vs. RNase-treated gradients: indicated by area under the curve, (ii) position of the peaks representing the apparent molecular weight of the protein, (iii) direction and distance of the shift, (iv) amplitude difference between control and RNase curves at each maxima and (v) statistical significance in the difference between amplitude maxima in the control and RNase-treated gradients. A protein was classified as RNA dependent if the distance between a peak in the control and a peak in the RNase-treated gradients was at least greater than one fraction with statistically significant difference between the two gradients for at least one peak (Supplementary Data 1). Next, the proteins were classified according to their shift and direction as follows: (i) left shifting proteins: proteins whose maxima were detected in lower fractions in RNase-treated gradients compared to control gradients indicating a lower apparent molecular weight of the protein upon RNase treatment (Supplementary Fig. 2a, left shift, displacement to the left from green to red), (ii) right shifting proteins: proteins whose maxima were detected in later fractions in RNase-treated gradients compared to control gradients (Supplementary Fig. 2a, right shift, displacement to the right from green to red), (iii) precipitated proteins: proteins shifting to greater fractions (fraction >23) upon RNase treatment and (iv) non-shifting proteins: proteins whose maxima were detected in the

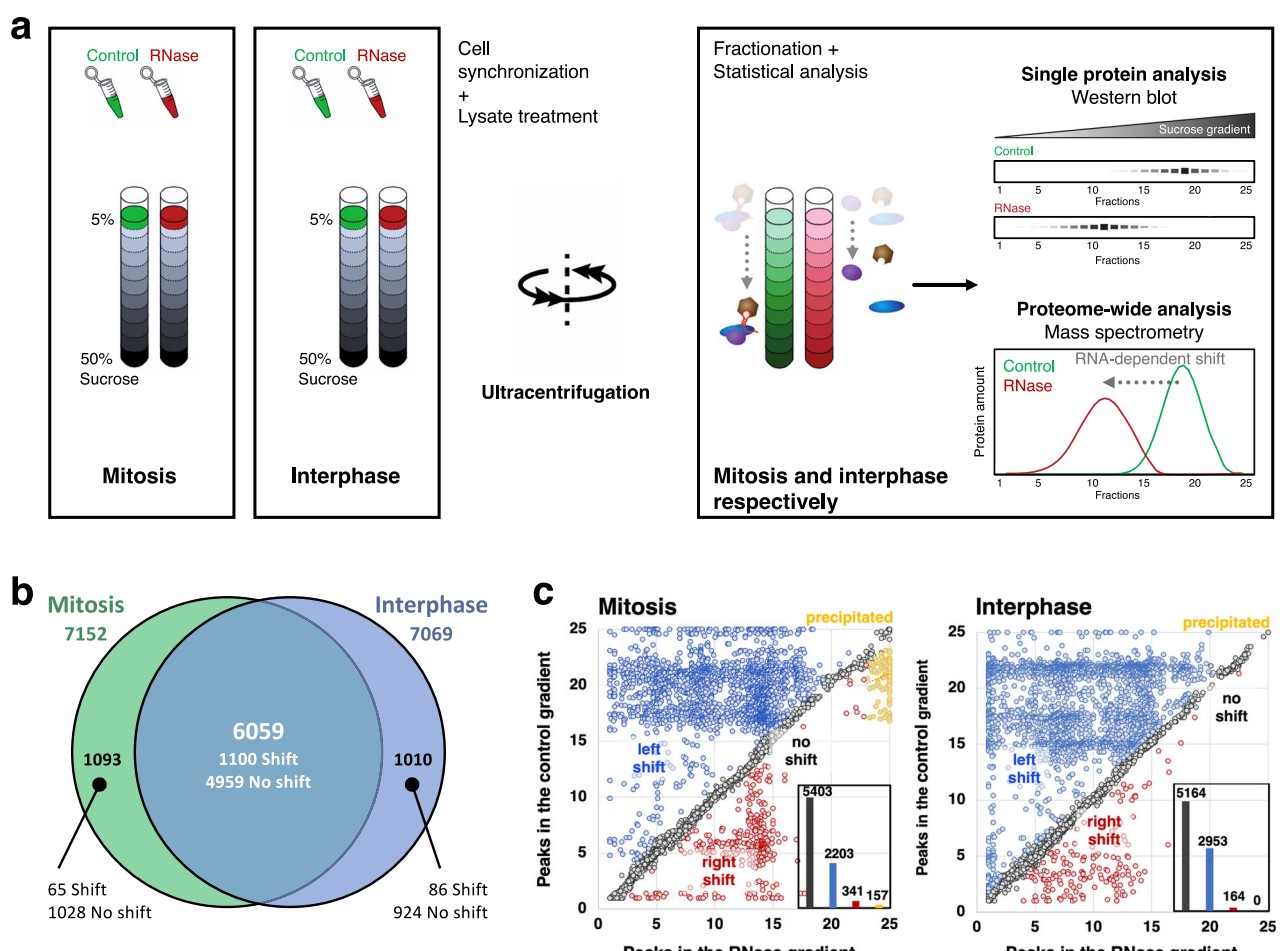

**Fig. 1 | R-DeeP in HeLa mitotic and interphasic cells. a** HeLa mitotic or interphasic untreated (control) or RNase-treated cell lysates were prepared and loaded onto sucrose density gradients (5–50%). Following ultracentrifugation, the gradients were fractionated into 25 fractions and subjected to either mass spectrometry (proteome-wide screen) or Western blot analysis (individual protein analysis) for validation of the screens ($N = 3$ in total). Adapted from[29]. **b** Venn diagram indicating the total number of proteins identified in each R-DeeP screen (mitosis: 7152 and interphase: 7069). 6059 proteins were commonly identified. **c** Each graph (mitosis and interphase, respectively) indicates the position of maxima for each shift in the control and RNase-treated gradients according to the mean of three replicates. The inset bar graph shows the number of proteins in each sub-category (left shift: blue, right shift: red, no shift: gray and precipitated proteins: orange). Source data for graphs are provided as Source Data files (see Data availability). Code for mass spectrometry analysis is available online (see Code availability).

same fraction in RNase-treated gradients compared to control gradients (Fig. 1c and Supplementary Fig. 2a). As commonly observed[28,30,31], RNA-dependent proteins were dominated by left-shifted proteins, i.e., proteins losing interaction partners upon RNase treatment (Fig. 1a, c, Supplementary Fig. 2a).

Out of the 7152 detected proteins in the mitosis screen, we identified 1751 proteins with at least one shift. Similarly in the interphase screen, 1912 out of the 7060 detected proteins depicted at least one shift (Supplementary Data 1, Supplementary Fig. 2b). Taking advantage of the quantitative aspect of the R-DeeP screen, a shifting coefficient was calculated according to the amount of protein in each peak of the control and RNase-treated gradients[28]. Based on the shifting coefficient, the proteins were further categorized as (i) partially RNA dependent (partial shift, i.e. only part of the protein amount shifts), (ii) completely RNA dependent (complete shift, i.e. the entire amount of the protein shifts) and (iii) RNA independent (no shift) (Supplementary Fig. 2c). As a quality control and to validate the RNA dependence of the newly identified RNA-dependent proteins, analyses on multiple parameters related to RBPs were performed by comparing the groups of shifting vs non-shifting proteins. For example, the recently defined RBP2GO composite score, which reflects the probability for a protein to be a true RBP[12], was consistently and significantly higher for shifting proteins as compared to non-shifting proteins in both mitotic and interphasic screens (Supplementary Fig. 2d). In addition, as expected from previous studies[14,26,28], mitotic and interphasic shifting proteins depicted higher isoelectric points (pI) than non-shifting proteins (Supplementary Fig. 2e).

Within left-shifting proteins, which was the largest category of shifting proteins, 948 proteins were identified in both mitotic and interphasic screens, out of which 160 proteins were more highly expressed in mitosis and 52 proteins were more highly expressed in interphase (Fig. 2a and Supplementary Data 1). While the 948 commonly left-shifting proteins were overall enriched in proteins with RNA-related functions, the 160 proteins more expressed in mitosis were enriched in proteins related to mitosis (Supplementary Data 1), pointing to RNA-dependent mitotic factors. An additional comparative analysis revealed two groups of RNA-dependent proteins specific for interphase and mitosis, respectively (Fig. 2b, c). A GO enrichment analysis indicated that these two groups shared a number of common GO terms, which were often linked to mitochondrial and endoplasmic reticulum components, which can be linked to nocodazole treatment in the synchronization procedure[24]. The two groups also shared the GO term "cytoskeleton", with the distinction though, that mitotic proteins were more related to microtubule

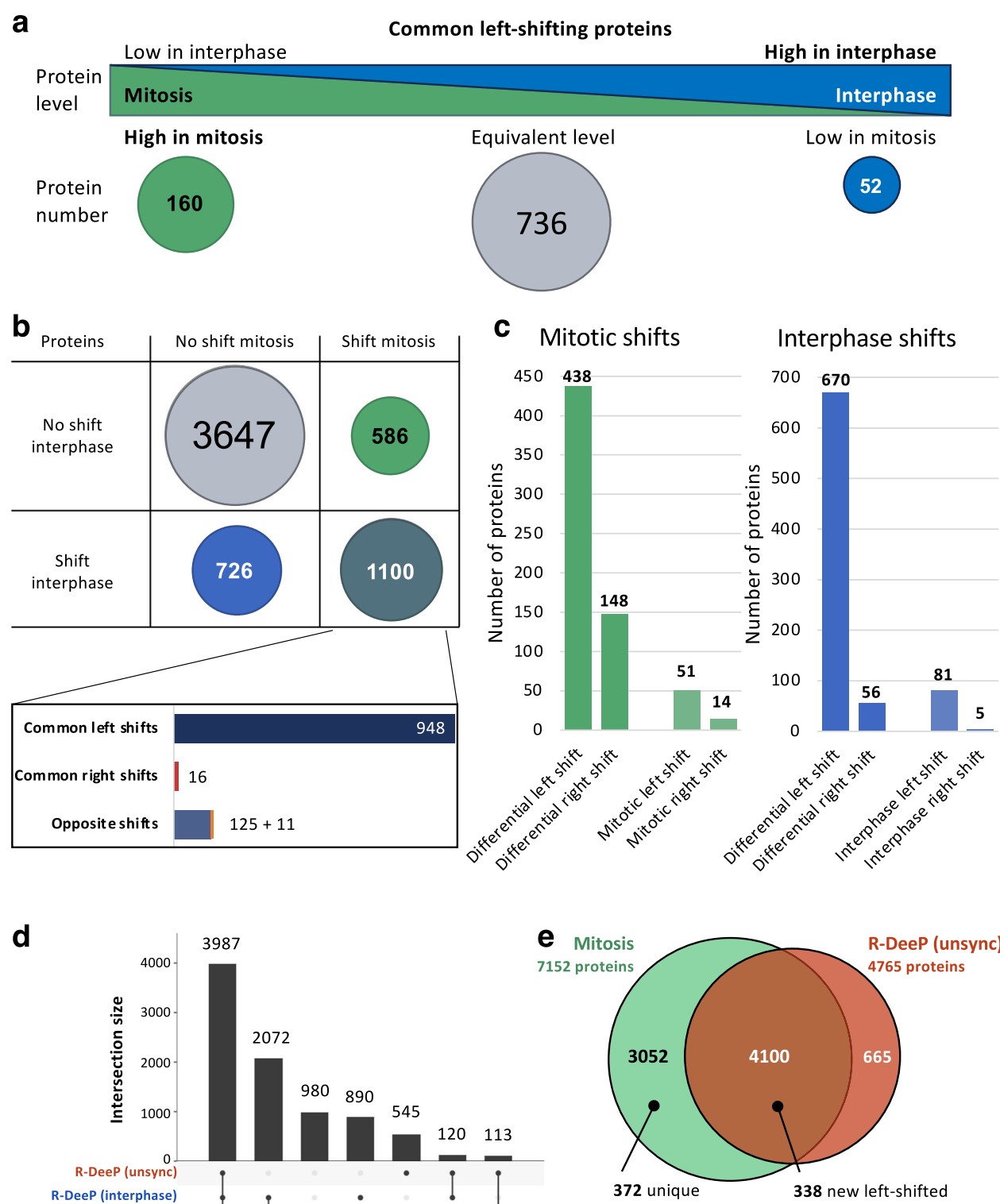

functions and interphasic proteins to actin functions (Supplementary Data 1), indicating a possible regulatory role of RNA in interphase via actin and in mitosis via microtubules. Finally, a group of 136 proteins with opposite shifts was identified (Fig. 2b), which contained 46 proteins with catalytic functions (Supplementary Data 1). Such proteins seemed to present a cell cycle-dependent relation to RNA, leading to loss (left shift) or gain (right shift) of interaction partners. Similar observations had been previously reported for proteins involved in cellular metabolism[24].

Importantly, we identified 389 and 426 RNA-dependent proteins in mitosis and interphase, respectively (Supplementary Fig. 2b), which had never been linked to RNA before according to the RBP2GO database[12]. While the detected proteins in the two new R-DeeP screens covered 84% (3987 out of 4765 proteins) of the previous screen in unsynchronized HeLa cells[28], they also added together new data (protein profiles, R-DeeP3.dkfz.de) for 3942 proteins (Fig. 2d). Moreover, we identified 338 proteins which had no significant shift in unsynchronized cells but were now presenting a significant shift in

**Fig. 2 | Differential analysis of R-DeeP in HeLa mitotic and interphasic cells.**
**a** Schematic representation of the differential protein levels of the common left-shifting proteins in mitotic and interphasic R-DeeP screens in HeLa cells. While 736 proteins depicted equal expression levels in both cell cycle phases, 160 proteins were higher expressed in mitosis and 52 proteins were preferentially expressed in interphase and thus showed differential expression in a cell cycle dependent manner. **b** Table depicting differential shifting behavior of the 6059 common proteins in mitosis as compared to interphase. While 1100 proteins presented a significant shift in both phases, we detected 726 proteins with interphase specific shifts and 586 mitotic proteins with specific shifts. 3647 proteins did not present RNA dependence. Within the 1100 proteins commonly shifting, most of the proteins were left shifted. There were also 136 proteins shifting in opposite directions. **c** Bar plot indicating the details of the mitotic- and interphase-shifting proteins. The

number of proteins which were shifting only in one cell cycle phase are indicated in the "differential" shift categories. (438 + 148 = 586 in mitosis and 670 + 56 = 726 in interphase). The number of proteins which were detected only in one cell cycle phase are indicated in the mitotic left/right shift category (51/14) and interphase left/right shift category (81/5), respectively. **d** Upset plot showing the intersection between the three R-DeeP screens: "R-DeeP unsync" in unsynchronized HeLa cells, and the two new "R-DeeP interphase" and "R-DeeP mitosis" datasets. **e** Venn diagram depicting the intersection (number of detected proteins) between the unsynchronized R-DeeP screen in HeLa cells (R-DeeP unsync) and the new mitotic R-DeeP screen. The newly detected left-shifted proteins are indicated below. The results of the R-DeeP analysis with the shifting/non-shifting proteins are available as Supplementary Data 1.

mitosis (Fig. 2e). These proteins were mainly linked to the mitochondrial and reticulum endoplasmic component (Supplementary Data 1). We also identified 372 mitotic-specific RNA-dependent proteins (Fig. 1e), of which, 111 had higher protein levels in mitosis as compared to interphase. These 372 proteins were strongly enriched in proteins related to the microtubules and the mitotic spindle (Supplementary Data 1), pointing again to RNA-dependent factors in mitosis and suggesting a regulatory role for RNA in this context.

Finally, we compared the results of the R-DeeP screen in mitosis to the results of the OOPS study, which used organic phase separation and thymidine/nocodazole treatment to synchronize U2OS cells in mitosis[24]. 346 proteins could be identified as RBP with increased detection in this experiment, of which, 234 were unique to the OOPS study and 112 were commonly detected with our mitotic R-DeeP screen. All commonly detected proteins depicted a significant mitotic R-DeeP shift. In addition to these 112 proteins, the mitotic R-DeeP screen specifically identified 1639 shifting proteins, of which 289 had higher protein levels in mitosis as compared to interphase (Supplementary Fig. 2f and Supplementary Data 1). In support of our results, the RBPs identified in the mitotic-synchronized cells with OOPS were enriched in GO terms related to mitosis, an observation which had not been mentioned in the OOPS study (Supplementary Data 1).

Collectively, our comprehensive analysis of the mitotic and interphasic R-DeeP screens combined with the analysis of the protein levels highlights the important contribution of these screens, which have not only largely broadened the available information, but also strengthened the data basis for the RNA dependence of multiple mitotic factors.

In order to provide an open and easy access to the R-DeeP screen data, we compiled the R-DeeP 3.0 database (https://R-DeeP3.dkfz.de, Supplementary Fig. 3a). R-DeeP 3.0 allows users to conveniently browse the present mitotic and interphasic R-DeeP screens, as well as the previously published screens that had been performed in unsynchronized HeLa[28] and A549 cells[30] to facilitate comparison of the results (Supplementary Fig. 3b). The database provides various search options such as the single or advanced search option using gene name or UniProt ID to either obtain a summary of a single protein or to compare a protein between different cell lines and cell cycle stages. These multiple analysis options provide various output formats which can be downloaded for further custom analyses. Detailed instructions on how to use the database are provided via a comprehensive user guide, which can be accessed from the documentation section (Supplementary Note 1).

## An atlas of RNA-dependent proteins in cell division highlights well-known mitotic factors as RNA-dependent proteins

Mitosis is critical for the faithful segregation of the chromosomes to the daughter cells during cell division. This is mediated by a complex and dynamic macromolecular machinery that consists of polymerized microtubules, centrosomes and various specialized proteins[43]. So far, most of the studies focused on the functional characterization of the

proteins and protein-protein interactions[48–50]. Only a few studies have focused on the role of RNA in mitosis[40,42–44]. The role of RNA at centrosomes has been discussed in the past[51] but not further explored. In sum, how RNA might be functionally involved during mitosis still requires further investigations. Our present R-DeeP screen in mitosis depicted an enrichment for proteins with microtubule-related functions (Supplementary Data 1). Taking advantage of these data together with the comprehensive knowledge from the RBP2GO database[11,12], we created a list of RNA-dependent proteins in mitosis. More specifically, we identified 800 RNA-dependent proteins using the advanced search option from the RBP2GO database along with GO terms related to mitosis (Supplementary Data 2). They overlapped with 153 proteins from the R-DeeP mitotic screen, for which R-DeeP profiles are now available. In addition, 26 mitotic proteins which had not been linked to RNA previously were detected as RNA-dependent in the mitotic screen, summing together to 826 RNA-dependent mitotic proteins. Thus, we generated an atlas of 826 RNA-dependent proteins from of a total of 1472 mitotic proteins, meaning that 56.1% of the mitotic proteins were related to RNA (Fig. 3a and Supplementary Data 2). 643 of these proteins (77.8%) did not contain any known RBD and were not linked to an RNA-related InterPro family ID[12]. The information on the localization of these RBPs in various critical mitotic structures such as spindle poles, spindle midzone, kinetochores and chromosomes were included in the map, showing that nearly all mitotic substructures contained RNA-dependent proteins. Many of these proteins have well-known and characterized functions during cell division, e.g., AURKA or TPX2, which are unrelated to RNA. Interestingly, the vast majority belong to the class of unconventional RBPs, awaiting further RNA-related functional characterization. Additionally, we categorized RNA-dependent proteins that are crucial for cell cycle transitions (Fig. 3b). We also integrated the protein lists from the MiCroKITS database into our analysis[52], to provide orthogonal information on spindle, centrosome, kinetochore and midbody proteins and confirmed that >50% of the proteins from these lists were classified as RNA-dependent proteins, as well (Supplementary Data 2).

Altogether, our atlas of RNA-dependent proteins in cell division highlights the extensive incidence of RNA-dependent proteins in cell division structures and events, including key mitotic factors, which strongly suggests RNA as a crucial co-factor.

## Aurora Kinase A is an RNA-dependent protein

Since AURKA is an essential kinase for cell cycle regulation, we further investigated its RNA dependence. AURKA belongs to the highly-conserved serine/threonine kinase family and is essentially activated by autophosphorylation at the $T^{288}$ amino acid residue[53]. It is a cell cycle-regulated protein whose expression level increases from late S-phase and is degraded during mitotic exit[54]. AURKA is first activated in late G2, where the protein interacts with PLK1 and CEP192 for centrosome maturation[50,55]. Upon nuclear envelope breakdown and onset of a Ran-GTP gradient from the chromosomes[56], AURKA interacts with TPX2[48,49]. AURKA and TPX2 activate each other and interact with

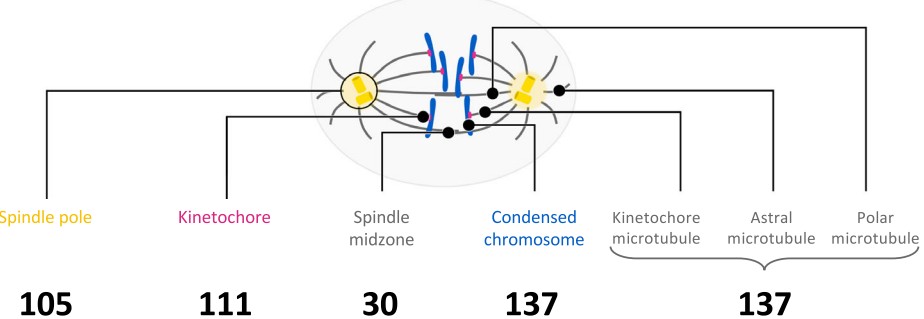

**a** RNA-dependent proteins during spindle assembly

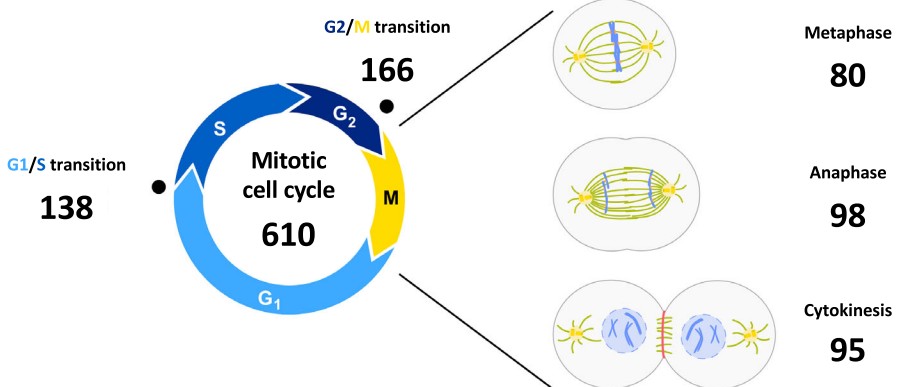

**b** RNA-dependent proteins in mitotic cell cycle

**Fig. 3 | An atlas of RNA-dependent proteins in cell division. a** Schematic representation of a mitotic spindle showing the localization of RNA-dependent proteins to mitotic structures, based on their association with the respective GO terms as indicated. Data from the 43 proteome-wide human RBP screens as compiled in the RBP2GO database were used for this analysis. The numbers indicate the number of proteins in each group. In total, 826 mitosis-related proteins (out of 1472 proteins) depict an RNA dependence in mitosis. The complete dataset can be consulted in Supplementary Data 2. **b** Same as in **a** for mitotic proteins involved in different cell cycle transitions and related to the mitotic cell cycle. The complete mitotic R-DeeP profile of the proteins is available within the R-DeeP3 database (R-DeeP3.dkfz.de).

several other proteins such as NEDD1, RHAMM, and γ-Tubulin Ring Complexes to promote microtubule nucleation, spindle focusing and bundling[50,57]. AURKA serves also as a therapeutic cancer target that is often overexpressed in several cancers[57,58].

In our present mitotic R-DeeP screen, the mass spectrometry analysis depicted a complete left shift in the AURKA profile upon RNase treatments from the control fraction 21 to the RNase fraction 5 (Fig. 4a). The RNA-dependent shift of AURKA in mitosis was validated per Western blot analysis. The quantification graph of the Western blot bands showed a similar left shifting pattern as seen in the mass spectrometry quantification, strongly validating the RNA dependence of AURKA in mitosis and suggesting a loss of interaction partners upon RNase treatment (Fig. 4b, c).

**KIFC1 is a novel and RNase-sensitive AURKA interaction partner**
Protein-protein interactions of AURKA as a key mitotic factor were extensively studied, largely based on proteomic identification of its mitotic substrates[59,60]. The loss of interaction partners upon RNase treatment motivated us to further investigate the RNA-dependent protein interactors of AURKA in mitosis. Therefore, we performed immunoprecipitation of AURKA in mitotic HeLa cell lysates, in the presence or absence of RNase treatment, followed by mass spectrometry analysis. Among the identified AURKA interacting proteins, 90.4% (1080 out of 1194) of the proteins were RNA-dependent according to the RBP2GO classification and our new mitotic R-DeeP

screen (Fig. 5a and Supplementary Data 3). We further analyzed the AURKA interactors which were sensitive to RNase treatment. In order to narrow down to the most interesting RNase-sensitive and RNA-dependent AURKA interactors, we computed the intersection between the RNase-sensitive AURKA interactors (163 proteins) and the RNA-dependent proteins of the mitotic R-DeeP screen that co-migrated with AURKA (same fraction, 406 proteins) in the control sucrose gradient (Supplementary Data 3). This resulted in 43 proteins, of which 7 were linked to mitosis-related GO terms (Supplementary Data 2): KIFC1, NOL6, CLASP1, CLASP2, MEAF6, RHAMM and TPX2, which could potentially be part of a complex (Fig. 5b). Among these proteins, KIFC1 was the one with the highest mitotic fold change in protein amount between mitosis and interphase (Supplementary Data 1), whose interaction with AURKA had not been characterized before. KIFC1 is a minus end directed motor protein localized to the nucleus due to the presence of a nuclear localization signal (NLS) in its tail domain[61]. Similar to TPX2, KIFC1 is bound by importins but released in a RanGTP-dependent manner to then function in spindle organization, microtubule focusing and crosslinking[62–64]. In the following, we will focus on validating the RNA-dependent interactions of AURKA with TPX2 and the new interactor KIFC1.

The data from our mitotic R-DeeP analysis confirmed that KIFC1 and TPX2 were RNA-dependent proteins, both with control peaks around fraction 21 (like AURKA) and an additional peak around fraction 17 for KIFC1, suggesting involvement of KIFC1 in two complexes of

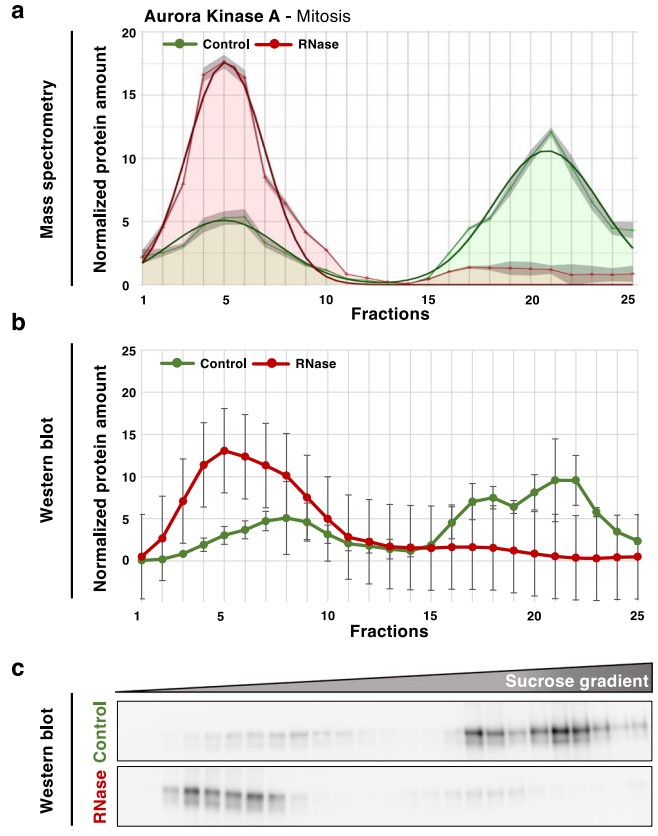

**Fig. 4 | AURKA is an RNA-dependent protein in mitosis. a** Graphical representation of the protein amount in 25 different fractions of control (green) and RNase-treated (red) sucrose density gradients analyzed by mass spectrometry. Raw data (mean of three replicates) are depicted by line with markers. Smooth lines represent the respective Gaussian fit. The overall protein amount of the raw data was normalized to 100. **b** Graph showing the quantitative analysis of Western blot replicates depicted by the mean of three replicates with standard error of the mean (SEM, $N = 3$). **c** Western blot representing the distribution of AURKA (46 kDa) in 25 different fractions in control and RNase-treated mitotic gradients. One replicate out of three biological replicates is shown. Source data for graphs and blots are provided as Source Data files.

different apparent molecular weights (Fig. 5c, d). Similar to AURKA and TPX2, KIFC1 was differentially expressed throughout the cell cycle and highly expressed during mitosis (Supplementary Fig. 4a, b, see R-DeeP3.dkfz.de for the comparative protein profiles in mitosis versus interphase, normalized to the protein amount in mitosis).

Next, we validated the mass spectrometry results of the AURKA pulldown using immunoprecipitation in presence and absence of RNase, followed by Western blot analysis. Upon RNase treatment and AURKA pulldown, the interaction with TPX2 and KIFC1 was significantly lost in both, HeLa and A549 mitotic cells (Fig. 5e, f, Supplementary Fig. 4c, d). We also tested for the interaction of TPX2 with KIFC1, as our data indicated that they could be part of the same complex. Our TPX2 pulldown revealed the interaction between TPX2 and KIFC1. Additionally, we observed a significant decrease of the interaction of TPX2 with KIFC1 and AURKA upon RNase treatment and TPX2 pulldown in mitotic lysates (Supplementary Fig. 4e, f). These observations indicated that the three proteins interacted with each other in an RNA-dependent manner and possibly function as a complex in mitosis.

To visualize the proximity of the proteins in the native cellular environment, we performed a proximity ligation assay (PLA). This technique supports the visualization of in situ interactions between two proteins, at endogenous levels[65]. First, we monitored the

interaction between AURKA, KIFC1 and TPX2 throughout mitosis. All tested interaction pairs, *i.e.* AURKA-KIFC1 (Fig. 6a), TPX2-AURKA (Supplementary Fig. 5a) and TPX2-KIFC1 (Supplementary Fig. 5b) depicted strong PLA signals from prophase (within the centrosomal asters) through metaphase to telophase (more at the spindle poles). This data indicated a colocalization of AURKA, KIFC1 and TPX2 during spindle assembly.

Furthermore, we repeated the PLA in presence or absence of RNase treatment. The resulting signal was quantified using an ImageJ-based analysis (Supplementary Fig. 6a). We observed a clear PLA signal in the control cells in prometaphase and metaphase around the centrosomal asters and the spindles respectively, indicating the proximity of the proteins. In contrast, the PLA signal strongly decreased upon RNase treatment, reaching a level comparable to the background level produced by each antibody individually (Figs. 6b–e, Supplementary Figs. 6b, c and 7). Altogether, these observations further validated that AURKA, TPX2 and KIFC1 interactions were mediated by RNA.

In order to confirm that the loss of interaction between the proteins was due to the removal of RNA, and not due to the adverse effects of RNase itself, the interaction between α- and β-tubulin was used as a negative control. α-tubulin and β-tubulin are the constituting subunits of the microtubules. Thus, we reasoned that the α/β-tubulin interaction should persist as long as microtubules are present in the cell. The overall PLA signal between the control and the RNase-treated samples remained at a high level, confirming the persistence of the α/β-tubulin interaction upon RNase treatment (Supplementary Fig. 8a, b). However, we noticed a modification of the α/β-tubulin PLA signal distribution, which seemed less homogeneously distributed after RNase treatment. To quantify the distribution of the α/β-tubulin PLA signal, the collected images were rotated and rescaled to finally be superposed, so that the variance of the signal could be evaluated pixel per pixel (Supplementary Fig. 8c). This analysis revealed an increased variance of the α/β-tubulin PLA signal throughout the metaphasic spindle structure after exposure to RNase, in particular at the poles of the structure (dark orange, Supplementary Fig. 8c). In addition to the PLA analysis, immunoprecipitation of β-tubulin was performed in mitotic lysates from cells that stably expressed GFP-α-tubulin. The Western blot analysis also confirmed the persistence of the interaction between β- and α-tubulin after RNase treatment (Supplementary Fig. 8d, e).

Altogether, we demonstrated that both AURKA and TPX2 interacted with KIFC1 throughout mitosis and that the interactions between AURKA, KIFC1 and TPX2 were RNA-dependent. While α/β-tubulin interaction persisted upon RNase treatment, the distribution of the PLA signal became less homogeneous, indicating the introduction of some perturbations in the microtubule-based structure. Thus, our results indicated that RNA differentially regulates protein-protein interactions during mitosis.

### KIFC1-interacting RNAs do not contain sequence-specific binding sites

After demonstrating that the interactions between the AURKA, KIFC1 and TPX2 were RNA dependent, we performed an individual-nucleotide resolution UV cross-linking and immunoprecipitation (iCLIP) assay[66–70] to determine whether the proteins directly interacted with RNA transcripts. Under conditions that were adapted to each protein, we observed their direct interaction with RNA in HeLa and A549 cells synchronized in mitosis. Upon increasing dilution of the RNase 1 (1:5 to 1:1000 depending on the protein), the RNA signal migrated to higher molecular weight, corresponding to protein-RNA complexes with increasing ranges of transcript length (Fig. 7a, b and Supplementary Figs. 9–14).

To further identify the interacting RNAs, we performed an iCLIP[66,67] for KIFC1, which presented a high radioactive signal of interacting RNAs (Fig. 7b) and sequenced these RNAs. About 80 million

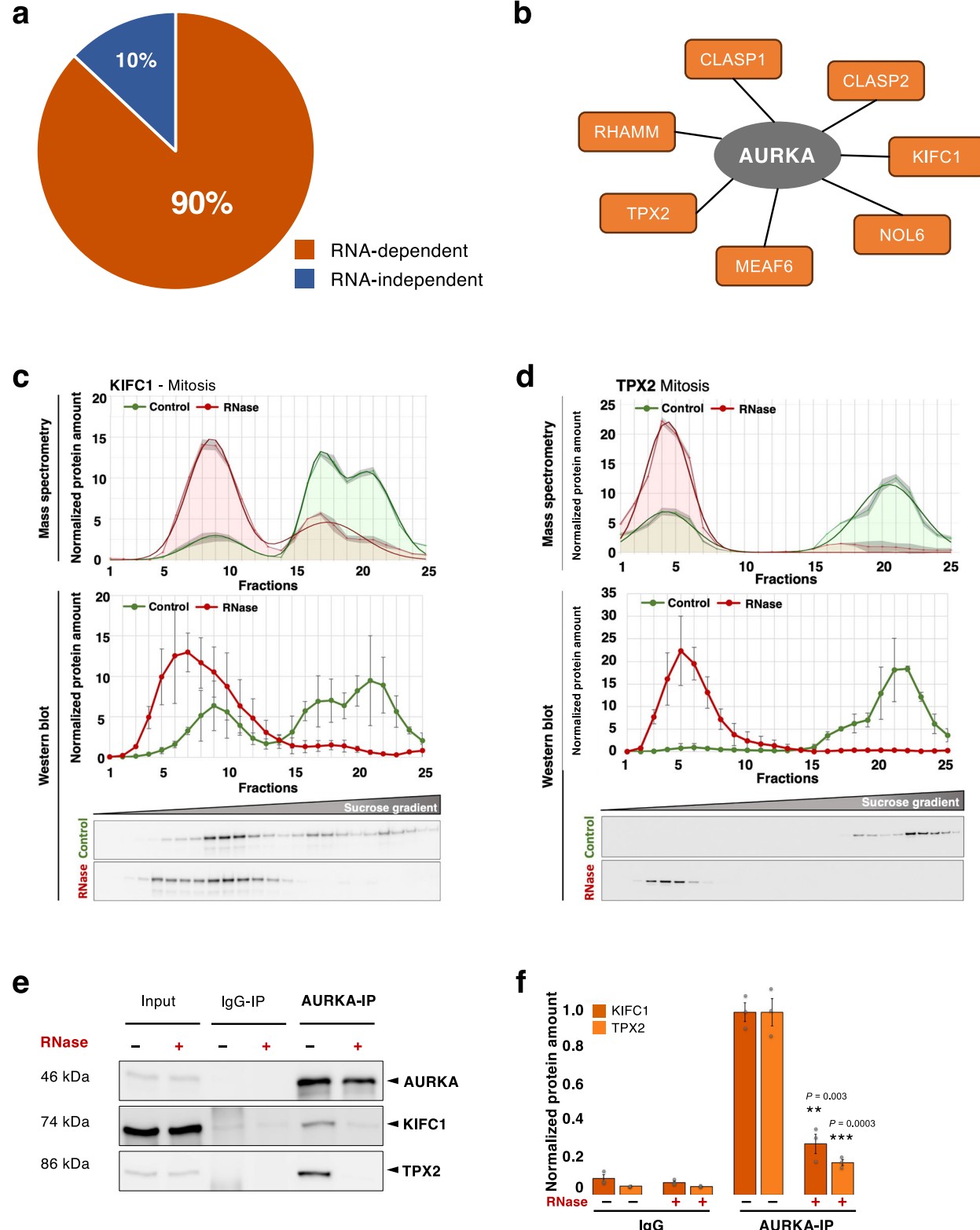

reads were obtained from each of the four replicates, which correlated well with each other (Supplementary Fig. 15a). Following quality control and alignment to rRNA sequences, we noticed that up to 45% of the sequenced reads before removal of the unmapped reads represented rRNA (Fig. 7c). Furthermore, the analysis of the uniquely mapped reads showed that the vast majority of the binding sites (~80%, 29465 out of 36934) were associated with protein-coding mRNA transcripts

(Fig. 7d–f, Supplementary Fig. 15b and Supplementary Data 4). In total, KIFC1-associated binding sites were localized in 5687 genes (Fig. 7d and Supplementary Data 4), in particular in the coding, intronic and 3'UTR sequences (Fig. 7e, f). The analysis of the pentamer frequency at the binding sites did not reveal a particular sequence specificity for the strong binding sites, which were defined according to the PureCLIP scoring system[71]. The most enriched pentamers contained U-stretches,

**Fig. 5 | RNA-dependent protein interactors of AURKA. a** Pie chart showing the percentage of AURKA protein interactors that are RNA dependent (dark orange) or RNA independent (dark blue), according to RBP2GO and our new mitotic R-DeeP screen[12], and identified by AURKA pulldown and mass spectrometry analysis in HeLa cells synchronized in mitosis (*N* = 3). See also Supplementary Data 3. **b** Schematic representation of AURKA interactors which are RNase-sensitive mitotic factors, RNA-dependent according to the mitotic R-DeeP screen and co-migrating around the control fraction 21 together with AURKA. See also Supplementary Data 3. **c, d** R-DeeP profile of KIFC1 (74 kDa) and TPX2 (86 kDa) respectively showing their RNA dependence in mitosis and Western blot validation as for AURKA (see Fig. 4). Top: graphical representation of the protein amount in 25 different fractions of control (green) and RNase-treated (red) sucrose density gradients analyzed by mass spectrometry. Raw data (mean of three replicates) are depicted by line with markers. Smooth lines represent the respective Gaussian fit. The overall protein

amount of the raw data was normalized to 100. Bottom: graph showing the quantitative analysis of Western blot replicates depicted by the mean of three replicates with standard error of the mean (SEM, *N* = 3). **e** Western blot analysis showing the immunoprecipitation of AURKA in mitotic HeLa cells. AURKA pulldown was performed in mitotic lysate treated in presence or absence of RNase I. Rabbit IgG was used as a negative control. KIFC1 (74 kDa) and TPX2 (86 kDa) were pulled down with AURKA (46 kDa) in control samples whereas their interaction was significantly reduced upon RNase treatment (reduction of the band intensity for KIFC1 and TPX2 in the last lane). **f** Graph representing quantification of the amount of KIFC1 and TPX2 present in IgG and AURKA pulldown samples treated or not with RNase I. The intensity of the bands as in **e** were quantified and represented in the bar graph with SEM (three biological replicates). *P*-values were evaluated using a two-tailed, paired t-test (**\**P*-value < 0.01, \*\*\**P*-value < 0.001). Source data for graphs and blots are provided as Source Data files.

most likely reflecting the uridine bias of UV crosslinking[72], and were rather localized in the lower 20% binding sites (Fig. 7g). In support of these findings, we observed a strong correlation of iCLIP crosslinks per gene with the number of RNA-seq reads per gene (Supplementary Fig. 15c). A correlation is to be expected since the total iCLIP signal on a transcript is usually dominated by background binding and hence scales with expression[68,73,74]. However, it is important to distinguish between crosslinks and binding sites, as not all crosslinks on genes will define a binding site. Therefore, an analysis focusing only on the signal within the binding sites indicated that the transcript expression influenced the strength of binding sites to some degree, as stronger binding sites resided on more highly expressed transcripts (Supplementary Fig. 15d). In summary, RNA sequences binding to KIFC1 were predominantly originating from rRNA and protein-coding genes. They were apparently lacking sequence specificity with binding frequency largely reflecting transcript abundance.

### AURKA phosphorylates KIFC1 at S$^{349}$ and T$^{359}$

Next, we reasoned that AURKA is one of the major mitotic kinases, whose main function is to phosphorylate several mitotic factors for their effective activation and promotion of cell division[50]. Since the interaction between KIFC1 and AURKA was not characterized, we tested the hypothesis that KIFC1 could be a substrate for AURKA, as suggested by earlier large-scale quantitative phosphoproteomic analyses[60]. Using the suggested consensus sequence of AURKA[60,75], we selected eight serine and threonine residues that were distributed across the KIFC1 sequence and localized in various domains such as disordered regions and the kinesin motor domain (Supplementary Fig. 16a, b). Using KIFC1 pulldown from HeLa cell lysates over-expressing KIFC1 wild-type (WT) or KIFC1 mutants with non-phosphorylatable alanine, we performed an in vitro kinase assay, in the presence or absence of purified AURKA. The assay revealed two interesting residues in KIFC1: S$^{349}$ and T$^{359}$, whose mutation to alanine induced a strong reduction of phosphorylation by AURKA (Supplementary Fig. 16c, d). Confirming these results, a mutant version of KIFC1, containing both mutations (S349A and T359A) depicted a >90% loss of phosphorylation as compared to the KIFC1 WT protein (Fig. 8a, b and Supplementary Fig. 17a). A kinase-dead AURKA D274A confirmed the specificity of the obtained phosphorylation of KIFC1 by AURKA. Altogether, our results show that AURKA and KIFC1 interact in an RNA-dependent manner and that AURKA phosphorylates KIFC1 at residues S$^{349}$ and T$^{359}$.

### RNA stimulates AURKA kinase activity and stabilizes AURKA conformation

Given that AURKA phosphorylates both, TPX2[48,49] and KIFC1, and knowing that these protein interactions were dependent on RNA, we speculated that also AURKA kinase activity could be dependent on RNA. To test this hypothesis, we performed an in vitro kinase assay to analyze the phosphorylation of KIFC1 and TPX2 by AURKA, using

purified proteins, both in the absence or presence of total RNA. The addition of total RNA resulted in a strong stimulation of the kinase activity, as seen from the increased phosphorylation signal of KIFC1 (Fig. 8c and Supplementary Fig. 17b) and TPX2 (Fig. 8d and Supplementary Fig. 17c) as compared to the samples without RNA. The control samples without AURKA or using AURKA dead-kinase (D274A) in presence of RNA did not show any phosphorylation signal, confirming that total RNA could only stimulate specifically the kinase activity of AURKA WT.

Noticing an increased autophosphorylation signal at the level of AURKA in presence of RNA in the in vitro kinase assay with TPX2 (Fig. 8d), we tested whether total RNA could also stimulate the autophosphorylation of AURKA. In support of this assumption, the addition of total RNA to purified AURKA resulted in an increased autophosphorylation signal, which seemed to be stronger when AURKA was alone than when mixed with KIFC1 or TPX2 (Fig. 8e and Supplementary Fig. 17d).

Protein-protein interactions but also the binding of protein to nucleic acids can induce structural changes in or change the stability of the conformation of the proteins, as reported during the assembly of the ribosomes[76]. In particular, binding to RNA can induce a disorder-to-order transition of intrinsically disordered protein regions[77,78], resulting in the stabilization of the protein structure. Therefore, to test whether total RNA could induce a stabilization of AURKA conformation, we performed a nano differential scanning fluorimetry (nanoDSF) experiment[79]. NanoDSF is used to monitor the conformational stability of a protein upon thermal or chemical stress conditions. Here, we applied a linear gradient of temperature ranging from 20 °C to 90 °C and followed the unfolding of the protein using the intrinsic fluorescence of tryptophan residues, which is expected to switch from 330 nm to 350 nm upon exposure to different environments, e.g. when the protein is unfolding. The addition of total RNA to purified AURKA resulted in an increased thermostability of AURKA, with a melting temperature shifting to a higher temperature (Fig. 8f and Supplementary Fig. 17e). This effect was annihilated under high-salt buffer conditions (Fig. 8f).

Thus, the interaction of RNA with AURKA is functionally relevant as it stimulates AURKA kinase activity. Mechanistically, the stabilization of AURKA conformation by RNA could contribute to the stimulation of its kinase function.

## Discussion

The field of RNA biology is exposed to a number of exciting new challenges linked to the discovery of an increasing number of RBPs[11,13–23]. On the one hand, unconventional RBPs, which interact with RNA in absence of canonical RBDs, represent a large majority of the RBPs in many species[12]. Usually, the cellular functions of these unconventional RBPs are well-characterized, but their interaction with RNA was overlooked and thus remained for most of them unexplored. On the other hand, in a traditional view, RNA-protein interactions and

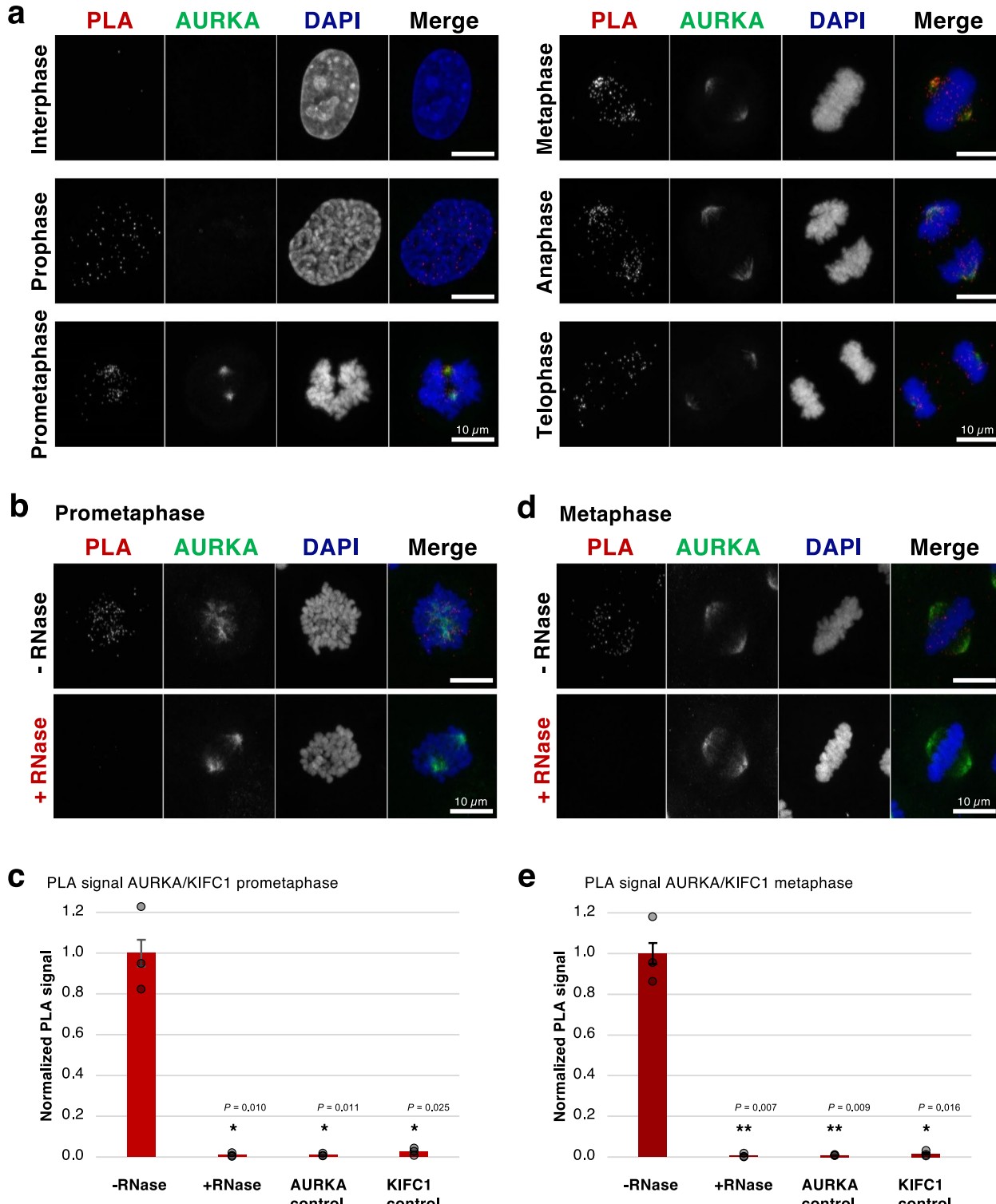

**Fig. 6 | RNA dependence of the identified interaction between AURKA and KIFC1. a** Representative proximity ligation assay (PLA) images indicating the close proximity of AURKA and KIFC1 in HeLa cells across the cell cycle (interphase to telophase as indicated, $N = 3$ with 10 images each). The interaction is represented by dots (PLA channel, red dots in the merge channel), AURKA was stained per immunofluorescence (green) and DNA was stained using DAPI (blue). Individual antibody controls of the PLA assay are shown in Supplementary Fig. 6. Scale bar, 10 μm. **b, d** Representative images showing the PLA assay of AURKA and KIFC1 in HeLa cells at prometaphase and metaphase, respectively (PLA channel, red dots in the merge channel). AURKA is seen in green (immunostaining) and DNA was stained

using DAPI (blue). The top images depict representative images in control cells. Bottom images depict representative images in RNase-treated cells. Scale bar, 10 μm. **c** and **e** Quantification of the PLA signal (as shown in b and d) for the PLA of AURKA and KIFC1 in HeLa prometaphase and metaphase cells in absence (-RNase) or presence (+RNase) of RNase treatment, as well as in the individual antibody control PLA assays (see Supplementary Fig. 6). The signal intensities in each sample was normalized to the signal intensity of the -RNase sample. The error bars indicate the SEM (for three biological replicates of 10 images each). The $P$-values were calculated using a two-tailed, unpaired t-test (*$P$-value < 0.05, **$P$-value < 0.01). Source data for raw images and graphs are provided as Source Data files.

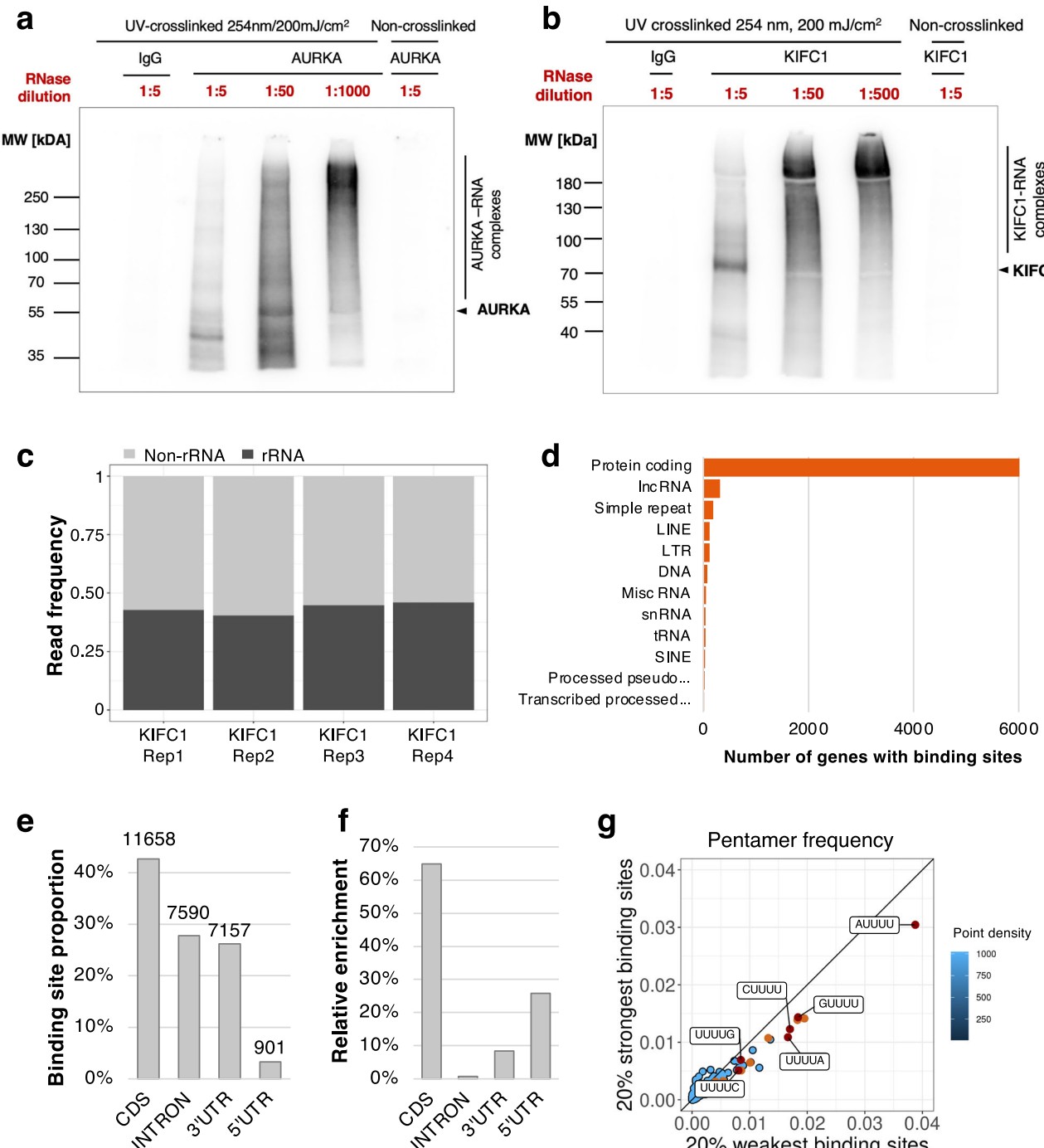

**Fig. 7 | Analysis of the RNA mediating the interaction of AURKA, KIFC1 and TPX2. a, b** Autoradiography indicating the direct binding of AURKA and KIFC1, respectively, to RNA by iCLIP indicated by shifting of the radioactive RNA signal towards higher molecular weights with decreasing RNase I concentrations in HeLa prometaphase cells. The non-crosslinked sample is used as a control for UV-crosslinking which indicates the absence of RNA signal due to the lack of covalent bond between the protein and RNA (one representative image out of $N = 3$ biological replicates is shown for each protein). The results of the corresponding IP showing the protein amounts are shown in Supplementary Figs. 9 and 12. **c** Bar plot depicting the ribosomal RNA (rRNA) and non-ribosomal RNA (non-rRNA) read frequencies in individual KIFC1 iCLIP-Seq replicates in HeLa prometaphase cells. The amount of rRNA reads was determines in the sequencing libraries, before removal of unmapped reads using an alignment to rRNA sequences. **d** Horizontal bar plot showing the KIFC1 target non-rRNA gene spectrum with number of genes

identified in iCLIP in HeLa prometaphase cells in decreasing order. The genes are classified as protein coding, lncRNA, simple repeat, LINE, LTR, DNA, Misc RNA, snRNA, tRNA, SINE, processed pseudogene and transcribed processed pseudogene. **e** Bar plot representing the proportion of binding sites in the respective transcript regions of protein coding genes. The number of regions is indicated at the top of each bar. **f** Bar plot representing the relative enrichment per region, that is, number of binding sites normalized by the summed length of respective bound transcript regions. **g** Scatter plot comparing the pentamer frequency within the 7-nt binding sites in the 20% strong binding sites vs. 20% weakest binding sites as defined by the PureCLIP score. The pentamers with the most extreme frequencies are colored in orange and red and contain U-stretches. These do not appear enriched in the strong binding sites. Source data for blots and code for the analysis are provided as Source Data files.

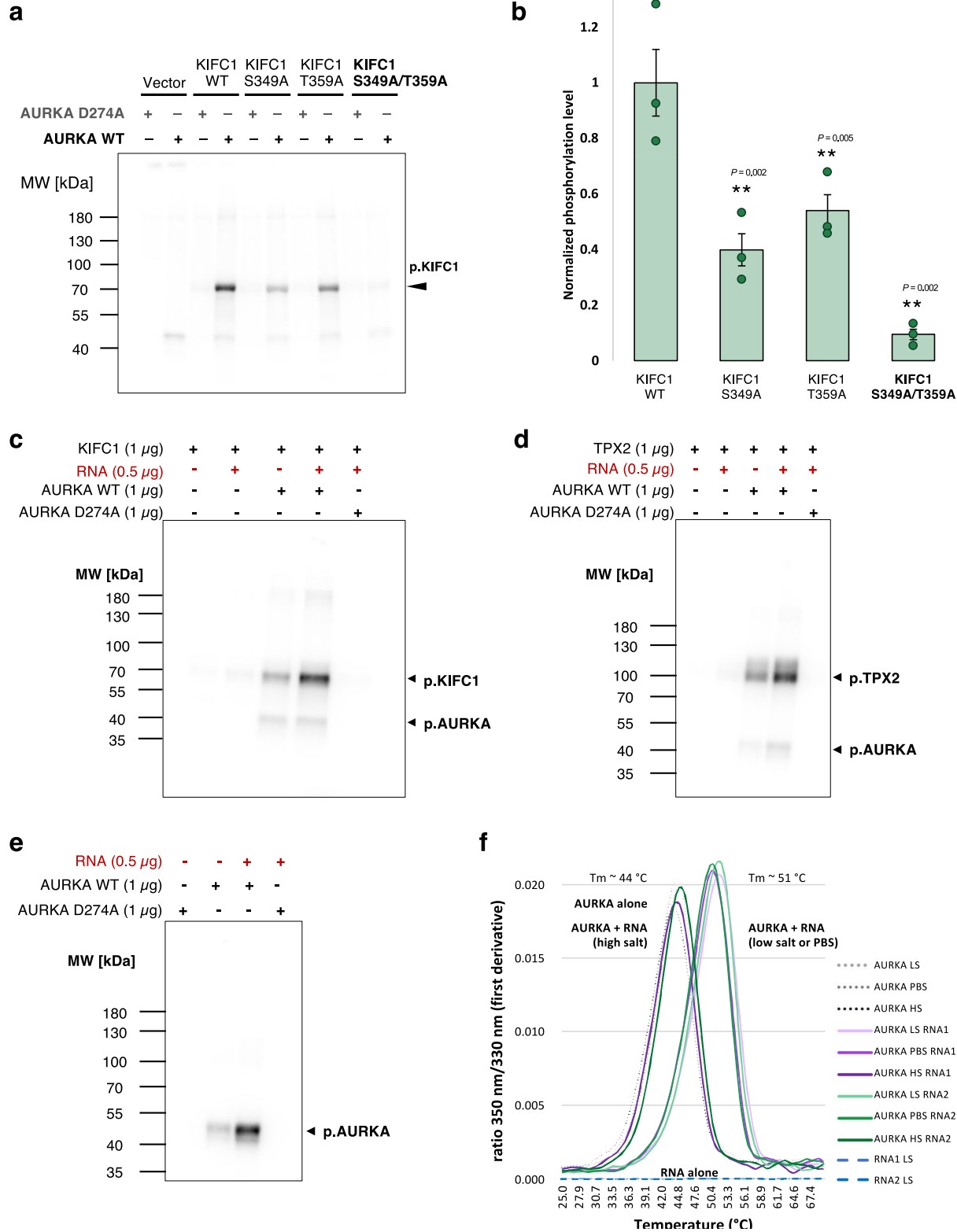

dynamic RNP assemblies are mainly seen as a mean for proteins to control the fate of cellular RNAs, from transcription to degradation. However, this view is being challenged by the expanding concept of riboregulation, according to which RNA transcripts bind to RBPs and regulate their localization, conformation, interactions and function[5,7,8,80].

Here, we took advantage of the R-DeeP strategy to investigate RNA-dependent proteins in a cell cycle-dependent manner. A protein is termed "RNA dependent" if its interactome is dependent on the presence of RNA. We identified 7152 and 7069 RNA-dependent proteins in mitosis and interphase, respectively, including hundreds of proteins that had not been previously related to RNA. Our screens highlighted both similarities and differential behaviors of the proteins in mitosis and interphase, which can be explored within the R-DeeP 3.0 database (R-DeeP3.dkfz.de), as well as compared to previous datasets from R-DeeP screens in unsynchronized HeLa[28] and A549 cells[30]. Limitations

**Fig. 8 | The kinase activity of AURKA is RNA dependent. a** Autoradiography indicating the phosphorylation intensity of KIFC1 in wild-type (WT) and non-phosphorylatable KIFC1 mutants in the presence of purified AURKA WT or AURKA kinase-dead mutant (D274A). One representative image is shown ($N = 3$).
**b** Quantification of the autoradiography image (as in **a**), representing the KIFC1 phosphorylation signal in the form of a bar graph with SEM ($N = 3$). The phosphorylation signal was normalized to the KIFC1 WT signal. $P$-values were calculated using two-tailed, paired t-test (**$P$-value < 0.01). **c** Autoradiography indicating the phosphorylation of KIFC1 by AURKA (lane 3) and the substantial increase of the phosphorylation signal in the presence of total RNA (lane 4). KIFC1 WT protein with or without RNA and AURKA kinase-dead mutant (D274A) were used as negative control (lanes 1, 2 and 5). One representative image is shown ($N = 3$).
**d** Autoradiography indicating the phosphorylation of TPX2 by AURKA and the increase in the phosphorylation signal in the presence of total RNA. Same as in **c** ($N = 3$). **e** Autoradiography indicating the autophosphorylation of AURKA and the

increase in the autophosphorylation signal in the presence of total RNA. AURKA kinase-dead mutant (D274A) was used as a negative control ($N = 3$). **f** Graph showing the stabilization of AURKA upon addition of total RNA as indicated by a shift towards higher temperatures of AURKA melting temperature Tm (light green and purple curves). Protein unfolding upon exposition to an increasing temperature gradient was monitored by analyzing the fluorescence emission intensities at 350 nm and 330 nm (the first derivative of the ratio F350/F330 is shown). AURKA alone and AURKA in presence of RNA in high-salt conditions (gray dotted curves, dark green and dark purple curves) had similar Tm (around 44 °C). Two different total RNA preparations were tested (RNA1 and RNA2) resulting in an increased Tm (around 51 °C). No changes in ratio were observed with RNA alone in low-salt conditions (blue curves). HS high salt, 750 mM NaCl, PBS phosphate-buffered saline, 137 mM NaCl, LS low salt, 10 mM NaCl. Source data for blots and graphs are provided as Source Data files.

of the R-DeeP strategy intrinsically linked to the preparation of the samples (use of detergent, ultracentrifugation) are discussed elsewhere[29]. The cut-off values applied in the analysis pipeline can result in absence of information for some proteins or false negative results. One example is the lack of data for AURKB in mitosis, a subunit of the chromosomal passenger complex (CPC) together with INCENP, Borealin and Survivin, known to interact with RNA[47,81]. The reason for this negative result remains unclear, but it seems that the CPC complex is challenging for proteome-wide RBP studies in general, as in 43 human studies, only INCENP was experimentally detected once[24] and AURKB only predicted once in silico[26]. Here, AURKB and INCENP were detected as RNA-dependent proteins in the interphasic R-DeeP screen. However, a strength of the R-DeeP methodology is its flexibility and adaptability to the need of the users. It is possible to increase the resolution of the gradient by using different sucrose concentration ranges. For example, the glycolytic enzyme ENO1 was a false negative when using gradient ranging from 5% to 50%. However, its RNA dependence became evident when using a more focused gradient ranging from 5% to 25%[7].

By integrating our results with the data from the RBP2GO database[11], we compiled an atlas of RNA-dependent proteins in cell division. The atlas reveals the localization of RBPs in various mitotic structures and their association with the mitotic cell cycle (Fig. 3 and Supplementary Data 2). For example, our atlas reports a great number of midbody-associated proteins as RNA-dependent proteins, in agreement with a recent work describing midbodies as assembly sites of RNP granules[82]. It also comprises a substantial number of centrosome-associated proteins, in line with previous debated studies reporting the presence and functional importance of RNA at the centrosomes. The potential role of RNA at the centrosome has been discussed[51] but not further experimentally studied. In addition, our atlas highlights 153 RNA-dependent proteins that were identified in the mitotic R-DeeP screen for which quantitative profiles can be downloaded from the R-DeeP 3.0 database. Such an atlas could be easily adapted to any other cellular processes of interest, using selected GO terms of interest within the advanced search option of the RBP2GO database (RBP2GO.dkfz.de). Altogether, this illustrates the utilities of integrating proteome-wide datasets into a central database, and to develop targeted tools that increase the functionality of the database as much as the impact of the included studies and datasets.

Over the past decades, we have been accustomed to the concept that a protein is regulated in terms of localization and functions through post-translational modifications that are mediated through protein-protein interactions. However, with the emergence of the concept of riboregulation, we have the opportunity to rethink this model and consider the possibility that proteins could also be regulated through interactions with RNA, especially in unexpected pathways such as cell division[5] - unexpected since one might not expect major activity of RNA and RBPs given the global transcriptional and

translational repression during cell division[83–85]. On the contrary, we identified multiple well-characterized major mitotic players such as AURKA, TPX2 and KIFC1 as RBPs, more specifically as unconventional RBPs. This raised the question of the functional meaning of their interactions with RNA.

AURKA is one of the major regulators of the cell cycle, and an oncogene involved in tumorigenesis in several types of cancers that promotes centrosome amplification and tumor growth[53,58,86,87]. It has evolved as an important anti-cancer target with several small molecule inhibitors being in clinical trials[88,89]. Here, we investigated the RNA dependence of AURKA and its interactions. While the interaction of some interactors to AURKA was RNase-insensitive, other interactions were lost upon RNase treatment. Also, we revealed that 90% of AURKA interactors were classified as RNA-dependent proteins themselves. We demonstrated that the interaction of AURKA with TPX2, an essential mitotic interactor, was RNA dependent (Fig. 5). This result may appear unexpected as a complex between AURKA and TPX2 had been crystalized[90]. However, the crystal structure mentioned here (www. rcsb.org, PDB code 1OL5) had been obtained using a truncated from of AURKA missing the first 121 aa of the N-terminal disordered region (AURKA 122-403) and a truncated version of TPX2 consisting of the first 43 aa of the N-terminal region (TPX2 1-43). In this context, it is conceivable that the interaction between truncated variants of the proteins happens in absence of RNA molecules, e.g. if the RNA would mainly affect the disordered region that is absent in the truncated form. Another study reported the activation of AURKA by TPX2 using purified proteins[49], without mentioning RNA as being part of the mechanism. However, there is also no mention of RNase treatment in this study, so that a role of RNA molecules in principle cannot be ruled out.

Our work uncovered the interaction of both AURKA and TPX2 with KIFC1, which had not been studied before. These interactions were also RNA dependent. Interestingly, the mitotic R-DeeP profile of AURKA, TPX2 and KIFC1 depicted similar peaks around fraction 21 of the control gradient, indicating that they co-migrated to the same fraction, probably as one complex in the presence of RNA. Together with the proximity labeling assays, our findings strongly suggest that the three proteins are part of the same complex, relying on RNA to be assembled. In view of the recent literature and the respective role of the proteins, one could speculate about a role for KIFC1 together with AURKA and TPX2 in microtubule nucleation, branching and/or spindle pole organization[64,91–94]. However, KIFC1 is described normally as a non-essential kinesin motor[95], so that a functionality of such a complex could be limited to specific cells such as cancer cells presenting centrosomal amplification or meiotic cells[95,96]. Therefore, future studies will aim to identify these cell types with AURKA-TPX2-KIFC1 essentiality and characterize the functional relevance of the two AURKA phosphorylation sites on KIFC1 at $S^{349}$ and $T^{359}$, that we revealed here. Possible model systems may also involve the *Xenopus*

*laevis* egg extract, e.g., to study the relevance of KIFC1 phosphorylation in absence of centrosomes. While the functional importance of KIFC1 phosphorylation by AURKA still needs to be further investigated, AURKA phosphorylation of TPX2 has been implicated in the regulation of metaphase spindle length[97], but is also not fully understood.

A fundamental step in understanding the molecular and cellular functions of RBPs is to identify their interacting RNAs. After demonstrating that AURKA, TPX2 and KIFC1 directly bound to RNA, we analyzed per iCLIP the KIFC1-interacting RNAs and found predominantly rRNA and protein-coding transcripts, lacking specific binding site sequences though. Large amounts of rRNA can be interpreted as contamination, a reason why rRNA is very often omitted from further analyses. In addition, a recent study described CLIP data as being potentially noisy, especially in the context of nuclear factors such as PRC2 and CTCF[98]. While this could also affect the CLIP data in our study, recent publications have pointed to the enrichment of rRNA around the mitotic chromosomes and highlighted their role in mediating chromosome clustering[99–101]. Furthermore, rRNAs are also associated with the microtubules and mitotic spindles[39,42]. These data indicate an essential role of rRNA in spindle assembly and mitotic progression, although the molecular mechanisms at play remain unclear. Similarly, mRNAs are attached to the spindle, often mRNAs of mitotic factors, consistent with spindle-localized protein synthesis[37,38,102,103]. In addition, a study also pointed to the role of RNAs in regulating protein localization to the mitotic spindle[104]. Altogether, those previous findings confirm that rRNA and mRNA species are found at the mitotic microtubules and spindle and can effectively interact with mitotic factors, as indicated from our KIFC1 iCLIP data. Complementary analyses using orthogonal strategies will be needed to validate our findings.

One of the main findings of the present study is the stimulation of AURKA kinase activity in presence of total RNA, i.e., a mixture of rRNA, mRNA and other RNA species, as seen from the increased phosphorylation of both, KIFC1 and TPX2 (Fig. 8). This points to the functional relevance of the interaction of these proteins with RNA in terms of phosphorylation efficiency. RNA molecules can act by stabilizing the conformation of the proteins, potentially their disordered regions. Using nanoDSF, we indeed could reveal a stabilization effect of total RNA on the conformation of AURKA (Fig. 8f). This raises the question whether the increased kinase activity in presence of RNA could be directly linked to the RNA-induced stabilization of AURKA conformation.

AURKA, TPX2 and KIFC1 belong to the class of unconventional RBPs, because they lack known RBDs. Currently, the RNA-binding characteristics of such proteins are not well understood, but a recent study reported the lack of sequence specificity for the great majority of the 492 investigated unconventional RBPs[105] - well in line with our findings for KIFC1. The same study also claimed that these unconventional RBPs were often highly abundant proteins and suggested that their identification in RNA interactome studies could occur via weak non-sequence-specific interactions with RNA. Thus, it might not be surprising that we did not identify any sequence specificity in the binding sites of the KIFC1-interacting RNAs. In addition, AURKA, KIFC1 and TPX2 all have high isoelectric points due to their positively charged amino acids and contain intrinsically disordered regions[54,106,107]. Both aspects facilitate their binding to RNA. Thus, one can speculate that unspecific interaction with RNA possibly stabilizes their structure via a disorder-to-order transition, which in turn promotes interactions with other molecules, such as proteins or RNAs[108,109] or stimulates their function, as seen here for AURKA. However, further targeted investigations of the RNA-protein interactions, e.g. for selected unconventional mitotic RBPs, will certainly help understanding the driving mechanisms, that might not be simply limited to sequence characteristics.

For nuclear proteins such as TPX2 and KIFC1[62,110], the proteins can already interact with nuclear-retained mRNAs throughout interphase. However, at the onset of cell division, they are suddenly exposed to the highly abundant cytoplasmic pool of both mRNA and rRNA transcripts[111,112] upon nuclear envelope breakdown. In the context of riboregulation, it is conceivable that multiple RNA transcripts collectively "crowd-control" AURKA, KIFC1 and TPX2 interactions. This hypothesis may be also expanded to other RBPs in mitosis and may provide the starting point for a more general concept of crowd-controlled riboregulation upon nuclear envelope breakdown in mitosis to be further investigated in the future. This idea also fits very well to the concept of self-organization, defined as the emergence of an ordered pattern in space and time, as the result of the collective interactions of individual constituents. Noteworthy, it is well-established, that such physical principle is the basis of the "self-organized" mitotic spindle[113] and other key cellular structures[114].

In sum, our study provides an atlas of RNA-dependent mitotic factors, suggesting an important role of RNA in orchestrating multiple aspects of cell division. Our detailed analysis of the RNA dependence of AURKA, TPX2 and KIFC1 reveals the importance of RNA in riboregulating mitotic protein-protein interactions. More specifically, we demonstrate that RNA stimulates AURKA kinase activity and stabilizes its conformation, which can have implication for the activation of its substrates. Our Atlas of RNA-dependent mitotic proteins points to the high regulatory potential of RNA in many substructures of the dividing cells. This view is supported by past studies but still a matter of debate[42–44,51,104]. To address these challenging questions within the native cellular environment, dedicated tools and new technologies will be needed. Thus, a completely new complexity level including mitotic protein-RNA interactions promises to advance our understanding of the molecular mechanisms regulating cell division.

## Methods
For catalog numbers and further details such as sequences and URL please refer to Supplementary Data 5.

### Gene ontology analysis
A gene ontology (GO) enrichment analysis was performed using the GO enrichment analysis tool from the Gene Ontology Resource[115] on RNA-dependent proteins as identified in the R-DeeP screen performed in HeLa cells (1751 proteins in total)[28].

### Cell culture
Hela WT cells and HeLa WT cells stably expressing GFP-α-tubulin[116] were grown in DMEM high glucose medium supplemented with 10% FBS and A549 WT cells were grown in RPMI 1640 medium supplemented with 10% FBS in a humidified incubator at 37 °C with 5% CO2. Both cell lines were obtained from the lab of Prof. Diederichs, authenticated and frequently tested for the absence of mycoplasma contamination.

### Cell synchronization
On day 1, 2.5–3 million cells were seeded for synchronization in prometaphase and 2 million cells were seeded for synchronization at interphase in 15 cm dishes. On day 2, thymidine was added to the cells (2 mM final concentration) for 16 h to arrest the cells in S-phase. The cells were washed once with warm PBS and released with fresh media for 9 h. On day 3, thymidine was added again to the cells (2 mM final concentration) for 16 h to arrest the cells in S-phase (double thymidine block). To further synchronize the cells in prometaphase, the cells were washed once with PBS and fresh media was added to that contained 100 ng/ml nocodazole. Finally, after 12 h, the cells were harvested, flash frozen and stored at −80 °C until lysate preparation.

To synchronize cells at metaphase, 5 million cells were seeded on a 15 cm plate on day 1. On day 2, thymidine was added to the cells

(2 mM final concentration) for 24 h to arrest the cells in S-phase. On day 3, cells were washed once with warm PBS and further incubated with fresh medium containing 40 ng/ml nocodazole for 12 h. Cells were washed three times with warm PBS and released with fresh media for 40 min. Finally, the cells were harvested, flash frozen and stored at −80 °C until lysate preparation.

## R-DeeP screen

*Sucrose density gradients*: the cells were cultured and synchronized in prometaphase and interphase as stated above. The gradients, cell lysate preparation, RNase treatment, ultracentrifugation and fractionation were performed as previously published[28,29]. Briefly, the cell lysates remained either untreated (control gradients) or treated with an RNase cocktail (RNase-treated gradients), loaded on the sucrose gradients and subjected to ultracentrifugation. The gradients were fractionated to 25 different fractions, and further analyzed either via mass spectrometry (proteome-wide analysis on a Fusion Orbitrap Lumos mass spectrometer) or Western blot analysis (individual candidate analysis).

*Bioinformatic analysis*: mass spectrometry datasets were analyzed as explained in detailed in our published protocol[29].

## SDS-PAGE and Western blot analysis

Sodium dodecyl sulfate–polyacrylamide gel electrophoresis (SDS-PAGE) was performed to separate proteins based on their molecular weight. The samples in 1× SDS sample buffer (30% (v/v) glycerol, 12% (w/v) SDS, 3.6 M DTT, 0.012% (w/v) bromophenol blue, and 500 mM Tris-HCl (pH 6.8)) or 1x LDS buffer were boiled at 95 °C for 5 min or 70 °C for 10 min respectively, and briefly spun down. Next, the samples were loaded onto appropriate Bio-Rad pre-cast protein gels and run at 120 V in the electrophoresis chamber containing 1x SDS Running buffer (25 mM Tris base, 192 mM glycine, 0.1% (w/v) SDS).

Western blot analysis was performed on nitrocellulose membrane in a Trans blot turbo wet transfer system using 1x Trans-Blot Turbo Transfer Buffer containing 20% ethanol (mixed molecular weight program). The membrane was blocked with 5% milk in Tris-buffered saline (blocking solution: 24.7 mM Tris-HCl (pH 7.4), 137 mM NaCl, 2.7 mM KCl) containing 0.05% Tween-20 (TBST) for 1 h at room temperature (RT). Furthermore, the membrane was incubated at 4 °C overnight with the respective antibodies. The following day, the membrane was washed three times with TBST for 5 min at RT and incubated with the appropriate HRP-conjugated secondary antibody at 1:5000 dilution in blocking solution for 1 h at RT. The membrane was washed three times with TBST for 5 min at RT. Finally, the membrane was incubated with ECL reagent for 5 min and the blots were imaged using an INTAS ECL Chemocam imager. Quantitative analysis of Western blot images was performed using Fiji (Image J) software.

## AURKA immunoprecipitation (IP) followed by LC-MS/MS-based protein analysis

**AURKA immunoprecipitation.** Hela cells synchronized in prometaphase were used for immunoprecipitation. The cell pellets were lysed in three pellet volume of lysis buffer (50 mM HEPES-KOH, pH-7.5, 150 mM KCl, 0.5% NP-40, 2 mM EDTA, 1 mM NaF, 0.5 mM DTT and 1x complete EDTA free protease inhibitor cocktail), incubated on ice for 30 min and centrifuged at 17,000 × *g* for 20 min at 4 °C. The supernatant was transferred to fresh tubes and centrifuged again for 17,000 × *g* for 20 min at 4 °C. The supernatant was stored in a fresh tube on ice until the beads were prepared for the pre-clearing step (see below).

30 μl of pierce ChIP-grade protein A/G magnetic beads were used per sample in the IP. The beads were aliquoted to a fresh 1.5 ml tube and washed three times with 1 ml lysis buffer. Next, the lysate was added on the beads and incubated for 1 h at 4 °C on a rotator to remove unspecific binding to the beads. After 1 h, the beads were removed from the lysate on a magnetic stand and the pre-cleared lysate was transferred to fresh 1.5 ml tubes. A BCA assay was performed to measure the protein concentration.

The lysate was split into 2 different samples (containing 4 mg total lysate each) for overnight protein-antibody complex formation at 4 °C on a rotator. Here, 0.8 μg AURKA antibody was used for AURKA IP and rabbit IgG was used as a negative control.

On the next day, beads were prepared by washing them three times in 1 ml lysis buffer. The washed beads were split in 2 tubes, each lysate-antibody mix was added to the beads and was incubated for 2 h at 4 °C on a rotator. After incubation, the beads were removed from the lysate on a magnetic stand and the flow through (FT) was discarded. The beads-antibody-protein complexes were washed three times with 1 ml wash buffer I (50 mM HEPES-KOH, pH-7.5, 150 mM KCl, 0.5% NP-40, 0.5 mM DTT and 1x complete EDTA free protease inhibitor cocktail). Within the last wash step, each tube was split into two tubes for control and RNase treatment (total 4 tubes). Using a magnetic stand, the supernatant was discarded and the beads were resuspended with 100 μl wash buffer I.

For the control tubes, 10 μl wash buffer I was added to the sample. For the RNase-treated samples, 10 μl RNase cocktail was added (RNase cocktail: equal volume of RNase A, RNase I, RNase III, RNase H and RNase T1) and incubated for 1 h at 4 °C on a rotator. After incubation, the beads were captured using a magnetic stand and FT were collected for LC-MS/MS-based protein analysis. The beads were washed three times with wash buffer II (50 mM HEPES-KOH, pH-7.5, 300 mM KCl, 0.5% NP-40, 0.5 mM DTT and 1x complete EDTA free protease inhibitor cocktail). Finally, the protein complexes were eluted using 30 μl 1x LDS containing 100 mM DTT and were analyzed using SDS-PAGE/Western blot and LC-MS/MS-based protein analysis at Proteomics Core Facility (Mass spectrometry-based protein analysis unit).

**Protein digestion of AURKA IP samples for LC-MS/MS analysis.** IP eluates were run for 0.5 cm into an SDS-PAGE and the entire piece was cut out and digested using trypsin according to Shevchenko et al.[117] adapted on a DigestPro MSi robotic system (INTAVIS Bioanalytical Instruments AG).

**LC-MS/MS analysis of AURKA IP.** The LC-MS/MS analysis was carried out on an Ultimate 3000 UPLC system (Thermo Fisher Scientific) directly connected to an Orbitrap Exploris 480 mass spectrometer for a total of 120 min. Peptides were online desalted on a trapping cartridge (Acclaim PepMap300 C18, 5 μm, 300 Å wide pore; Thermo Fisher Scientific) for 3 min using 30 μl/min flow of 0.05% TFA in water. The analytical multistep gradient (300 nl/min) was performed using a nanoEase MZ Peptide analytical column (300 Å, 1.7 μm, 75 μm × 200 mm, Waters) using solvent A (0.1% formic acid in water) and solvent B (0.1% formic acid in acetonitrile). For 102 min the concentration of B was linearly ramped from 4% to 30%, followed by a quick ramp to 78%, after two minutes the concentration of B was lowered to 2% and a 10 min equilibration step appended. Eluting peptides were analyzed in the mass spectrometer using data dependent acquisition (DDA) mode. A full scan at 120k resolution (380–1400 m/z, 300% AGC target, 45 ms maxIT) was followed by up to 2 seconds of MS/MS scans. Peptide features were isolated with a window of 1.4 m/z, fragmented using 26% NCE. Fragment spectra were recorded at 15k resolution (100% AGC target, 54 ms maxIT). Unassigned and singly charged eluting features were excluded from fragmentation and dynamic exclusion was set to 30 s. Each sample was followed by a wash run (40 min) to minimize carry-over between samples. Instrument performance throughout the course of the measurement was monitored by regular (approx. one per 48 h) injections of a standard sample and an in-house shiny application.

**Data analysis of LC-MS/MS data.** Data analysis was carried out by MaxQuant version 1.6.14.0[118] using an organism specific database extracted from Uniprot.org (human reference database, containing 74,811 unique entries from 27th February 2020). Settings were left at default with the following adaptions. Match between runs (MBR) was enabled to transfer peptide identifications across RAW files based on accurate retention time and m/z. Fractions were set in a way that MBR was only performed within replicates. Label free quantification (LFQ) was enabled with default settings. The iBAQ-value[119] generation was enabled. Peptides from AURKA interactors were analyzed using the log2(iBAQ) values from four replicates. Replicate four was excluded from the analysis due to unproper clustering pattern as compared to the other three replicates. Three conditions were analyzed (IgG IP, AURKA IP, AURKA IP RNase treatment in three replicates, 9 samples in total). Interactors were filtered and only further analyzed if they were detected in at least 70% of the samples. Missing values were imputed using random values based on a gaussian distribution centered around the median of the sample and outliers were adjusted based on the mean method. Ratios between the (AURKA IP)/(IgG) samples and (AURKA IP)/(AURKA IP RNase treatment), i.e. differences of the log2(iBAQ) values were calculated for each replicate and adjusted p-values were computed by applying a t-test, corrected for multiple testing (FDR method). AURKA interactors were selected based on an at least two-fold increased (AURKA IP)/(IgG) ratio (adjusted p-values < 0.05). RNase sensitive AURKA interactors were identified based on an at least two-fold increased (AURKA IP)/(AURKA IP RNase treatment) ratio (adjusted p-values < 0.05).

**AURKA immunoprecipitation (IP)**
HeLa and A549 cells synchronized in prometaphase or metaphase as explained above were used for immunoprecipitation followed by Western blot analysis. Each cell pellets were lysed in 2 ml lysis buffer (50 mM HEPES-KOH, pH-7.5, 150 mM KCl, 0.5% NP-40, 2 mM EDTA, 1 mM NaF, 0.5 mM DTT and 1x complete EDTA free protease inhibitor cocktail) as explained in the previous section. BCA assay was performed to measure the protein concentration in the lysate. The lysate was diluted to 2 mg/ml with lysis buffer and the total lysate was split into 4 tubes containing 2 mg lysate each.

2 μl of turbo DNase were added to each tube. 10 μl of lysis buffer was added to control samples while, 10 μl RNase I was added to RNase-treatment samples and were incubated at 37 °C for 3 min at 1100 rpm in a thermomixer to digest the DNA and RNA in the lysates. The samples were cooled down by incubating on ice for 3 min. Further, the samples were centrifuged at 17,000 × g for 20 min at 4 °C and the supernatants were transferred to fresh tubes. The lysates were filtered through a proteus clarification mini spin column by centrifuging at 16,000 × g for 2 min at 4 °C. The filtered lysates were then transferred to fresh 2 ml tubes and kept on ice until the beads were prepared for the pre-clearing step (see previous section).

The lysates were incubated with the respective antibodies overnight at 4 °C on a rotator to form protein-antibody complexes. For AURKA IP: 0.4 μg AURKA antibody per IP and rabbit IgG was used as a negative control. For TPX2 IP: 1.5 μg TPX2 antibody per IP and mouse IgG1 was used as a negative control.

On the next day, the beads were prepared by washing three times in 1 ml lysis buffer. The washed beads were added to the lysate-antibody mix and incubated for 2 h at 4 °C on a rotator. After incubation, the beads were removed from the lysate on a magnetic stand and the FT was discarded.

For AURKA IP: the beads-antibody-protein complexes were washed three times with 1 ml wash buffer (HeLa: 50 mM HEPES-KOH, pH-7.5, 150 mM KCl, 0.5% NP-40, 0.5 mM DTT and 1x complete EDTA free protease inhibitor cocktail, A549: 50 mM HEPES-KOH, pH-7.5, 15 mM KCl, 0.5% NP-40, 0.5 mM DTT and 1x complete EDTA free protease inhibitor cocktail). Later, the beads were resuspended in 20 μl

lysis buffer, 2 μl RNase I and incubated at 37 °C for 3 min at 1100 rpm in a thermomixer. Lastly, the samples were eluted by adding 7.5 μl of 4x LDS (with 200 mM DTT) and boiling at 70 °C for 10 min.

For TPX2 IP: the beads-antibody-protein complexes were washed three times with 1 ml wash buffer (50 mM HEPES-KOH, pH-7.5, 300 mM KCl, 0.5% NP-40, 0.5 mM DTT and 1x complete EDTA free protease inhibitor cocktail). Finally, the samples were eluted by adding 30 μl of 1x LDS (with 100 mM DTT) and boiling at 70 °C for 10 min.

The samples were stored in fresh tubes at −20 °C until SDS-PAGE/Western blot analysis.

**Proximity Ligation Assay (PLA)**
All the buffers and solutions were provided in the Duolink® in-situ PLA kit.

120,000 cells/well were seeded on a coverslip in a 12-well plate and were allowed to grow overnight. On the following day, cell medium was aspirated, washed once with warm PBS and fixed with either methanol or 4% PFA (depending on the antibody) for 10 min or 15 min respectively and RT. After fixation, the cells were permeabilized in 0.25% Triton in PBS for 10 min at RT. After permeabilization, cells were washed twice with warm PBS and blocked with 40 μl Duolink® blocking solution for 1 h at 37 °C in a heated humidity chamber. Meanwhile, primary antibodies were diluted in the Duolink® antibody diluent to appropriate concentrations. After blocking, the cells were incubated with primary antibody overnight at 4 °C in a humidity chamber. All subsequent steps were performed following the instructions as provided with the Duolink® in-situ PLA kit. Mouse or rabbit secondary antibody Alexa Fluor 488 was added to the amplification mix in a 1:500 dilution.

For PLA +/− RNase: 200,000 cells/well were seeded on a coverslip in a 12-well plate. The cells were synchronized in metaphase based on the protocol described above. Following the synchronization, the media was removed, and the cells were first treated with 0.1% Triton in PBS with RNase (100 μg/ml) and without RNase for control slides for 30 s at RT. Further, the cells were fixed with methanol at RT for 10 min, washed once with warm PBS and permeabilized with 0.25% Triton for 10 min at RT. After the permeabilization, the cells were washed twice with warm PBS and the rest of the PLA protocol from blocking step was followed as described above.

*Imaging*: images were acquired on a Zeiss LSM 980 Airyscan NIR, in confocal acquisition mode (best signal setting) or on a Zeiss LSM 710 ConfoCor 3, both equipped with diodes for the excitation of DAPI (405 nm), Alexa 488 (488 nm) and Alexa 594 (561 nm/555 nm) fluorophores. Samples within one replicate were all acquired with the same settings. Z-stacks were acquired and maximal projection images (512 ×512 pixels, 8 bits) were analyze using Fiji (ImageJ) software. For intensity calculation, background pixel values up to a value of 10 were removed.

**Individual-nucleotide resolution UV Cross-linking and Immunoprecipitation (iCLIP2)**
*Cell culture*: frozen pellets from HeLa and A549 cells synchronized in prometaphase (AURKA and KIFC1) or metaphase (TPX2) were used for the iCLIP2 assay. *Reverse transfection*: for iCLIP2 siRNA control versus siRNA AURKA, 3 million HeLa cells were plated out on a 15 cm dish on day 1. On day 2, they were reverse transfected with siRNA (siRNA NegC as control and siPOOL AURKa-c1) using lipofectamine RNAiMAX, based on the manufacturer's instructions. After 6 h, thymidine was added to the cells (2 mM final concentration) for 24 h to arrest the cells in S-phase. To further synchronize the cells in prometaphase, the cells were washed once with PBS and fresh media was added to that contained 40 ng/ml nocodazole. Finally, after 14 h, the cells were harvested, flash frozen and stored at −80 °C until lysate preparation. *Transfection and synchronization*: 5 million HeLa cells were plated out on 15 cm plates. On day the next day, they were transfected with

plasmids (EGFP-delta-Cm/ccdB as empty vector control and EGFP-AURKA) using lipofectamine 2000 at a DNA:lipofectamine ratio of 1:2.5, based on the manufacturer's instructions. After 6 h, the plates were washed once with warm PBS and fresh media containing 2 mM Thymidine was added to the cells. After 24 h, the cells were washed once with warm PBS and fresh media containing 40 ng/ml Nocodazole was added. After approximately 14 h, the cells were UV cross-linked (254 nm, 200 mJ/cm$^2$) or not crosslinked and harvested. The pellets were flash frozen and stored at −80 °C until further use.

*Beads-antibody preparation*: the antibodies were conjugated to beads first. Dynabeads Protein A for AURKA/KIFC1 and Dynabeads protein G beads for TPX2 iCLIP2 (100 µl per IP) were washed three times with 1 ml lysis buffer. After the last wash the beads were resuspended in 500 µl (AURKA/KIFC1 iCLIP2) or 400 µl (TPX2 iCLIP2) lysis buffer and split into two tubes, one for the IgG control (100 µl) and one for the protein of interest AURKA/KIFC1/TPX2 (400 µl/300 µl), and further incubated with the antibodies (IgG/AURKA/KIFC1: 2 µg per IP and IgG$_1$/TPX2: 8 µg per IP) for 1 h at RT on a rotator (10 rpm). The bead–antibody complexes were captured on a magnetic rack and washed once with 1 ml high-salt wash buffer (50 mM Tris-HCl pH 7.4, 1.5 M NaCl, 1 mM EDTA pH 8.0, 1% Igepal CA-630, 0.1% SDS, 0.5% sodium deoxycholate) and twice with 1 ml lysis buffer. The beads were then resuspended in 100 µl lysis buffer for IgG or 400 µl/ 300 µl lysis buffer for AURKA, KIFC1 and TPX2 pulldown. Alternatively (EGFP-AURKA iCLIP2), the GFP-Trap Magnetic Agarose beads (50 µl per IP) were washed three times with 1 ml lysis buffer and resuspended in 1 ml lysis buffer. The beads were distributed equally to 2 ml low-bind tubes (1 tube per IP). The beads were captured on a magnetic rack and the supernatant was removed just before adding the lysates.

**Cell lysis**. UV cross-linked (254 nm, 200 mJ/cm$^2$) and non-crosslinked cells were lysed in 2 ml lysis buffer per cell pellet (50 mM Tris-HCl pH 7.4, 100 mM NaCl, 1% Igepal (CA-630), 0.1% SDS, 0.5% Sodium deoxycholate, 1x protease inhibitor cocktail). BCA assay was performed to measure the protein concentration and the lysate was diluted to 2 mg/ml (AURKA/KIFC1) and 5 mg/ml (TPX2). The lysate was distributed into different 1.5 ml low-bind tubes containing 1 ml total lysate. The lysates were treated with 4 µl turbo DNase and different RNase I dilutions ranging from 1:5 to 1:1000 for AURKA, 1:5 to 1:500 for KIFC1 and 1:5 to 1:50 for TPX2. The RNase and DNase treatments were performed at 37 °C at 1100 rpm on a thermomixer for 3 min. Next, the samples were incubated on ice for 3 min and centrifuged at 17,000 × g for 20 min at 4 °C. The supernatant was collected into a fresh 2 ml low-bind tube and filtered through proteus clarification mini spin column by centrifuging at 16,000 × g for 2 min at 4 °C. Further, the filtered lysates were transferred to a fresh 2 ml low-bind tubes and kept on ice. Alternatively (EGFP-AURKA iCLIP2), UV cross-linked (254 nm, 200 mJ/cm$^2$) and non-crosslinked cells were lysed in 1 ml lysis buffer per cell pellet (50 mM Tris-HCl pH 7.4, 100 mM NaCl, 1% Igepal (CA-630), 0.1% SDS, 0.5% Sodium deoxycholate, 1x protease inhibitor cocktail, 1x PhosSTOP). BCA assay was performed to measure the protein concentration and the lysate was diluted to the lowest concentration (7.4 mg/ml). The lysate was distributed into different 1.5 ml low-bind tubes containing 1 ml total lysate. The lysates were treated with 4 µl turbo DNase and different RNase I dilutions (1:5 or 1:50). The RNase and DNase treatments were performed at 37 °C at 1100 rpm on a thermomixer for 3 min. Next, the samples were incubated on ice for 3 min and centrifuged at 17,000 × g for 20 min at 4 °C. The supernatant was collected into a fresh 2 ml low-bind tube and filtered through a proteus clarification mini spin column by centrifuging at 16,000 × g for 2 min at 4 °C. Further, the filtered lysates were transferred to a fresh 2 ml low-bind tubes and kept on ice.

**Pulldown/Immunoprecipitation**. 100 µl of the resuspended beads were added to the respective tubes containing cleared lysate and incubated for 2 h rotating at 4 °C for immunoprecipitation. After 2 h, the complex was captured on a magnetic rack. The FT was removed and the beads were washed twice with 1 ml high-salt wash buffer with rotation at 10 rpm at 4 °C for 1 min and then washed twice with 1 ml PNK wash buffer (KIFC1:20 mM Tris-HCl pH 7.4, 10 mM MgCl$_2$, 0.2% Tween-20, AURKA/ TPX2: 20 mM Tris-HCl pH 7.4, 10 mM MgCl$_2$, 0.2% Tween-20, PhosStop). During the last wash, the beads were transferred to fresh 1.5 ml low-bind tubes and stored at 4 °C. Alternatively (EGFP-AURKA iCLIP2), the lysates were added to the tubes containing the beads and incubated 1 h rotating at 10 rpm at 4 °C for immunoprecipitation. After 1 h, the complex was captured on a magnetic rack. The FT was removed and the beads were washed twice with 1 ml high-salt wash buffer with rotation at 10 rpm at 4 °C for 1 min and then washed twice with 1 ml PNK wash buffer (20 mM Tris-HCl pH 7.4, 10 mM MgCl$_2$, 0.2% Tween-20, 1x Protease Inhibitor Cocktail, 1x PhosStop). The beads were stored at 4 °C until the following day.

On the following day, for AURKA/TPX2 iCLIP2: the samples were placed on magnetic stand, PNK buffer containing PhosStop was removed and the beads were resuspended in 1 ml PNK buffer (20 mM Tris-HCl pH 7.4, 10 mM MgCl$_2$, 0.2% Tween-20) without PhosStop.

Further, 100 µl of beads were transferred to a fresh tube for Western blot and 900 µl were used for labeling the RNA. For Western blot, beads were captured, the supernatant was removed, and the protein complexes were eluted using 1× LDS buffer containing 50 mM DTT at 70 °C for 10 min. The eluate was collected and stored at −20 °C to check for immunoprecipitation efficiency using Western blot analysis. The SDS-PAGE and Western blot analysis was performed as described earlier.

**RNA labeling with radioactive $^{32}$P**. RNA labeling was performed using the remaining 900 µl sample. The radioactive labeling of RNA using $^{32}$P was performed using a master mix containing 11.85 µl nuclease free water, 0.75 µl T4 PNK enzyme, 1.5 µl 10× PNK buffer, and 0.9 µl $^{32}$P-γ-ATP per sample. Beads were captured on ice, the supernatant was removed and the beads were resuspended in 15 µl PNK mix. The samples were incubated on a thermomixer at 37 °C for 5 min at 1100 rpm for labeling the RNA. Later, the samples were washed twice with 1 ml PNK wash buffer to get rid of excess radioactivity and eluted in 25 µl 1× LDS buffer containing 50 mM DTT on a thermomixer at 70 °C for 10 min at 1100 rpm.

To visualize the RNA-labeling, SDS-PAGE and Western blot analysis were performed. The samples were loaded in a 7.5% Mini-PROTEAN® TGX™ precast protein gel and run 120 V in a vertical electrophoresis chamber filled with 1× SDS running buffer (25 mM Tris base, 192 mM glycine, 0.1% (w/v) SDS). Western blot was performed using 0.45 µm nitrocellulose membrane with a wet transfer system with transfer buffer (25 mM Tris base, 192 mM glycine) containing 20% methanol for 1.5 h at 120 V in an ice bath. Finally, the membrane was washed once in nuclease-free water, covered with plastic wrap and exposed to a phosphor imager screen and imaged using a Typhoon laser scanner phosphor imager at 200 µm, high speed and intensity 3.

### iCLIP2 library preparation and sequence analysis

The iCLIP2 library preparation was performed based on the publication "Improved library preparation with the new iCLIP2 protocol"[66]. For adapter, barcodes or primer sequences refer to Supplementary Data 5.

For KIFC1 iCLIP2 library preparation, UV-crosslinked prometaphase cells synchronized and harvested on four different dates were used and IgG was used as a negative control. Here, RNase treatment was performed on the lysate at 1:10 dilution. All the steps from lysate preparation, beads preparation, and pulldown were performed as described in the above section (Individual Nucleotide Resolution and UV Cross-linked Immunoprecipitation (iCLIP2).

**Dephosphorylation.** 3′ dephosphorylation of RNA was performed using the master mix containing 1x PNK buffer pH-6.5 (350 mM Tris-HCl, pH 6.5, 50 mM MgCl$_2$, 5 mM DTT), 0.5 μl SUPERase-In, 0.5 μl of T4 PNK enzyme) in a total volume of 15 μl per sample. The PNK buffer was removed from the samples, and the beads were resuspended in 20 μl of the dephosphorylation master mix and incubated for 20 min at 37 °C at 1100 rpm on a thermomixer. After the incubation, the beads were washed once with 1 ml PNK wash buffer (20 mM Tris-HCl pH 7.4, 10 mM MgCl$_2$, 0.2% Tween-20), twice with 1 ml high-salt wash buffer (50 mM Tris-HCl pH 7.4, 1.5 M NaCl, 1 mM EDTA pH 8.0, 1% Igepal CA-630, 0.1% SDS, 0.5% Sodium deoxycholate) for 2 min at 4 °C on a rotator and twice again with 1 ml PNK wash buffer (20 mM Tris-HCl pH 7.4, 10 mM MgCl$_2$, 0.2% Tween-20).

3′ adapter ligation: adapter ligation was performed using the following ligation mix containing (4x ligation buffer (200 mM Tris-HCl, pH 7.8, 40 mM MgCl$_2$, 4 mM DTT, PEG 400, 3 μl of L3-App-Fluo adapter (10 μM) (/rApp/AGATCGGAAGAGCGGTTCAG/ddC/), 0.5 μl SUPERase-In, 1 μl of T4 ligase in a total volume of 6.5 μl per sample). The PNK buffer was removed from the beads and the beads were resuspended in 20 μl ligation mix and incubated overnight at 16 °C at 1100 rpm on a thermomixer. The following day, the beads were washed once with 0.5 ml PNK wash buffer (20 mM Tris-HCl pH 7.4, 10 mM MgCl$_2$, 0.2% Tween-20), twice with 1 ml high-salt wash buffer (50 mM Tris-HCl pH 7.4, 1.5 M NaCl, 1 mM EDTA pH 8.0, 1% Igepal CA-630, 0.1% SDS, 0.5% sodium deoxycholate) for 2 min at 4 °C on a rotator and once again with 0.5 ml PNK wash buffer. During the last wash, the samples were transferred to a fresh 1.5 ml low-bind tubes and the beads were re-suspended in 1 ml PNK wash buffer. 100 μl of the sample were aliquoted for Western blot analysis. The remaining 900 μl of the samples were used for RNA extraction. The samples were placed on a magnetic stand, buffer was removed and the samples were eluted using 35 μl 1x LDS buffer, boiled at 70 °C for 10 min. The samples were loaded 7.5% Mini-PROTEAN® TGX™ precast protein gel and run at 120 V in a vertical electrophoresis chamber filled with 1× SDS running buffer (25 mM Tris base, 192 mM glycine, 0.1% (w/v) SDS). Western blot was performed using 0.45 μm nitrocellulose membrane using a wet transfer system with transfer buffer (25 mM Tris base, 192 mM glycine) containing 20% methanol for 1.5 h at 120 V on an ice bath.

**Proteinase K digestion.** Following the Western blot, the membrane was cut at 90–150 kDa to 4-5 small pieces and transferred into a 2 ml low-bind tube. Further the master mix (2x proteinase K buffer, 1 mg proteinase K) for proteinase K digestion was prepared. 400 μl of the mix was added to each tube with cut membrane, vortexed for 20 seconds and incubated for 1 h 30 min at 55 °C at 1000 rpm on a thermomixer.

**RNA extraction.** For RNA extraction, 2 volumes of acidic phenol-chloroform-IAA (pH 6.5–6.9) was directly added to the proteinase K digested samples, mixed by inverting for 15 s and incubated at RT for 5 min. Meanwhile, phaselock gel heavy tubes were prepared by spinning them at 12,000 × $g$ for 30 s. The supernatant except the membrane pieces were transferred to the prepared phaselock gel heavy tubes, incubated at RT for 5 min at 1200 rpm on a thermomixer and centrifuged at 17,000 × $g$ for 15 min at RT. The aqueous layer was transferred to new 2 ml low-bind tubes. Further, 2 volumes of RNA binding buffer from the Nucleospin plasmid isolation kit and equal volume of 100% ethanol were added and mixed well. The mixed samples were then transferred to Zymo-spin column, centrifuged at 15,000 × $g$ for 30 s at RT and the flow through was discarded. This step was repeated until all the samples were loaded and spun through the column. 400 μl RNA prep buffer was added to the column, centrifuged at 15,000 × $g$ for 30 s at RT and the flow through was discarded. Later, 700 μl of RNA wash buffer was added to the column, centrifuged at 15,000 × $g$ for 30 s at RT and the flow through was discarded. This step

was repeated again with 400 ml RNA wash buffer with additional centrifugation for 2 min to get rid of excess RNA wash buffer. Finally, the RNA was eluted on fresh low-bind tube with nuclease free water by centrifuging at 15,000 × $g$ for 1 min at RT. The RNA was stored at −80 °C until reverse transcription.

**Reverse transcription.** dNTPs and RT oligo mix (2 μl of RT oligo (1 μM), 1 μl dNTPs (10 mM each), and nuclease free water in a total volume of 5 μl per sample) were prepared and added to each RNA extracted from the membrane. The samples were mixes, briefly centrifuged and incubated in a thermomixer for 5 min at 65 °C and then on ice for at least 1 min. 5x superscript IV buffer (SSIV) was vortexed, briefly centrifuge and the reverse transcription (RT) reaction mix was prepared (4 μl of 5x SSIV buffer, 1 μl 0.1 M DTT, 1 μl RNase OUT, 1 μl superscript IV in a total volume of 7 μl per sample). 7 μl of the RT reaction mix was added to each tube containing the RNA and incubated in a thermomixer at 25 °C for 5 min, 42 °C for 20 min, 50 °C for 10 min, 80 °C for 5 min and hold at 4 °C. Later, 1 μl RNase H was added to each tube and incubated at 37 °C for 20 min to cleave the RNA in the DNA-RNA hybrid.

**Cleanup I.** MyONE beads were used for the cleanup of the cDNA. The MyONE beads were mixed by vortexing and 10 μl beads were used per sample. The beads were washed once with 500 μl RLT buffer, resuspended in 125 μl RLT buffer and added to each sample and finally transferred to new 1.5 ml low-bind tube and mixed well by pipetting. 150 μl of 100% ice cold ethanol was added to the cDNA-beads complex, mixed well and incubated at RT for 5 min. The samples were further incubated for 5 min at RT after mixing them once again by pipetting up and down. Next, the samples were placed on a magnetic stand, the supernatant was discarded, and the beads were re-suspended in 900 μl of freshly prepared 80% ice cold ethanol and mixed by pipetting. The mix was transferred to a new low-bind tube, the supernatant was discarded and the above step was repeated twice. The tubes were spun briefly to remove as much ethanol as possible, air dried at RT. Finally, the beads were re-suspended in 5 μl nuclease free water.

**Adapter ligation.** 2 μl adapters L02clip2.0 (IgG), L02clip2.0 (KIFC1 replicate 1), L05clip2.0 (KIFC1 replicate 2), L10clip2.0 (KIFC1 replicate 3), L19clip2.0 (KIFC1 replicate 4), and L02clip2.0 (nuclease free water) from 10 μm stock were used from the publication Buchbender et al.[66]. 1 μl of 100% DMSO was added to each tube, mixed well and heated on a thermomixer for 2 min at 75 °C and the samples were immediately kept on ice for less than 1 min. The ligase mix (2 μl 10x NEB RNA ligase buffer with 10 mM DTT, 0.2 μl 100 mM ATP, 9 μl 50% PEG 8000, 0.5 μl high conc. RNA ligase in a total volume of 12 μl per sample) were prepared and added to the tubes containing the beads and the adapters samples and mixed well. Additionally, 1 μl of high conc. RNA ligase was added to each sample, mixed well and the samples were incubated overnight at RT (20 °C) at 1100 rpm on a thermomixer.

**Cleanup II.** MyONE beads were used for the second cleanup procedure and steps were followed as mentioned in the cleanup I section (see above). At the end, the beads were resuspended in 23 μl nuclease free water, incubated at RT for 5 min, and the supernatant without the beads were transferred to new PCR tubes.

**cDNA pre-amplification.** Phusion master mix (2.5 μl P3Solexa_s and P5Solexa_s mix, 10 μM each, 25 μl 2x Phusion HF PCR master mix to a total volume of 27.5 μl per sample) was added to 22.5 μl cDNA and PCR amplification (98 °C for 30 s, 98 °C for 10 s, 65 °C for 30 s, 72 °C for 15 s, 72 °C for 3 min) was performed for 6 cycles. The amplified cDNA was then size selected using the ProNex beads to reduce the primer-dimers formed during the PCR reaction (see below).

**ProNex size selection I.** To discard fragments less than 55 nt and to retain fragments longer than 75 nt, size selection using ProNex beads was performed. Ultra-low range (ULR) ladder was used for reference (1 µl ULR ladder, 49 µl water) and for size selection (1 µl ULR ladder, 25 µl 2x Phusion HF PCR master mix in a total volume of 50 µl). First, the beads were equilibrated to RT for 30 min on a rotator. 145 µl ProNex beads were added per sample (beads to sample ratio: 1:2.9), mixed well by pipetting and incubated at RT for 10 min. The supernatant was discarded, 300 µl ProNex buffer was added to the beads, incubated for 30–60 s and the supernatant was discarded. This step was repeated once more, the beads were air-dried and the beads were resuspended in 23 µl nuclease free water and the ULR ladder for size selection was resuspended in 50 µl nuclease free water and incubated at RT for 5 min. The samples were placed on a magnetic stand and the eluted cDNA was carefully transferred to a fresh PCR tube. The size was checked using ULR reference ladder and ULR ladder for size selection using sensitivity D1000 tape station kit.

**PCR cycle optimization.** 1 µl size selected cDNA was used for PCR cycle optimization. Phusion master mix (0.5 µl P3Solexa_s and P5Solexa_s mix, 10 µM each, 5 µl 2x Phusion HF PCR master mix to a total volume of 9 µl per sample) were added to 1 µl cDNA and PCR amplification (98 °C for 30 s, 98 °C for 10 s, 65 °C for 30 s, 72 °C for 30 s, 72 °C for 3 min) was performed for 7 and 10 cycles. Depending on the amount of cDNA obtained and to limit the amplification within 10 cycles to minimize the PCR duplicates, for this library we decided to continue with PCR cycle 8 and 9 for preparative PCR.

**Preparative PCR.** Preparative PCR was performed with 10 µl cDNA for 8 and 9 cycles. Phusion master mix (2 µl P3Solexa_s and P5Solexa_s mix, 10 µM each, 20 µl 2x Phusion HF PCR master mix to a total volume of 30 µl per sample) was added to 10 µl cDNA and amplification was performed. 2 µl of the amplified library was used for run with a High Sensitivity D1000 tape station kit. The library from 8 and 9 cycles were combined and all four samples (KIFC1 replicates 1–4) were multiplexed. Further, a second size selection using ProNex beads was preformed to remove residual primer-dimers from the PCR amplification.

**ProNex size selection II.** ProNex selection was performed as described in the above section ProNex size selection I with samples to beads ratio of 1:2.2. Finally, the library samples were eluted in 63 µl nuclease free water and the concentration was measured using a Qubit device. The samples were sequenced using Illumina Inc., NextSeq 550 high output v2.5, 150 cycles, 320 million reads platform at the DKFZ high-throughput core facility.

**Sequence analysis.** The 150 nt long reads were mapped to hg38 human genome (GENCODE v39) using STAR 2.5.3a[120] and further analysis was performed using uniquely mapped reads, after evaluation of the amount of rRNA sequences as previously described[69,71]. Binding sites were defined with BindingSiteFinder v2.0.0 as described in Busch et al.[67]. The binding site width was set to 7 nt. 36934 binding sites were in common for at least 3 out of 4 replicates. Each binding site was assigned the maximum of the PureCLIP score from the replicates and the reads were overlapped with gene annotations from GENCODE on human genome (GENCODE v43). The binding sites with the 5% lowest and highest scores were removed and after assigning the binding sites to distinct genes and gene regions, the target spectrum of KIFC1 was generated (33240 binding sites remaining). The pentamer frequencies were evaluated in the top 20% binding site as compared to the bottom 20% binding sites.

## Cloning

For primer sequences and catalog numbers, please refer to Supplementary Data 5.

A cDNA clone containing the full length KIFC1 open reading frame (ORF) was purchased from Sino biologicals. The ORF of KIFC1 WT was PCR-amplified using primers containing sequences for gateway recombination technology (Thermo Fischer Scientific, Germany). The resulting PCR product was recombined into pDONOR221 cloning vector. The ORF was amplified by PCR (25 µl 2x Phusion high-fidelity PCR master mix, 1 µl DNA (10 ng in total), 2.5 µl forward and reverse primer (10 µM), 1.5 µl DMSO in a total volume 50 µl per reaction) amplification at 98 °C for 1 min, 98 °C for 10 s, 55 °C for 30 s, 72 °C for 1 min, 72 °C for 10 min and hold at 4 °C) for 40 cycles. The amplicon was run on 0.8% agarose gel, the band was cut out, purified using the GeneJET gel extraction kit according to the manufacturer's instructions and finally eluted in 50 µl nuclease free water. For the recombination into pDONOR221, BP reaction (1 µl PCR product (-50 ng/µl), 1 µl pDONOR221 (150 ng/µl), 6 µl TE buffer pH 8, 2 µl 5x BP clonase enzyme mix in a total volume of 10 µl) was carried out at 25 °C overnight in a PCR machine. Finally, 1 µl proteinase K was added, incubated at 37 °C for 10 min in a PCR machine to terminate the reaction.

Transformation: One shot top 10 chemically competent E. coli cells were transformed with 1.5 µl BP reaction mix by incubating on ice for 30 min, heat shock at 42 °C for 2 min, incubated on ice again for 2 min. Further, 250 µl LB medium was added to the cells and incubated for 1 h at 37 °C with mild shaking. Later, 100 µl cells were plated out on LB agarose plates containing 10 µg/ml kanamycin and incubated overnight at 37 °C for the bacteria to grow.

The following day, 2 colonies were picked per plate, plasmid isolation was performed using the Nucleospin plasmid isolation kit based on manufacturers protocol and were sequenced prior to use.

The sequence-validated KIFC1 WT ORF in the universal vector pDONOR221 was shuttled into the transient expression vector pFRT-Flag-HA-ΔCmR-ΔccdB (source) by gateway LR recombination (Thermo Fischer Scientific, Darmstadt Germany) according to manufacturer instructions. Transformation, isolation and validation were carried out as described above, except that the selection of colonies after transformation was done with 100 µg/ml ampicillin instead of kanamycin.

Generation of mutation constructs: Serine and threonine residues were mutated to alanine- an amino acid residue which cannot be phosphorylated (KIFC1-S6A, KIFC1-S26A, KIFC1-S31A, KIFC1-S96A, KIFC1-T187A, KIFC1-S221A, KIFC1-S349A, KIFC1-T359A, KIFC1-S349A/T359A). The mutations were introduced by the use of the Q5 Site-Directed Mutagenesis Kit from NEB. Primers were ordered to perform single or double base substitutions using PCR amplification. The mutations were performed on the KIFC1 WT sequence on pFRT-Flag-HA-ΔCmR-ΔccdB plasmid backbone. Site-Directed mutagenesis reaction: 1 µg KIFC1-WT DNA template, 12.5 µl Q5 Hot Start High-Fidelity 2x Master Mix, 1.25 µl 10 µM forward and reverse primer each, in a total volume of 25 µl per sample) at (Initial denaturation 98 °C for 30 seconds, denaturation at 98 °C for 10 seconds, annealing at 61–72 °C for 30 seconds depending on the individual primers, extension at 72 °C for 3 min, final extension at 72 °C for 2 min and hold at 4 °C for 25 cycles). 10% of the PCR amplified product was run on 0.8% agarose gel to check the amplicon size and integrity.

Transformation: Kinase, ligase and DpnI (KLD) treatment (PCR product 1 µl, 5 µl 2x KLD reaction buffer, 1 µl 10x KLD enzyme were added to a total volume of 10 µl per reaction) was performed prior to transformation for 5 min at RT. Following the treatment, 5 µl of the KLD reaction mixture was directly added to 50 µl NEB® 5-alpha Competent E. coli (High Efficiency) cells, incubated for 30 min on ice, heat shock at 42 °C for 30 seconds, followed by incubation on ice for 5 min. Further, 950 µl SOC media was added to the cells and incubated at 37 °C for 1 h with gentle shaking. Later, 50–100 µl of the cells were plated on LB agar plate with appropriate selection antibiotics and incubated overnight at 37 °C for the bacteria to grow.

The following day, 2 colonies were picked per plate/mutant, plasmid isolation was performed using the Nucleospin plasmid

isolation kit (Macherey Nagel) based on manufacturers protocol and sequences were verified to use.

## In vitro kinase assay

**Transfection.** 5 million HeLa cells were plated out on a 15 cm dish on day 1. On day 2, they were transfected with plasmids (pFRT-Flag-HA-ΔCmR-ΔccdB as empty vector control, FlagHA-KIFC1_WT, FlagHA-KIFC1_S6A, FlagHA-KIFC1_S26A, FlagHA-KIFC1_S31A, FlagHA-KIFC1_S96A, FlagHA-KIFC1_T187A, FlagHA-KIFC1_S221A, FlagHA-KIFC1_S349A, FlagHA-KIFC1_T359A) using lipofectamine 2000 at a DNA:lipofectamine ratio of 1:2.5, based on the manufacturer's instructions. After 6 h, the plates were washed once with warm PBS and fresh media was added to the cells. 24 h post transfection, the cells were harvested, the pellets were flash frozen and stored at −80 °C until further use.

**Cell lysis.** Each cell pellet was lysed in 1 ml lysis buffer (50 mM Tris-HCl pH 7.4, 150 mM NaCl, 1 mM EDTA, 1 mM NaF, 1% triton X-100, 1x complete EDTA free protease inhibitor cocktail), incubated on ice for 30 min and centrifuged at 17,000 g for 20 min at 4 °C. The supernatant was collected into a fresh 2 ml low-bind tube and filtered through proteus clarification mini spin column by centrifuging at $16,000 \times g$ for 2 min at 4 °C. BCA assay was performed to determine the protein concentration and the lysates were diluted to 1.5 mg/ml using lysis buffer.

**Beads preparation and immunoprecipitation.** Anti-DYKDDDDK magnetic agarose beads were mixed by gentle vortexing and 37.5 μl beads were used per sample containing 1.5 mg lysate. The beads were first washed three times with 1 ml lysis buffer. After the last wash, the beads were transferred to a fresh 1.5 ml tube, collected on a magnetic stand, the supernatant was discarded and lysate was added to the beads. The lysate-beads mix were incubated for 2 h at 4 °C on a rotator. Further, the samples were placed on a magnetic stand, the supernatant was removed and the beads were washed 4 times with 1 ml wash buffer (50 mM Tris-HCl pH 7.4, 350 mM NaCl, 1% triton X-100, 1x complete EDTA free protease inhibitor cocktail) for 5 min at 4 °C on a rotating wheel. During the last wash, the beads were transferred to a fresh 1.5 ml tube, the supernatant was discarded and the beads were resuspended in 1 ml PNK buffer (20 mM Tris-HCl pH 7.4, 10 mM MgCl$_2$, 0.2% tween-20, 1x complete EDTA free protease inhibitor cocktail) and stored at 4 °C overnight. The following day, 100 μl of the samples were used for Western blot analysis to check for IP efficiency.

**In vitro kinase assay for single mutants.** 1x kinase assay buffer was prepared from 10x kinase assay buffer (500 mM Tris- HCl pH 7.5, 100 mM NaCl, 100 mM MgCl$_2$, 10 mM DTT, 1x complete EDTA free protease inhibitor cocktail). The insect cell purified AURKA protein was diluted to 0.5 mg/ml in 1x kinase assay buffer and the kinase assay master mix was prepared (negative control without purified AURKA protein: 2 μl 1x kinase assay buffer, 2 μl 10x kinase assay buffer, 2 μl 1 mM cold ATP and 0.5 μl $^{32}$P-γ-ATP; kinase assay with purified AURKA protein: 2 μl purified AURKA protein 0.5 mg/ml, 2 μl 10x kinase assay buffer, 2 μl 1 mM cold ATP and 0.5 μl $^{32}$P-γ-ATP in a total volume of 6.5 μl per sample). Further, supernatant was removed from the beads, 6.5 μl of the master mix was added to the samples and incubated for 30 min at 30 °C in a thermomixer at 800 rpm. After the incubation, the samples were eluted by adding 5 μl 4x LDS supplemented with 200 mM DTT, boiled at 70 °C for 10 min.

To visualize the phosphorylation signal, SDS-PAGE analysis was performed. The samples were loaded in a 7.5% Mini-PROTEAN® TGX™ precast protein gel and run at 120 V in a vertical electrophoresis chamber filled with 1× SDS running buffer (25 mM Tris base, 192 mM glycine, 0.1% (w/v) SDS). Later, the gel was fixed with 15 ml fixation solution (50% methanol, 10% acetic acid) for 1 h at RT with slow rocking, washed three times with nuclease free water for 5 min and dried

for 1–1.5 h at 80 °C. The dried gel was then exposed to phosphor imager screen and was scanned after appropriate amount of time using Typhoon laser scanner phosphor imager at 200 μm, high speed and intensity 3.

**In-vitro kinase assay for double mutant.** 5 million HeLa cells were seeded on a 15 cm dish. On the next day, the cells were transfected with plasmids (pFRT-Flag-HA-ΔCmR-ΔccdB (empty vector control), FlagHA-KIFC1_WT, FlagHA-KIFC1_S349A, FlagHA-KIFC1_T359A, FlagHA-KIFC1_S349A/T359A) using lipofectamine 2000 at DNA:lipofectamine ratio of 1:2.5, based on the manufacturer's instructions. After 6 h, the plates were washed once with warm PBS and fresh media was added to the cells. 24 h post transfection, the cells were harvested, the pellets were flash frozen and stored at −80 °C until further use.

The kinase assay was performed just as described above in this section. During the kinase reaction, insect cell purified AURKA kinase dead mutant D274A was used as a negative control and AURKA WT protein was used for the kinase activity to demonstrate the phosphorylation of KIFC1 WT vs the phosphorylation mutants. Further, after fixing the gel, the gel was first stained with Coomassie stain (to visualize protein bands) overnight at RT on a shaker, washed with nuclease free water for 1 h at RT, imaged and then dried and exposed to phosphor imager screen.

**In vitro kinase assay +/−total RNA.** AURKA WT, AURKA D274A (kinase dead mutant), TPX2 WT and KIFC1 WT proteins purified from insect cells were used for the assay. Total RNA extracted from the HeLa cells synchronized in prometaphase was used to study the RNA dependent phosphorylation of TPX2 and KIFC1 by AURKA WT protein.

First, the protein dilutions and the master mixes were prepared. 1x kinase assay buffer was prepared from 10x kinase assay buffer (500 mM Tris-HCl pH 7.5, 100 mM NaCl, 100 mM MgCl$_2$, 10 mM DTT, 1x complete EDTA free protease inhibitor cocktail). The insect cell purified proteins (AURKA WT, AURKA D274A, KIFC1 WT and TPX2 WT) was diluted to 0.5 mg/ml in 1x kinase assay buffer. The total RNA was diluted to a concentration of 1 μg/μl in 1x kinase assay buffer. Later, the kinase assay master mixes were prepared (master mix without purified AURKA protein: 1 μl 10x kinase assay buffer, 2 μl 1 mM cold ATP and 0.5 μl $^{32}$P-γ-ATP; kinase assay with purified AURKA protein (WT or D274A): 2 μl purified AURKA protein 0.5 mg/ml, 1 μl 10x kinase assay buffer, 2 μl 1 mM cold ATP and 0.5 μl $^{32}$P-γ-ATP).

After preparing the master mixes, the proteins and RNA were pipetted into appropriate tubes. The total volume of each sample was adjusted to 10 μl with 1x kinase assay buffer. For control, substrate proteins alone (2 μl) or substrate proteins with total RNA were used (2 μl proteins+ 1 μl RNA). Further, 2 μl substrate proteins, without RNA and 2 μl substrate proteins, with 1 μl RNA were pipetted into appropriate tubes to compare the phosphorylation levels in the presence or absence of RNA. Here, as a last sample (2 μl substrate proteins, 1 μl RNA), AURKA D274A was used a control for phosphorylation mediated by active AURKA. Further, 3.5 μl of the master mix without AURKA was added to control samples (substrate only or substrate with RNA), 5.5 μl of master mix containing AURKA WT or AURKA D274A was pipetted into appropriate samples and incubated for 30 min at 30 °C in a thermomixer at 800 rpm. After the incubation, the samples were eluted by adding 2.5 μl 4x LDS supplemented with 200 mM DTT, boiled at 70 °C for 10 min.

To visualize the phosphorylation signal, SDS-PAGE analysis was performed. The samples were loaded in a 7.5% Mini-PROTEAN® TGX™ precast protein gel and run at 120 V in a vertical electrophoresis chamber filled with 1x SDS running buffer (25 mM Tris base, 192 mM glycine, 0.1% (w/v) SDS). Later, the gel was fixed with 15 ml fixation solution (50% methanol, 10% acetic acid) for 1 h at RT with slow rocking, washed three times with nuclease free water for 5 min and dried

for 1–1.5 h at 80 °C. The dried gel was then exposed to phosphor imager screen and was scanned after appropriate amount of time using Typhoon laser scanner phosphor imager at 200 μm, high speed and intensity 3.

For the autophosphorylation of AURKA: All the proteins and RNAs were diluted as mentioned above. AURKA WT protein and AURKA D274A were incubated with or without RNA for the kinase assay. Control: 2 μl AURKA D274A or AURKA WT + 3.5 μl master mix without AURKA. For checking autophosphorylation of AURKA in the absence or presence of RNA, (2 μl AURKA WT + 3.5 μl master mix without AURKA/ 2 μl AURKA WT + 1 μl RNA + 3.5 μl master mix without AURKA). For phosphorylation control, (2 μl AURKA D274A + 1 μl RNA + 3.5 μl master mix without AURKA) were pipetted into appropriate tubes. The kinase assay and the following steps were performed as mentioned above.

## RNA extraction
HeLa cells synchronized in prometaphase was used for RNA extraction. The cells were lysed with 1 ml trizol (Tri Reagent, cat. no. T9424, Sigma) and mixed by pipetting up and down. 200 μl chloroform was added to the samples, shaken vigorously for at least 30 s and incubated for 2–3 min at RT. Meanwhile, Phaselock gel heavy tubes were prepared by spinning the tubes for 30 s at 12,000 × g to settle the gel to the bottom of the tube. After the incubation, the samples were transferred to the prepared Phaselock gel heavy tubes, incubated for 5 min on a shaker at 1200 rpm. Later, the samples were centrifuged at 17,000 × g for 15 min at RT and the upper aqueous layer was transferred to a new RNase/ DNase free tube. Further, 1 volume of ethanol (95–100%) was added to 1 volume of the aqueous phase (1:1), mixed well and transferred to zymo-spin IC column. Further, the RNA was extracted based on the manufacturers protocol. The RNA concentration was measured using nanodrop and stored at −80 °C until further use.

## NanoDSF analysis
10 ul of AURKA (0.5 mg/ml in 10 mM HEPES pH 7.5, 500 mM KCl, 2 mM MgCl$_2$, 1 mM CaCl$_2$, 4 mM EGTA and 10% Glycerol) or AURKA (0.5 mg/ ml) mixed with RNA (0.5 mg/ml), in low salt buffer (10 mM HEPES pH 7.5 and 10 mM NaCl), PBS or high salt buffer (10 mM HEPES pH 7.5 and 750 mM NaCl), was loaded into standard grade Promotheus NT.48 capillary (NanoTemper Technologies GmbH). Intrinsic protein fluorescence was measured continuously at 330 nm and 350 nm with a temperature gradient ranging from 20 °C to 90 °C at a rate of 1.0 °C/ min. 85% excitation power was used. The Prometheus NT.48 nanoDSF system was used (NanoTemper Technologies GmbH). The $T_{onset}$ (temperature at which unfolding begins) and the $T_m$ (melting temperature, i.e., inflection point for the ratio 350 nm/330 nm) were calculated with the PR. ThermControl software (NanoTemper Technologies).

## Reporting summary
Further information on research design is available in the Nature Portfolio Reporting Summary linked to this article.

## Data availability
The accession number for the R-DeeP screen proteomic dataset is PXD056068 at ProteomeXchange. The accession number for the AURKA interactor analysis proteomic dataset is PXD056233 at ProteomeXchange. The accession number for the KIFC1 iCLIP datasets (4 replicates) is E-MTAB-14472 at ArrayExpress. The accession number for the RNA-seq datasets (total RNA, 3 replicates) is E-MTAB-14754 at ArrayExpress. Source Data and raw images are available at Zenodo [https://doi.org/10.5281/zenodo.14728287].

## Material availability
Requests for biological materials should be addressed to the corresponding authors per email.

## Code availability
The R-DeeP 3.0 database is available online at R-DeeP3 [https://R-DeeP3.dkfz.de]. The R-DeeP analysis code is available from the documentation section of the R-DeeP database [https://R-DeeP.dkfz.de]. Code for the bioinformatic analysis is available from a GitHub repository [https://github.com/maiwen-ch/Atlas_R-DeeP_Mitosis][121].

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

## Acknowledgements

The authors sincerely acknowledge the IT core facility of the German Cancer Research Center (DKFZ) for their precious support and help in deploying the R-DeeP 3.0 database. This study would not have been realized without the support of the microscopy core facility of the DKFZ (Felix Bestvater, Manuela Brom and Damir Krunic) and the support of all lab members especially Malte Hermes, Katharina Arnold and Elsa Wassmer in the initial stages of the study and our colleague Heike Wilhelm. We sincerely thank Anna Orekhova for her support with the iCLIP protocol, and Mirko Brüggemann for his support and advice with the analysis of the iCLIP datasets. We also thank the high-throughput sequencing core facility of the DKFZ, the Protein Expression and Purification Core Facility at the EMBL (Kim Remans and Karine Lapouge), and our collaborators for the feedback on the database, experimental design, figures and manuscript. Research on RNA–protein complexes in our labs is supported by the German Cancer Aid [Deutsche Krebshilfe 70113919] to S.D., NIH [R35GM119455] and NCI [P30CA023108 DCC Core grant] funding to A.N.K., and financed by the Baden-Württemberg Stiftung [BWST-ISF2019-027 to M.C.-H.]. MK position is supported by a grant from the Deutsche Forschungsgemeinschaft [DFG ZA 881/6-1 (project 541627224)]. Funding for open access charge: DKFZ Core Funding.

## Author contributions

M.C.-H. conceived the study. M.C.-H. and S.D. supervised the R-DeeP screens, the statistical analysis, and the analyses of the RNA dependence of the proteins. A.N.K. supervised the mass spectrometry analyses. J.K. and K.Z. supported the implementation of the iCLIP protocol and downstream bioinformatic analysis. M.S. and D.H. performed mass spectrometry for AURKA immunoprecipitation and provided strong support for the downstream analysis. R.W. provided strong support and expertise in plasmid design and tools for cell biology assays. V.R., J.S., I.N., S.C., J.T., F.H. and M.K. performed the experiments and bioinformatic analyses. B.K. integrated the datasets into the R-DeeP 3.0 database. V.R. wrote the initial manuscript. M.C.-H. and S.D. edited the manuscript including the contributions of the co-authors. All the authors have seen and approved the final version of the manuscript.

## Funding

## Competing interests

The authors declare the following competing interests: S.D. is co-owner of siTOOLs Biotech, Martinsried, Germany, without relation to this work. This study is part of the PhD thesis of V.R. The other authors declare no competing interests.
