## [Transparent Peer Review file · Nature Communications]

An atlas of RNA-dependent proteins in cell division reveals the riboregulation of mitotic protein-protein interactions

Corresponding Author: Dr Maiwen Caudron-Herger

Version 0:

Reviewer comments:

Reviewer #1

(Remarks to the Author)

Summary

Control of RNA metabolism has emerged as an important aspect of nearly all cellular functions. Recent work using a wide-variety of complementary approaches demonstrated that a huge number of proteins interact with RNA and that many RNA-interacting factors lack conventional RNA-binding domains. As a result, it has become important for the RNA biology community to understand how nonconventional RBPs interact with RNA and to determine if these interactions control protein function and/or RNA metabolism. Additionally, work from several groups has suggested that interaction of critical mitotic regulators with RNAs and conventional RBPs is important for several aspects of mitotic progression. Most previous purification strategies that used asynchronous cultured cells, which are predominately in interphase. As a result, the field only has a single catalog of mitotic RBPs and has little understanding of the mechanisms through which RNA could regulate mitosis. The authors of this manuscript previously developed a novel strategy, R-DeeP, that uses cellular fractionation coupled with RNase treatment and mass spectrometry to identify many novel RBPs. In this work they apply the R-DeeP strategy to cells synchronized in mitosis. The positive aspects of this manuscript are: 1. a novel catalog of mitotic RBPs, including some important mitotic regulators, 2. Good quality data and clear data presentation, 3. Interesting hypothesis for regulation of Aurora-A, TPX2, and KIF1C by RNA binding. The negative aspects of this manuscript are: 1. It does not move forward much from the original R-DeeP screen and the authors do not emphasize the novel discoveries in this manuscript, 2. The authors do not provide any insight into why or how some factors interact with RNA or if RNA is important for their regulation, 3. The authors identify a novel interaction between Kif1C and Aurora-A as well as a novel phosphorylation of Kif1c, however they do not explore the functional relevance of either, 4. There are many internal contradictions of the authors results and contradictions with the published literature that are not mentioned or discussed. In order for this manuscript to be of interest to a wide audience the authors need to provide a deeper insight into unconventional RBPs or insight into the role of RNA-RBP interactions during mitosis. In its current form this manuscript is a nice resource for the community of people interested in RBPs, but provides no biological insight.

Major Points:

1. The current manuscript does not extend the authors previous work (31076284) in a meaningful way. The previous work used the same methodology to identify a wide-range of new RBPs, including many mitotic regulators. The current work has extended this methodology to cells synchronized in mitosis, but it is not clear if they have discovered anything new. In Figure 1 and accompanying text the authors state that they have found 500 new RBPs that are only identified in mitosis, but they present no analysis of this potentially interesting set of proteins. Are there proteins that show a cell-cycle regulated interaction with RNA? If so, what are their functions? It would also be interesting to compare the shifting coefficient for each protein between interphase and mitosis. This could highlight cell-cycle regulated RNA-binding. The authors have not made a compelling case for why these new experiments are important and not just an incremental extension of their previous work. Figures 1-3 of this manuscript are essentially a direct repeat of the author's previous work. Additionally, a previous study (31123356) used organic phase separation to identify proteins that directly crosslink to RNA during mitosis. The authors should cite this work, compare their results to the previously published work, and discuss the similarities and differences between the two studies.
2. Figure 2 is not an effective way to present this data. The gene names will be unreadable once reduced to a journal size and most of the protein names indicated in the figure do not shift in their experiments, so they are irrelevant. The authors should find a way to present this data more concisely.
3. Figure 3 is a validation of a single protein from their experiment. Since the remainder of the manuscript focuses on Aurora-

A, TPX2 and Kif1C they should validate all of these factors by western blot in independent samples and include the data in the main Figure.

4. The authors should include the PLA quantitation +/- RNase treatment in the main Figure. This is an important orthogonal validation of their R-DeeP results and showing quantitative differences between +/- RNase samples is a critical confirmation of their R-deeP screen results.

5. In Figure 5 the authors report results of CLIP with TPX2, Kif1C and Aurora-A. This is a potentially novel analysis and needs to be presented in more depth. I have a few concerns about this data and presentation:

a. Each IP and western needs a no UV-crosslink control to ensure that the upshift of each band is indeed caused by crosslinking to RNA. The increasing RNase concentrations are suggestive of direct RNA binding, but a no crosslink control is critical.

b. In the analysis of the CLIP data the authors argue that there is no sequence specificity for RNA binding to Kif1C. However, the authors have presented a relatively cursory analysis and should include additional data. For example, it would be useful to present the number of reads mapped to each gene in the CLIP sample compared to the abundance of each transcript in a total extract. This would help determine if Kif1C is only binding to the most abundant transcripts in the cell or to a specific subset of transcripts. Second, the authors have used a somewhat unconventional method for calling 'binding sites' in their CLIP data. I think they should reprocess their data using the ENCODE project CLIP standards

(https://www.encodeproject.org/documents/3b1b2762-269a-4978-902e-0e1f91615782/@@download/attachment/eCLIP_analysisSOP_v2.0.pdf) to confirm that Kif1c does not have preferred binding sites. Finally, Figure 5D presents binding site proportions for each portion of a mRNA. These proportions should be

normalized to sequence length because they have vastly different lengths.

c. The authors should discuss recent data demonstrating that CLIP can be an extremely noisy assay and recover indirect RNA:protein interactions (38387462).

6. In Figure 6 the authors show that Aurora-A can phosphorylate Kif1C in vitro, but do not test the significance of these phosphorylation sites in cells. This could be accomplished through a Kif1c knockdown and complementation with a nonphosphorylatable rescue construct. The Aurora-A and Aurora-B consensus sequences are nearly identical. The authors should test if Aurora-B:Incenp can phosphorylate these residues. This could also be explored in cells using inhibitors specific to Aurora-A and Aurora-B. Finally, this data appears to have no relation to RNA binding by either of these proteins. Does RNA enhance or suppress this phosphorylation reaction? This data needs to be extended to be of general interest.

7. Throughout the manuscript the authors refer to proteins as being 'RNA-dependent', but this term is not well defined. I assume that the authors mean that the migration of the protein is RNA dependent. However, this term is used loosely and in several places the authors imply that the function of a protein is RNA-dependent but no data is provided to support this speculation.

8. There are several inconsistencies in this work that need to be addressed. First, In Figure 2 the authors report that INCENP migration does not change upon RNase treatment. However, in Figure 4B INCENP is reported as a RNase-sensitive interactor of Aurora-A. Second, INCENP is a stoichiometric cofactor with Aurora-B and is not known to bind to Aurora-A. This needs to be discussed. Third, Aurora-A and TPX2 form a tight, stoichiometric complex that has been crystalized (14580337, 12699628). RNA is not part of the mechanism reported for TPX2 binding or activation of Aurora-A. The authors should discuss their model for how RNA is involved in this complex formation and why their model differs from the published literature. Fourth, the CPC has been reported to be an RNA-binding complex in several different organisms but is not identified in the R-DeeP screen. The authors should discuss why their results differ from the published literature.

Minor Points

1. In Figure 1E it is distracting and redundant to have each gene name followed by '_HUMAN'

2. On line 61 the authors identify organic phase separation as a method to identify RNA binding proteins. This line should include the following references (30607034, 30824702)

Reviewer #2

(Remarks to the Author)

The manuscript by Rajagopal et al used the R-DEEP method to investigate RNA-dependent proteins in mitosis, and further characterize the RNA-dependent interaction of three proteins, AURKA, TPX2 and KIFC1.

Overall, the manuscript is well presented and scientifically sound, although some sections could be extended to describe the results in more details.

Comments:

1: how did the authors characterize the effectiveness of the synchronization procedure to produce mitotic and interphase samples? Was this checked and confirmed (for example by FACS)?

2: In the section "An atlas of RNA-dependent proteins in cell division highlights well-known mitotic factors as RNA dependent proteins", it is not clear how the 826 RNA-dependant mitotic proteins were identified. Was it only by comparing the mitotic R-DeeP results and the RBP2GO resource, or was it by comparing the mitotic vs interphase R-Deep datasets? In other words, how many of these 826 proteins are found in the 1093 proteins present only in the mitotic R-DeeP dataset, compared to the interphase dataset? Overall, the manuscript will benefit from a better comparison of the mitotic and interphase datasets. Are there proteins that are only identified in the mitotic or the interphase samples? Are there proteins that are RNA independant in one dataset but RNA dependant in the other dataset?

Additionally, it would be interesting to compare the mitotic-specific as well as the interphase-specific R-Deep dataset with previous R-DeeP datasets done (by the same authors) on unsynchronised cells. I would expect a strong overlap between

the interphase and the unsynchronised datasets, but a difference between the mitotic and unsynchronised datasets. How many of the 826 mitotic specific proteins could have been identified using unsynchronised cells (for example, could AURKA, TPX2 and KIFC1 be identified as RBPs in the unsynchronised datasets?). This comparison would surely provide a better justification of the rationale to perform R-DeeP specifically on mitotic arrested cells.

3: In the section "AURKA has both RNase-sensitive and RNase-insensitive interactions with other proteins", the choice of studying further KIFC1 should be better justified. Why choosing this protein specifically out of the 1080 identified? The sentence "We also identified KIFC1 as one of the most frequently detected RNase-sensitive candidates, whose interaction with AURKA had not been characterized before" is not clear. What do the authors mean by most frequently detected?

4: The description of the PLA assay (lanes 265-268) is oversimplified. The author should either not describe it and refer the original article, or describe it better. The sentence "represented by signal in the form of dots resulting from a ligation followed by an amplification reaction" is too simplistic.

5: In the section "AURKA phosphorylates KIFC1 at S349 and T359", did the authors observe any cell cycle or morphological defects in cells transfected with the S349A/T359A mutant, in comparison to the WT protein? If these phosphorylation events are biologically important, these mutations should have consequences on cell division, migration, ...

6: In the discussion section (about the RNA dependant complex between AURKA, TPX2 and KIFC1), would it be possible to match the R-DEEP mitotic RNA-sensitive proteins, especially those eluting at fraction 21, and the AURK4 RNA-dependant interactors. This might give a hint of other proteins that could be in the same complex, and could provide avenues for further work on the characterization of this complex.

Minor comments:

Lane 324: "were rather localized in the lower 20% binding sites (Figure 5D)". This is shown in Figure 5F

Reviewer #3

(Remarks to the Author)

The manuscript investigates the role of RNA-binding proteins (RBPs) and RNA-dependent protein interactions during cell cycle, with a particular focus on mitosis. The authors utilized the R-DeeP strategy, a proteomic approach they had previously developed, to identify RNA-dependent proteins in HeLa cells synchronized at mitosis and interphase. Their analysis yielded a comprehensive list of RNA-dependent proteins. Interestingly, several key mitotic proteins, including AURKA, KIFC1, and TPX2, exhibited RNA-dependent interactions despite lacking canonical RNA-binding domains. Immunoprecipitation, western blotting and UV-crosslinking assays showed that AURKA, KIFC1, and TPX2 associate with RNA and interact with one another in an RNA-dependent manner. Based on PLA data, these interactions appear to occur during mitotic spindle assembly. Additionally, the authors performed iCLIP-seq, revealing that KIFC1 interacts with a diverse set of RNAs, including rRNAs and mRNAs, without apparent sequence specificity. Mutagenesis and in vitro kinase assays further demonstrated that AURKA phosphorylates KIFC1 at residues S349 and T359.

The findings suggest that these proteins may function as unconventional RBPs and that RNA could play a role in mediating protein-protein interactions, particularly during spindle assembly. The R-DeeP3 online database is as a valuable and accessible resource for the research community.

While this study is comprehensive and intriguing, a significant limitation is the lack of data demonstrating the functional significance of RNA-mediated interactions.

Major points:

1. I recommend that the authors explore the role of RNA in AURKA activity by performing in vitro kinase assays with RNA. Does RNA enhance the kinase activity of AURKA on KIFC1? Even if the result might be negative and RNA may act through other mechanisms, this relatively straightforward experiment could provide some insights into the mechanism.

2. In Figure 5A, the evidence for direct RNA binding and the specificity of the antibody should be reinforced. Could the authors knock down AURKA and perform the same experiment to rule out the possibility of antibody cross-reactivity? Testing synchronized cells in interphase could also serve as a specificity control.

Minor points:

1. In Figure 4C, there appear to be two RNA-dependent peaks at fractions 17-18 and fractions 21-22. This should be mentioned in the text.

2. rRNAs account for approximately 45% of the iCLIP reads, but it is unclear whether rRNA reads are enriched in the library. Clarifying whether these data were obtained after removing unmappable reads and indicating the enrichment fold relative to input would improve the clarity of the figures.

Version 1:

Reviewer comments:

Reviewer #1

(Remarks to the Author)

Summary

This is a revision of a previous manuscript that utilized the R-DeeP fractionation and mass spectrometry methodology to identify a catalog of RNA binding proteins during interphase and mitosis. The authors have performed a tremendous amount of work in this revision and addressed all the concerns that I raised. The new data included in Figure 2, S15c-d, and 8d-f are particularly valuable. I am now convinced that this is important contribution to the literature and is suitable for publication. I have two minor points that should be addressed prior to publication.

Minor points

1. Line 309-311. Co-IP assay does not demonstrate a direct interaction between TPX2 and KIF1C. This interaction could be bridged by other proteins or RNA.
2. Line 404-405. "in vitro kinase assay using purified AURKA in the absence or presence of total RNA for both, KIFC1 and TPX2.". It is difficult to understand what was meant by this sentence.

(Remarks on code availability)

Reviewer #2

(Remarks to the Author)

Following the reviewers comments, the manuscript has been significantly improved and I have no further concerns. It is a nice study.

(Remarks on code availability)

Reviewer #3

(Remarks to the Author)

The authors have provided additional experimental data, which fully addressed my earlier concerns. I recommend the publication of this manuscript.

(Remarks on code availability)

An atlas of RNA-dependent proteins in cell division reveals the riboregulation of mitotic protein-protein interactions

Rajagopal et al.

Point-by-point response letter

First, we would like to thank the editor and the reviewers for their efforts in reviewing our work, for their positive and encouraging comments and their constructive suggestions to improve the present study, which is much appreciated. We have now revised our manuscript accordingly and addressed each comment and issues separately as described in the point-by-point response below. We hope that the reviewers will now find the manuscript acceptable for publication in *Nature Communications*.

Reviewer #1 (Remarks to the Author)

Summary

Control of RNA metabolism has emerged as an important aspect of nearly all cellular functions. Recent work using a wide-variety of complementary approaches demonstrated that a huge number of proteins interact with RNA and that many RNA-interacting factors lack conventional RNA-binding domains. As a result, it has become important for the RNA biology community to understand how nonconventional RBPs interact with RNA and to determine if these interactions control protein function and/or RNA metabolism. Additionally, work from several groups has suggested that interaction of critical mitotic regulators with RNAs and conventional RBPs is important for several aspects of mitotic progression. Most previous purification strategies that used asynchronous cultured cells, which are predominately in interphase. As a result, the field only has a single catalog of mitotic RBPs and has little understanding of the mechanisms through which RNA could regulate mitosis. The authors of this manuscript previously developed a novel strategy, R-DeeP, that uses cellular fractionation coupled with RNase treatment and mass spectrometry to identify many novel RBPs. In this work they apply the R-DeeP strategy to cells synchronized in mitosis. The positive aspects of this manuscript are: 1. a novel catalog of mitotic RBPs, including some important mitotic regulators, 2. Good quality data and clear data presentation, 3. Interesting hypothesis for regulation of Aurora-A, TPX2, and KIF1C by RNA binding. The negative aspects of this manuscript are: 1. It does not move forward much from the original R-DeeP screen and the authors do not emphasize the novel discoveries in this manuscript, 2. The authors do not provide any insight into why or how some factors interact with RNA or if RNA is important for their regulation, 3. The authors identify a novel interaction between Kif1C and Aurora-A as well as a novel phosphorylation of Kif1c, however they do not explore the functional relevance of either, 4. There are many internal contradictions of the authors results and contradictions with the published literature that are not mentioned or discussed. In order for this manuscript to be of interest to a wide audience the authors need to provide a deeper insight into unconventional RBPs or insight into the role of RNA-RBP interactions during mitosis. In its current form this manuscript is a nice resource for the community of people interested in RBPs, but provides no biological insight.

Major Points:

1. The current manuscript does not extend the authors previous work (31076284) in a meaningful way. The previous work used the same methodology to identify a wide-range of new RBPs, including many mitotic regulators. The current work has extended this methodology to cells synchronized in mitosis, but it is not clear if they have discovered anything new. In Figure 1 and accompanying text the authors state that they have found 500 new RBPs that are only identified in mitosis, but they present no analysis of this potentially interesting set of proteins. Are there proteins that show a cell-cycle regulated interaction with RNA? If so, what are their functions? It would also be interesting to

compare the shifting coefficient for each protein between interphase and mitosis. This could highlight cell-cycle regulated RNA-binding. The authors have not made a compelling case for why these new experiments are important and not just an incremental extension of their previous work. Figures 1-3 of this manuscript are essentially a direct repeat of the author's previous work. Additionally, a previous study (31123356) used organic phase separation to identify proteins that directly crosslink to RNA during mitosis. The authors should cite this work, compare their results to the previously published work, and discuss the similarities and differences between the two studies.

We thank the reviewer for these important suggestions. We have performed further analyses of the mitotic versus interphasic R-DeeP screens in HeLa cells, as well as comparisons with the initial R-DeeP screen in unsynchronized HeLa cells and the aforementioned study based on organic phase separation (see lines 172-215 and added Figures: **Figure 2** (new) and **Supplementary Fig. 2f**).

First, our additional analysis focusing on the differential shifting behavior of proteins in mitosis as compared to interphase highlighted 586 proteins with specific shifts in mitosis and no shifts in interphase and 726 proteins with specific shifts in interphase and no shifts in mitosis. This new analysis has been integrated into the manuscript as **Figure 2b**. In addition, 65 shifting proteins were unique to the mitotic dataset and 86 proteins were unique to the interphase dataset (**Figure 2c**). In addition, 136 commonly shifting proteins were depicting shifts in opposite directions in mitosis versus interphase, while 948 proteins were commonly left-shifting (a left shift is a shift toward lower sucrose density fractions upon RNase treatment = loss of apparent molecular weight). Usually, most of the shifting proteins are left shifting.

GO analyses of these different protein groups indicated that

- i) 948 common left-shifted proteins were enriched for proteins related to RNA processes (**Supplementary Table 1**, GO_Common_Left_Shift).
- ii) The 651 (586 + 65) mitotic-specific shifting proteins and the 812 (726 + 86) interphasic-specific shifting proteins (**Figure 2c**) depicted a number of common GO terms, which were linked to mitochondrial and ER membrane-related cellular component. Also, the two groups shared the GO term "cytoskeleton" but with the distinction that the mitotic proteins were more related to microtubules and interphasic proteins were more related to actin (**Supplementary Table 1**, GO_Mitotic_Only_Shifts, GO_interphase_Only_Shifts). This could indicate a cell cycle dependence of the interaction of subgroups of proteins with RNA within specific GO categories. Also, as indicated in the Queiroz et al. study (PMID: 31123356), nocodazole treatment during synchronization seemed to affect mitochondrial and metabolic activities, which could be here reflected via differential protein behavior, amount and detection in our screens (lines 179-182).
- iii) In addition to the mitotic proteins, the interphasic protein group was also enriched for proteins related to nuclear components.
- iv) Interestingly, the group of 136 proteins with opposite shifts contained 46 proteins with catalytic functions. Apparently, such proteins could have a cell-cycle dependent relation with RNA, leading to loss (left shift) or gain (right shift) of interaction partners. Similar observations had been reported in the Queiroz et al. study (PMID 31123356), which we now cited in the text (see lines 182, 189, 204).

Second, a comparative analysis of the mitotic R-DeeP as compared to the previous unsynchronized R-DeeP study (**Figure 2d and 2e**) demonstrated the valuable gain of depth of the analysis in mitotic synchronized cells. In total, the new analysis provided additional data for 3052 proteins, for which there were no R-DeeP data available (**Figure 2e**). Together with the interphase R-DeeP dataset, new information for 3096 proteins was provided (**Figure 2d**). In addition, 372 proteins not detected in unsynchronized cells plus 338 proteins previously detected but not shifting in unsynchronized cells

were uncovered to significantly shift in mitosis (**Figure 2e**). Thus, the depth and information content of these R-DeeP screens relevantly exceeds our previous study (lines 192-201).

The 372 newly identified proteins were enriched for proteins related to microtubules and the mitotic spindle (**Supplementary Table 1**, GO_Unique_Left_Mitosis_vs_Unsyn). Such an enrichment for mitotic factors was not mentioned in the Queiroz et al. study (PMID 31123356). Therefore, we believe that our findings are complementary to the findings of the OOPS study. The 338 common proteins with previously no shift in unsynchronized but a shift in mitotic HeLa cells depicted GO terms, which were linked to mitochondrial and ER membrane-related cellular components, which could indicate some cell cycle specific regulation or a response to nocodazole treatment (see above). See also lines 202-211.

Third, an important aspect of the new R-DeeP analysis appeared when analyzing the protein amounts between the mitotic and the interphasic screens (**Supplementary Table 1**). The expression levels of 1859 proteins were significantly increased in mitosis and this subset included 160 left-shifting, i.e. RNA-dependent proteins. Also, 111 out of the 372 shifting proteins detected specifically in the new mitotic screen as compared to the previous unsynchronized screen also depicted high protein levels in mitosis, highlighting the better sensitivity of this screen for mitotic RNA-dependent factors.

Lastly, the analysis of the intersection between the OOPS study (PMID 31123356) and the mitotic R-DeeP screen revealed 112 proteins commonly detected as RBP/RNA-dependent, respectively. However, the mitotic R-DeeP screen provided additional information for 1639 RNA-dependent proteins, of which 286 presented up-regulated protein expression levels in mitosis (**Supplementary Fig. 2f** and **Supplementary Table 1**). Also, our analysis of the mitotic OOPS proteins revealed an enrichment in mitotic-related GO terms, which had not been mentioned in that study.

Altogether, our growing family of R-DeeP screen datasets provides an important resource broadly usable for further tailored analyses for the scientific community as exemplified in our present analysis and results. This is further facilitated by the updated database **R-DeeP3 (R-DeeP3.dkfz.de)** and the related database **RBP2GO**, which allow for a keyword-based exploration of GO terms (**RBP2GO.dkfz.de**). In addition, the Atlas of mitotic factors (**Supplementary Table 2**) strongly indicates that RNA has remained largely underexplored as regulatory molecule during mitotic events. We highlight here a great number of key mitotic factors as RNA-dependent proteins and potentially RNA-binding proteins, which directly interact with RNA. Our results demonstrate that RNA riboregulates the kinase activity of AURKA (**Figure 8**, see comments below, point 7). We anticipate that RNA probably riboregulates many more key mitotic factors, a regulatory level that has been largely overseen so far.

2. Figure 2 is not an effective way to present this data. The gene names will be unreadable once reduced to a journal size and most of the protein names indicated in the figure do not shift in their experiments, so they are irrelevant. The authors should find a way to present this data more concisely.

We have modified this figure, which now appear as **Figure 3** in the new version of the manuscript. For simplicity, we only indicate the number of proteins associated to each part of the mitotic spindle (**Figure 3a**) or associated to GO term related to the mitotic cell cycle (**Figure 3b**). A complete list of the corresponding proteins is provided for re-usage in further research in **Supplementary Table 2**. This is indicated in the figure legend accordingly.

3. Figure 3 is a validation of a single protein from their experiment. Since the remainder of the manuscript focuses on Aurora-A, TPX2 and Kif1C they should validate all of these factors by western blot in independent samples and include the data in the main Figure.

The western blot validation for KIFC1 and TPX2 were included in the former Figure 4c for KIFC1 and in the former Supplementary Figure S3A for TPX2. For more clarity, we have now moved these two important validations into the new main **Figure 5c** and **5d**. The western blot validation of AURKA is shown in **Figure 4b** and **4c**. The western blot validations were performed for all proteins in independent triplicate experiments. The raw images of the triplicates are available in the Source Data files.

4. The authors should include the PLA quantitation +/- RNase treatment in the main Figure. This is an important orthogonal validation of their R-DeeP results and showing quantitative differences between +/- RNase samples is a critical confirmation of their R-deeP screen results.

As suggested, we have included the PLA quantification for the AURKA/KIFC1 interaction into the new **main Figure 6c** and **6e**. The other PLA images and quantifications for TPX2/KIFC1 and TPX2/AURKA can be found in **Supplementary Fig. 7**.

5. In Figure 5 the authors report results of CLIP with TPX2, Kif1C and Aurora-A. This is a potentially novel analysis and needs to be presented in more depth. I have a few concerns about this data and presentation:

a. Each IP and western needs a no UV-crosslink control to ensure that the upshift of each band is indeed caused by crosslinking to RNA. The increasing RNase concentrations are suggestive of direct RNA binding, but a no crosslink control is critical.

The no UV-crosslink controls which had not been included in the previous version of the manuscript are now shown for all proteins (**Figure 7a** and **7b**, **Supplementary Figs. 9 to 14**). The absence of signal in the no UV-crosslink controls demonstrate that the three proteins AURKA, KIFC1 and TPX2 directly bind to RNA.

We have also added the western blots showing the comparable IP efficiency between the samples. All experiments have been performed in three independent replicates. The raw images of the triplicates are available in the Source Data files.

b. In the analysis of the CLIP data the authors argue that there is no sequence specificity for RNA binding to Kif1C. However, the authors have presented a relatively cursory analysis and should include additional data. For example, it would be useful to present the number of reads mapped to each gene in the CLIP sample compared to the abundance of each transcript in a total extract. This would help determine if Kif1C is only binding to the most abundant transcripts in the cell or to a specific subset of transcripts.

Following the reviewer's suggestion, we investigated the relationship of iCLIP crosslinks to transcript abundance. For this, we performed an additional RNA sequencing experiment (mitotic HeLa cells, 3 biological replicates). Indeed, we observed a strong correlation of iCLIP crosslinks per gene with the number of RNA-seq reads per gene (**Supplementary Figure 15c**, also depicted below). We note that a correlation is to be expected since the total iCLIP signal on a transcript is usually dominated by background binding and hence scales with expression. A similar relation was previously also shown in Körtel et al. NAR 2021 (<https://doi.org/10.1093/nar/gkab485>, see #supplementary data Figure S3B of

this reference) and Molitor et al. NAR 2023 (<https://doi.org/10.1093/nar/gkac1237>, see Figure S6D of this reference) and is also discussed in the following review: Chakrabarti et al. Annual Review of Biomedical Data Science 2018 (<https://doi.org/10.1146/annurev-biodatasci-080917-013525>, see “Challenge 3: How to account for variable RNA abundance?” of this reference). However, it is important to distinguish between crosslinks and binding sites, as not all crosslinks will define a binding sites. Therefore, an additional analysis was performed on binding site strength as explained below.

Supplementary Figure 15c shows the raw number of iCLIP crosslinks per gene against the number of reads found for the gene in the RNA-seq experiment (mean of three replicates).

For KIF1C, we repeated the analysis focusing only on the signal within the binding sites. Comparing the apparent KIF1C binding strength to the expression of the transcripts indicated that the transcript expression influences the strength of binding sites, as stronger binding sites resided on more highly expressed transcripts (Supplementary Figure 15d - also depicted below).

Supplementary Figure 15d shows the expression of genes with bottom (bot), middle (mid) or top KIF1C binding sites (stratified by the apparent KIF1C binding strength). Bot= bottom 20% of the binding sites according to their PureCLIP score; top = top 20% of the binding sites according to their PureCLIP score; mid = binding sites between bottom 20% and top 20% of the binding sites according to their PureCLIP score.

Altogether, both analyses support the notion that KIF1C binds to RNAs rather in an unspecific manner in cells with binding frequency largely reflecting transcript abundance. These data are now

included in the new version of the manuscript as **Supplementary Fig. 15c** and **15d**. See also lines 372-379.

Second, the authors have used a somewhat unconventional method for calling 'binding sites' in their CLIP data. I think they should reprocess their data using the ENCODE project CLIP standards (https://www.encodeproject.org/documents/3b1b2762-269a-4978-902e-0e1f91615782/@@download/attachment/eCLIP_analysisSOP_v2.0.pdf) to confirm that Kif1c does not have preferred binding sites.

We would respectfully disagree with this comment. The reviewer refers to the common eCLIP processing protocol as used by ENCODE. However, the ENCODE eCLIP processing protocol cannot be used for iCLIP data. This is because iCLIP, in contrast to eCLIP, does not use size-matched input (SMI) samples (see Lee et al. *Technology Review* 2018, <https://doi.org/10.1016/j.molcel.2018.01.005>) and the peak caller (CLIPper, <https://www.nature.com/articles/nsmb.2699>) used in the ENCODE processing depends on SMI samples for the detection of binding sites. A processing of iCLIP data with the ENCODE processing protocol is therefore not possible.

The processing and peak calling method, which we used here, is specifically developed for iCLIP data. It relies on the peak caller PureCLIP (Krakau et al. *Genome Biology* 2017, <https://doi.org/10.1186/s13059-017-1364-2>) that was designed for both eCLIP and iCLIP data. The binding site definition based on the PureCLIP-called crosslink sites has been published in Busch et al. *Methods* 2020 (<https://doi.org/10.1016/j.ymeth.2019.11.008>) and Lewinski et al. *Nature Protocols* 2024 (<https://doi.org/10.1038/s41596-023-00935-3>) and has been used in several iCLIP studies, for example Alvelos et al. *Life Science Alliance* 2021 (<https://doi.org/10.26508/lsa.202000825>), Molitor et al. *NAR* 2023 (<https://doi.org/10.1093/nar/gkac1237>) and de Oliveira Freitas Machado et al. *NAR* 2023 (<https://doi.org/10.1093/nar/gkac1225>).

Finally, Figure 5D presents binding site proportions for each portion of a mRNA. These proportions should be normalized to sequence length because they have vastly different lengths.

According to the reviewer's suggestion, we have added a panel showing the relative enrichment of binding sites into the different regions, normalized to the summed length of each region (**Figure 7f**), confirming the enrichment for coding regions.

c. The authors should discuss recent data demonstrating that CLIP can be an extremely noisy assay and recover indirect RNA:protein interactions (38387462).

The study by Guo et al. *Mol Cell* 2024 (<https://doi.org/10.1016/j.molcel.2024.01.026>) reports on the use of the newly developed CLAP method (covalent linkage and affinity purification) to analyze the RNA-binding affinity of chromatin factors such as PRC2, CTCF, YY1 etc. The topic is highly debated in the field and has led to the publication of a series of comments on PRC2-RNA interactions (<https://doi.org/10.1016/j.molcel.2024.09.007>, <https://doi.org/10.1016/j.molcel.2024.09.010> and <https://doi.org/10.1016/j.molcel.2024.09.010>). While we are not experts in the PRC2 field, our experience with the analysis of proteome-wide studies to identify RNA-binding proteins in different organisms has shown that orthogonal approaches and complementary methods are crucial since each strategy is linked to intrinsic advantages and limitations. While CLIP might be noisy for specific proteins, it has the advantage of capturing the proteins in their endogenous form. While CLAP might be less noisy for some proteins, it might be impossible to use the technology for some proteins. For example, mitotic factors can be very sensitive to the addition of tag at their N- or C-termini. Their function can be impaired and this would need to be tested, adding a level of complexity to the assay. In addition, the overexpression of mitotic factors can be very challenging for the cells, as it can lead to mitotic aberrations (overexpression of microtubule-associated proteins for example can led to aberrant

microtubule bundles, impairing proper cell division processes), necessitating the knockdown/knockout of the endogenous protein, again, adding a certain level of complexity to the experimental setting. In conclusions, we believe that progresses in the RNA-protein interaction fields will be driven by the use of complementary and orthogonal strategies, applied to analyze protein of interest and specific RNA-protein complexes.

Nonetheless, we have added the aforementioned reference to the Discussion section of the manuscript and mention potential shortcomings of the CLIP approach. See lines 525-527 and 536-537 of the manuscript.

6. In Figure 6 the authors show that Aurora-A can phosphorylate Kif1C in vitro, but do not test the significance of these phosphorylation sites in cells. This could be accomplished through a Kif1c knockdown and complementation with a nonphosphorylatable rescue construct.

We agree with this reviewer that this is an important question. Therefore, we have tried to analyze the functional relevance of KIFC1 phosphorylation in KIFC1 KO HeLa and A549 cell lines (which were in preparation at the time of the original submission of our manuscript), as we used HeLa and A549 in this study. Unfortunately, a phenotype due to KIFC1 knockout was difficult to characterize. This finding is in concordance with the available literature, according to which KIFC1/HSET is a normally non-essential kinesin, except for cells presenting centrosomal amplification (a high percentage of cells with more than two centrosomes) (Kwon et al. *Genes & Development* 2008, <https://doi.org/10.1101/gad.1700908> and Kim et al. *Cell Structure and Function* 2013, <https://doi.org/10.1247/csf.12014>). In such cells, KIFC1 depletion seems to lead to the formation of an increased occurrence of multipolar spindles, where KIFC1 should help clustering the extra-numerous centrosomes into two poles for successful cell division. HeLa and A549 cells do not appear to present remarkable centrosomal amplification, which can explain the lack of characteristic phenotype in the KIFC1 KO versions of these cell lines.

To further test this, we have cultured MDA-MB-231 cells (authenticated, a cell line known to present centrosomal amplification) and tested the overexpression of KIFC1 WT and KIFC1 double mutant under KD of KIFC1 per siPOOLS (targeting the 3'UTR region to only silence the endogenous KIFC1 gene). While the knockdown of KIFC1 was successful (see **Figure R1** below), we could not detect a significant increase in multipolar spindles in the KIFC1 knockdown sample (only about 5% in both samples, siRNA control as compared to siRNA KIFC1 in MDA-MB-231 cells).

Figure R1 top panel shows the successful depletion of KIFC1 upon siRNA treatment in MDA-MB-231 cells. The bottom panel shows representative images of mitotic spindles. In each sample, about 5% of spindles showed multipolar spindles. 40 spindles were qualitatively assessed in each sample).

Accordingly, we could not obtain data on the functionality of the KIFC1 double mutant due to the lack of a conclusive phenotype and assay system. While we agree that investigating the functionality of the phosphorylation of KIFC1 is important, it might be only applicable for specific cell types (e.g. cancer cells with amplified centrosomes, possibly meiotic cells in absence of centrosomes). We have added a short discussion on this point in the manuscript (lines 511-520).

However, in our study, we believe that we managed to present a strong general finding with the effect of RNA on AURKA kinase activity and thermostability (see next point below).

The Aurora-A and Aurora-B consensus sequences are nearly identical. The authors should test if Aurora-B:Incenp can phosphorylate these residues. This could also be explored in cells using inhibitors specific to Aurora-A and Aurora-B. Finally, this data appears to have no relation to RNA binding by either of these proteins. Does RNA enhance or suppress this phosphorylation reaction? This data needs to be extended to be of general interest.

In the study by Kettenbach et al. *Science Signalling* 2011 (<https://doi.org/10.1126/scisignal.2001497>), the mitotic phosphorylation sites of AURKA, AURKB and PLK1 were mapped through the combined use of quantitative phosphoproteomics and the selective targeting of the kinase activities by small molecule inhibitors, such as MLN8054 (selective for AURKA at lower concentrations) and AZD1152 or ZM447439 (Aurora B). This study identified proteins that appeared to be phosphorylated by both AURKA and AURKB, but despite the similarities in the consensus sequences of the two kinases, this study also found proteins that were specifically phosphorylated specifically by AURKA and other proteins that were specifically phosphorylated by AURKB. In particular, they clearly identified KIFC1 T359 as an AURKA substrate, while they did not detect any AURKB phosphorylation site on KIFC1. Therefore, we did not pursue investigating a possible phosphorylation of KIFC1 via AURKB:INCENP in the frame of our study.

To investigate the functional relevance of RNA with regard to AURKA activity, we have used purified proteins (AURKA and a kinase-dead mutant of AURKA as a control) and performed an *in vitro* kinase assay to monitor the phosphorylation of both KIFC1 and TPX2 in absence and presence of RNA molecules (total RNA). All three biological replicates for KIFC1 and TPX2 showed an increased phosphorylation signal in the presence of RNA, demonstrating the stimulating effect of RNA on AURKA kinase activity. These results have been integrated into the manuscript as **Figure 8c** and **8d**. In addition, we also observed an increased autophosphorylation of AURKA in the presence of RNA (**Figure 8e**). Altogether, we conclude that RNA stimulates the kinase activity of AURKA.

A possible mechanism could be the stabilization of the 3D conformation of AURKA by RNA (e.g. stabilization of the disordered region localized at the N-terminal part aa 1-122 of the protein). To test this hypothesis, we have performed a nanoDSF analysis of AURKA in absence and presence of total

RNA. This analysis showed an increase of the melting temperature of AURKA upon addition of total RNA, which demonstrated a stabilizing effect of the RNA molecules on the conformation of AURKA. These results have been added to the manuscript as **Figure 8f**.

Thus, these new findings (lines 401-431) give both functional and mechanistic insights into how RNA can play an important role in riboregulating protein-protein interactions and protein functions.

7. Throughout the manuscript the authors refer to proteins as being ‘RNA-dependent’, but this term is not well defined. I assume that the authors mean that the migration of the protein is RNA dependent. However, this term is used loosely and in several places the authors imply that the function of a protein is RNA-dependent but no data is provided to support this speculation.

We apologize for the confusion. The term “RNA dependent” originates from the first R-DeeP screen study (Caudron-Herger et al. Mol Cell 2019, <https://doi.org/10.1016/j.molcel.2019.04.018>). It is defined in the introduction (67-69) and in the first section of the results (lines 130-133). For more clarity, we have edited this paragraph of the results, with the aim to clarify what RNA dependent proteins are: RNA-dependent proteins are proteins, whose interactome depends on RNA - *i.e.* which are shifting in the R-DeeP gradient, either toward lower sucrose gradient fractions (left shift, loss of interaction partners) or toward higher sucrose gradient fractions (right shift, gain of interaction partners), see lines 136-156.

With regards to the RNA-dependent function of the proteins, we have carefully re-evaluated the manuscript and reformulated the statement where needed and according to our new results (see lines 33, 46, 115-116, 118, 401-431, 538-545, 557-558 and 575-577).

8. There are several inconsistencies in this work that need to be addressed. First, In Figure 2 the authors report that INCENP migration does not change upon RNase treatment. However, in Figure 4B INCENP is reported as a RNase-sensitive interactor of Aurora-A.

INCENP was listed in Figure 2 under the proteins associated with the condensed chromosomes. It was not labeled in “bold” because the analysis of the mitotic R-DeeP screen did not detect a significant shift of the protein. Since the INCENP protein amount is in the 38% weakest protein amount in mitosis and 34% weakest protein amount in interphase, these lower amounts of protein could be challenging for the R-DeeP screening method. As can be seen from the R-DeeP graph of INCENP in mitosis (R-DeeP3.dkfz.de, **Figure R2**), the control and RNase profile for INCENP were different, but the differences were not significant according to our analysis. It is also noted that the protein seemed to partially precipitate in the mitosis sample, as it depicts a peak at fraction 25, both in the control and RNase-treated gradient (**Figure R2**).

For some proteins, rescaling the gradient concentrations could be helpful to obtain a significant shift upon RNase treatment. This was for example very helpful for the ENO1 proteins (Huppertz et al. Mol Cell 2022, <https://doi.org/10.1016/j.molcel.2022.05.019>), where a gradient from 5% to 25% instead of 5% to 50% was used, allowing to “zoom” into the shift and make it clear. Such an approach could probably be helpful for INCENP. This is now discussed in lines 448-461 of the discussion addressing the limitations of this study.

Figure R2 shows the profile of INCEP (INCE_HUMAN) in the interphase R-DeeP screen (significant shift) as compared to the mitotic R-DeeP screen (not significant shift).

Second, INCEP is a stoichiometric cofactor with Aurora-B and is not known to bind to Aurora-A. This needs to be discussed.

According to our results, INCEP was in the list of the RNase-sensitive interactor candidates of AURKA in mitotic lysate (**Supplementary Table 3**). Although we agree that this result was somewhat surprising, in agreement with the above discussion (point 5c), it is essential to have orthogonal validation of the results. As suggested by Reviewer #2, the data from our mitotic R-DeeP gradient provided an excellent orthogonal strategy to analyze protein-protein complexes, as protein interacting within the same complexes were expected to co-migrate into the same control fraction (see our original publication in which we integrated the R-DeeP dataset with the dataset from the CORUM database, Caudron-Herger et al. Mol Cell 2019, <https://doi.org/10.1016/j.molcel.2019.04.018> and CORUM database: <https://mips.helmholtz-muenchen.de/corum/>; CORUM publication: <https://doi.org/10.1093/nar/gkac1015>). As seen from the intersection of the IP and R-DeeP data, INCEP was not found in the group of proteins co-migrating with AURKA around the mitotic control fraction 21, and thus, not validated as AURKA interaction partner through these results. This analysis is detailed in **Supplementary Table 3** and in the results lines 284-292.

Third, Aurora-A and TPX2 form a tight, stoichiometric complex that has been crystalized (14580337, 12699628). RNA is not part of the mechanism reported for TPX2 binding or activation of Aurora-A. The authors should discuss their model for how RNA is involved in this complex formation and why their model differs from the published literature.

We thank the reviewer for raising this point to further integrate our data into the available literature. First, for the publication of the crystal structure (PMID 14580337, [https://doi.org/10.1016/s1097-2765\(03\)00392-7](https://doi.org/10.1016/s1097-2765(03)00392-7)), it appears that truncated forms of both the proteins have been used (AURKA residues 122-403 and TPX2 residues 1-43; PDB code 1OL5, <https://doi.org/10.2210/pdb1OL5/pdb>). In principle, it is not possible to compare the complex formation of the full-length proteins with a truncated protein since either additional interaction surfaces or sterically hindering or suppressive regions may be missing. For example, the truncated form of AURKA (aa 122-403) is missing the N-terminus of AURKA, which is disordered. Disordered regions are regions of proteins that could be important to mediate interactions with RNA molecules (Castello et al. Cell 2012, <https://doi.org/10.1016/j.cell.2012.04.031>). Second, another publication (PMID 12699628, [https://doi.org/10.1016/S0960-9822\(03\)00166-0](https://doi.org/10.1016/S0960-9822(03)00166-0)) identified TPX2 as an activator of AURKA. There is no mention of RNase treatment in this publication, that would exclude the importance of RNA for AURKA/TPX2 interaction.

Our results demonstrate that the interaction between AURKA and TPX2 was modulated by RNA in multiple ways: i) TPX2 was found as one of the seven most promising RNase sensitive AURKA-interacting partner (**Figure 5b**), ii) TPX2 was co-migrating with AURKA around fraction 21 in the mitotic R-DeeP sucrose gradient and depicting an RNA-dependent left shift (loss of interaction partners upon RNase treatment, **Figure 5d**), iii) the co-IP of TPX2 by AURKA was strongly reduced upon RNase treatment (**Figure 5f**), iv) the reverse co-IP of AURKA by TPX2 was strongly reduced upon RNase treatment (**Supplementary Fig. 4d**), v) the PLA signal between TPX2 and AURKA was reduced to background level upon RNase treatment (**Supplementary Fig. 7e-7h**), and vi) the addition of total RNA into the *in vitro* AURKA kinase assay stimulated the phosphorylation of TPX2 (**Figure 8d**). All these findings argue in favor of an RNA-mediated interaction of TPX2 with AURKA.

We hypothesized that RNA could act as a structural stabilizing agent. In particular, RNA could play a role in stabilizing the disordered part of the proteins, residues 1 to 122 for AURKA (which were not included in the crystal structure from publication 14580337 mentioned above) and a large proportion of TPX2 (see MoBiDB and alpha-fold webpages).

For AURKA, we could demonstrate that RNA had indeed a stabilizing effect on AURKA conformation (**Figure 8f**, nanoDSF data showing a shift of the melting temperature of AURKA toward higher temperature upon addition of RNA molecules). For TPX2, the nanoDSF data were inconclusive as there were no changes in the 350 nm/330 nm ratio detected during unfolding of TPX2 upon exposure to the linear temperature gradient ranging from 20°C to 90°C. Technically, it was not possible to analyze the impact of total RNA addition on TPX2 conformational stability.

We have added a short discussion in lines 494-503 of the manuscript.

Fourth, the CPC has been reported to be an RNA-binding complex in several different organisms but is not identified in the R-DeeP screen. The authors should discuss why their results differ from the published literature.

The chromosome passenger complex (CPC) is composed of the AURKB, INCENP, borealin and Survivin/BIRC5 subunits. According to the RBP2GO database, those proteins were generally not well detected in proteome-wide studies: borealin and Survivin had not been detected at all, AURKB was detected only once in a computational prediction but never experimentally (Brannan et al.). INCENP had been detected only once in the Queiroz et al. study (PMID 31123356) out of 43 human datasets. Thus, all of these proteins were very rarely detected in proteome-wide studies and hence it is reasonable that these were not detected here. Nonetheless, we detected AURKB and INCENP in our R-DeeP interphase screen.

Possibly, the CPC complex is precipitated with the chromosomes during cell lysate preparation and initial cleaning centrifugation, complicating its detection when the protein complexity (whole cell extract) is high. Altogether, R-DeeP performs equally to one and superior to all other proteome-wide studies with regard to identifying the RNA dependence of members of this complex. We have discussed this point in lines 448-461 of the manuscript, addressing the limitations of this study.

Minor Points

1. In Figure 1E it is distracting and redundant to have each gene name followed by ‘_HUMAN’

We have used in Figure 1E (now **Supplementary Fig. 2c**) the Entry name (UniProt database, <https://www.uniprot.org>) of the proteins and not the gene name to unambiguously define the protein. The Entry name is the ID used to best search the R-DeeP3 database, the reason why we would suggest

to keep using it in this figure, directly related to the R-DeeP gradients. The same applies to **Supplementary Fig. 3b** (screen shot from the database).

2. On line 61 the authors identify organic phase separation as a method to identify RNA binding proteins. This line should include the following references (30607034, 30824702)

We have added to corresponding references to the two studies (line 63).

Reviewer #2 (Remarks to the Author)

The manuscript by Rajagopal et al used the R-DEEP method to investigate RNA-dependent proteins in mitosis, and further characterize the RNA-dependent interaction of three proteins, AURKA, TPX2 and KIFC1.

Overall, the manuscript is well presented and scientifically sound, although some sections could be extended to describe the results in more details.

Comments:

1: how did the authors characterize the effectiveness of the synchronization procedure to produce mitotic and interphase samples? Was this checked and confirmed (for example by FACS)?

The effectiveness of the synchronization procedure was validated by western blot analysis of the H3pS10 which is expected to be highly increased in mitotic cells. We have added this validation as **Supplementary Fig. 1b and 1c** (biological triplicates). Although the synchronization procedure is very reproducible, before harvesting the cells, the plates were systematically assessed at the microscope for a high density of rounded cells (qualitative control), which is characteristic of mitotic cells.

2: In the section “An atlas of RNA-dependent proteins in cell division highlights well-known mitotic factors as RNA dependent proteins”, it is not clear how the 826 RNA-dependant mitotic proteins were identified. Was it only by comparing the mitotic R-DeeP results and the RBP2GO resource, or was it by comparing the mitotic vs interphase R-DeeP datasets? In other words, how many of these 826 proteins are found in the 1093 proteins present only in the mitotic R-DeeP dataset, compared to the interphase dataset?

We have rephrased this part of the manuscript (lines 238-242) to clarify that the 826 RNA-dependent mitotic proteins comprised 800 proteins listed as RNA-binding protein candidates listed in the RBP2GO database (overlapping with 153 proteins from the mitotic R-DeeP screen) and additional 26 proteins, that were identified only in the present R-DeeP mitotic screen as RNA dependent proteins. These proteins are listed in **Supplementary Table 2** (spreadsheet “Mitotic proteins”).

Overall, the manuscript will benefit from a better comparison of the mitotic and interphase datasets. Are there proteins that are only identified in the mitotic or the interphase samples? Are there proteins that are RNA independant in one dataset but RNA dependant in the other dataset?

As shown in the **Figure 1b**, there were 6059 proteins commonly detected in both screens. Additionally, 1010 proteins were found only in interphase and 1093 proteins only in mitosis. We have now created a dedicated figure to the comparative analysis of the mitotic and interphasic R-DeeP screens (see new **Figure 2**). Figure 2 focuses on the differential shifting behavior of the proteins in the screens. For example, there were 586 proteins which shifted in mitosis but not in interphase and there were 726 proteins which shifted in interphase but not in mitosis (**Figure 2b**). A GO enrichment analysis indicated that these two groups shared a number of common GO terms, which were often linked to mitochondrial and endoplasmic reticulum components, which can be linked to nocodazole treatment in the synchronization procedure. The two groups also shared the GO term “cytoskeleton”, with the distinction though, that mitotic proteins were more related to microtubule functions and interphasic proteins to actin functions (**Supplementary Table 1**), indicating a possible regulatory role of RNA in interphase via actin and in mitosis via microtubules (lines 182-185).

Interestingly, there were also 136 proteins shifting in opposite direction (**Figure 2b**), which contained 46 proteins with catalytic functions (**Supplementary Table 1**). The text has been updated accordingly (lines 185-186).

Additionally, it would be interesting to compare the mitotic-specific as well as the interphase-specific R-Deep dataset with previous R-DeeP datasets done (by the same authors) on unsynchronised cells. I would expect a strong overlap between the interphase and the unsynchronised datasets, but a difference between the mitotic and unsynchronised datasets. How many of the 826 mitotic specific proteins could have been identified using unsynchronised cells (for example, could AURKA, TPX2 and KIFC1 be identified as RBPs in the unsynchronised datasets?). This comparison would surely provide a better justification of the rationale to perform R-DeeP specifically on mitotic arrested cells.

According to this reviewer's suggestion, we have added the analysis of the intersections between the unsynchronized R-DeeP data set in HeLa cells with the two cell cycle specific datasets (**Figure 2d**). Interestingly, the analysis of the intersection between the R-DeeP mitotic dataset and the R-DeeP unsynchronized dataset revealed that on the one hand, 372 proteins were newly detected as left shifts in the mitotic dataset (**Figure 2e**). In addition, 338 common proteins were not significantly shifting in the unsynchronized dataset and were now detected with a left shift in the mitotic dataset. A GO analysis of those two groups revealed that the 372 newly detected and shifting proteins were enriched in GO terms related to mitosis (see **Supplementary Table 1**) of which, 111 had a high protein expression level in mitosis as compared to interphase. For the 338 common but now detected as shifting proteins, there is an interesting enrichment for GO terms associated to ER- and mitochondrial-related membrane, maybe due to nocodazole treatment (see Reviewer #1, comment 1 above). Regarding the number of detected proteins, there was a strong overlap with the former dataset (3987 proteins out of 4765 detected in the previous unsynchronized screen = 84%) as seen from the Upset plot. However, the two new datasets had a relevantly better coverage of the proteome and added additional information for 2072 proteins detected in both new datasets plus 1870 proteins newly detected in only one of the new datasets (**Figure 2d**).

The proteins AURKA, KIFC1 and TPX2 had been detected in the unsynchronized R-DeeP dataset. For AURKA, there was a significant shift detected in the previous dataset, but this was based on very limited signals in the mass spectrometry, so that the new screen has strongly improved this dataset. For TPX2, a shift was detected in the unsynchronized dataset, but was showing a precipitation of the protein, which has now been uncovered using the mitotic screen to be a typical left shift indicating RNA dependence. For KIFC1, there was no shift detected in the previous dataset, while it is highly evident now in the mitosis dataset. In summary, the mitosis screen has not only largely broadened the available information, but also strengthened the data basis for the RNA dependence of these three proteins.

The superiority of the new dataset is also documented when analyzing the protein amounts compared between the mitotic and the interphasic samples (**Supplementary Table 1**). The expression of 1859 proteins was significantly increased in mitosis and this subset included 160 left-shifting, i.e. RNA-dependent proteins. Notably, KIFC1 was one of the proteins with the strongest mitosis/interphase ratio (**Supplementary Table 1**, Mean_Protein_Amount_Fold_Change_FDR), which underlines the improvement by analyzing the protein in the mitotic screen.

We have updated the manuscript accordingly. See lines 192-201.

3: In the section "AURKA has both RNase-sensitive and RNase-insensitive interactions with other proteins", the choice of studying further KIFC1 should be better justified. Why choosing this protein specifically out of the 1080 identified? The sentence "We also identified KIFC1 as one of the most

frequently detected RNase-sensitive candidates, whose interaction with AURKA had not been characterized before” is not clear. What do the authors mean by most frequently detected?

We have rephrased this confusing part of the text to better clarify the rationale for further studying the interaction between AURKA and KIFC1 (lines 284-292). First, KIFC1 appeared as mitotic an RNA-dependent protein (**Supplementary Table 1**). Second, KIFC1 was an RNase-sensitive interaction partner of AURKA (**Supplementary Table 3**). Third, KIFC1 shifted in the mitotic R-DeeP screen from the same fraction as AURKA (around fraction 21, **Figure 5b**, see point 6 below). Fourth, KIFC1 was listed as protein associated to mitotic-related GO terms (**Supplementary Table 2**). Fifth, KIFC1 was one of the proteins showing the highest increased fold-change in protein expression level in mitosis as compared to interphase (**Supplementary Table 1**). Finally, KIFC1 interaction with AURKA (and TPX2) had not been characterized before.

4: The description of the PLA assay (lanes 265-268) is oversimplified. The author should either not describe it and refer the original article, or describe it better. The sentence “represented by signal in the form of dots resulting from a ligation followed by an amplification reaction” is too simplistic.

As suggested by the reviewer, we have removed the oversimplified explanation from the text and refer to the original article (line 316).

5: In the section “AURKA phosphorylates KIFC1 at S349 and T359”, did the authors observe any cell cycle or morphological defects in cells transfected with the S349A/T359A mutant, in comparison to the WT protein? If these phosphorylation events are biologically important, these mutations should have consequences on cell division, migration, ...

We totally agree that this is an important question. Therefore, we have tried to analyze the functional relevance of KIFC1 phosphorylation in KIFC1 KO HeLa and A549 cell lines (which were in preparation at the time of the original submission of our manuscript), as we used HeLa and A549 in this study. Unfortunately, a phenotype due to KIFC1 knockout was difficult to characterize. This finding is in concordance with the available literature, according to which KIFC1/HSET is a normally non-essential kinesin, except for cells presenting centrosomal amplification (a high percentage of cells with more than two centrosomes) (Kwon et al. *Genes & Development* 2008, <https://doi.org/10.1101/gad.1700908> and Kim et al. *Cell Structure and Function* 2013, <https://doi.org/10.1247/csf.12014>). In such cells, KIFC1 depletion seems to lead to the formation of an increased occurrence of multipolar spindles, where KIFC1 should help clustering the extra-numerous centrosomes into two poles for successful cell division. HeLa and A549 cells do not appear to present remarkable centrosomal amplification, which can explain the lack of characteristic phenotype in the KIFC1 KO versions of these cell lines.

To further test this, we have cultured MDA-MB-231 cells (authenticated, a cell line known to present centrosomal amplification) and tested the overexpression of KIFC1 WT and KIFC1 double mutant under KD of KIFC1 per siPOOLS (targeting the 3'UTR region to only silence the endogenous KIFC1 gene). While the knockdown of KIFC1 was successful (see **Figure R1** below), we could not detect a significant increase in multipolar spindles in the KIFC1 knockdown sample (only about 5% in both samples, siRNA control as compared to siRNA KIFC1 in MDA-MB-231 cells).

Figure R1 top panel shows the successful depletion of KIFC1 upon siRNA treatment in MDA-MB-231 cells. The bottom panel shows representative images of mitotic spindles. In each sample, about 5% of spindles showed multipolar spindles. About 40 spindles were qualitatively assessed in each sample).

Accordingly, we could not obtain data on the functionality of the KIFC1 double mutant due to the lack of a conclusive phenotype and assay system. While we agree that investigating the functionality of the phosphorylation of KIFC1 is important, it might be only applicable for specific cell types (e.g. cancer cells with amplified centrosomes, possibly meiotic cells in absence of centrosomes). We have added a short discussion on this point in the manuscript (lines 511-520).

However, in our study, we believe that we managed to present a strong general finding with the effect of RNA on AURKA kinase activity and thermostability.

We have used purified proteins (AURKA, a kinase-dead mutant of AURKA as a control) and performed an *in vitro* kinase assay to monitor the phosphorylation of both KIFC1 and TPX2 in absence and presence of RNA molecules (total RNA). All three biological replicates for KIFC1 and TPX2 show an increased phosphorylation signal in the presence of RNA, demonstrating the stimulating effect of RNA on AURKA kinase activity on these substrates. These results have been integrated into the manuscript as **Figure 8c** and **8d**. In addition, we also observe an increased autophosphorylation of AURKA in the presence of RNA (**Figure 8e**). Altogether, we conclude that RNA stimulates the kinase activity of AURKA.

A possible mechanism could be the stabilization of the 3D conformation of AURKA by RNA (e.g. stabilization of the disordered region localized at the N-terminal part aa 1-122 of the protein). To test this hypothesis, we have performed a nanoDSF analysis of AURKA in absence and presence of total RNA. This analysis showed an increase of the melting temperature of AURKA upon addition of total RNA, which demonstrated a stabilizing effect of the RNA molecules on the conformation of AURKA. These results have been added to the manuscript as **Figure 8f**.

Thus, these new findings (lines 401-431) give both functional and mechanistic insights into how RNA can play an important role in riboregulating protein-protein interactions and protein functions.

6: In the discussion section (about the RNA dependant complex between AURKA, TPX2 and KIFC1), would it be possible to match the R-DEEP mitotic RNA-sensitive proteins, especially those eluting at fraction 21, and the AURK4 RNA-dependant interactors. This might give a hint of other proteins that could be in the same complex, and could provide avenues for further work on the characterization of this complex.

Following the suggestion of the reviewer, we have performed the proposed analysis, which resulted in a group of 43 proteins which were co-sedimenting with AURKA around control fraction 21, shifting in the mitotic R-DeeP screen and RNase-sensitive in the AURKA IP (**Supplementary Table 3**). In addition, we labeled the proteins listed as mitotic-related proteins according to our analysis of the proteins linked to mitotic GO terms (**Supplementary Table 2**). This identified KIFC1, NOL6, CLASP1, CLASP2, MEAF6, RHAMM and TPX2 as strong candidates. We have added this finding as a supplementary spreadsheet in **Supplementary Table 3** and discussed this in the text lines 285-292.

Minor comments:

Lane 324: “were rather localized in the lower 20% binding sites (Figure 5D)”. This is shown in Figure 5F

Thank you for noticing this mistake in the reference to the figure. We have now assigned the description to the new **Figure 7g** (line 372).

Reviewer #3 (Remarks to the Author):

The manuscript investigates the role of RNA-binding proteins (RBPs) and RNA-dependent protein interactions during cell cycle, with a particular focus on mitosis. The authors utilized the R-Deep strategy, a proteomic approach they had previously developed, to identify RNA-dependent proteins in HeLa cells synchronized at mitosis and interphase. Their analysis yielded a comprehensive list of RNA-dependent proteins. Interestingly, several key mitotic proteins, including AURKA, KIFC1, and TPX2, exhibited RNA-dependent interactions despite lacking canonical RNA-binding domains. Immunoprecipitation, western blotting and UV-crosslinking assays showed that AURKA, KIFC1, and TPX2 associate with RNA and interact with one another in an RNA-dependent manner. Based on PLA data, these interactions appear to occur during mitotic spindle assembly. Additionally, the authors performed iCLIP-seq, revealing that KIFC1 interacts with a diverse set of RNAs, including rRNAs and mRNAs, without apparent sequence specificity. Mutagenesis and *in vitro* kinase assays further demonstrated that AURKA phosphorylates KIFC1 at residues S349 and T359.

The findings suggest that these proteins may function as unconventional RBPs and that RNA could play a role in mediating protein-protein interactions, particularly during spindle assembly. The R-Deep3 online database is as a valuable and accessible resource for the research community.

While this study is comprehensive and intriguing, a significant limitation is the lack of data demonstrating the functional significance of RNA-mediated interactions.

Major points:

1. I recommend that the authors explore the role of RNA in AURKA activity by performing *in vitro* kinase assays with RNA. Does RNA enhance the kinase activity of AURKA on KIFC1? Even if the result might be negative and RNA may act through other mechanisms, this relatively straightforward experiment could provide some insights into the mechanism.

We thank the reviewer for this suggestion. We have used purified proteins (AURKA, a kinase-dead mutant of AURKA as a control) and performed an *in vitro* kinase assay to monitor the phosphorylation of both KIFC1 and TPX2 in absence and presence of RNA molecules (total RNA). All three biological replicates for KIFC1 and TPX2 show an increased phosphorylation signal in the presence of RNA, demonstrating the stimulating effect of RNA on AURKA kinase activity on these substrates. These results have been integrated into the manuscript as **Figure 8c** and **8d**. In addition, we also observe an increased autophosphorylation of AURKA in the presence of RNA (**Figure 8e**). Altogether, we conclude that RNA stimulates the kinase activity of AURKA.

A possible mechanism could be the stabilization of the 3D conformation of AURKA by RNA (e.g. stabilization of the disordered region localized at the N-terminal part aa 1-122 of the protein). To test this hypothesis, we have performed a nanoDSF analysis of AURKA in absence and presence of total RNA. This analysis showed an increase of the melting temperature of AURKA upon addition of total RNA, which demonstrated a stabilizing effect of the RNA molecules on the conformation of AURKA. These results have been added to the manuscript as **Figure 8f**.

Thus, these new findings (lines 401-431) give both functional and mechanistic insights into how RNA can play an important role in riboregulating protein-protein interactions and protein functions.

2. In Figure 5A, the evidence for direct RNA binding and the specificity of the antibody should be reinforced. Could the authors knock down AURKA and perform the same experiment to rule out the possibility of antibody cross-reactivity? Testing synchronized cells in interphase could also serve as a specificity control.

To answer this questions, we have performed two experiments: first, we have performed a CLIP of AURKA using an EGFP-fused construct of AURKA and added an EGFP vector as well as a non-crosslinked sample as controls (**Supplementary Fig. 10a**, also depicted below) and second, we have performed a CLIP of AURKA after siRNA-mediated knockdown of AURKA (**Supplementary Fig. 10b**).

Supplementary Fig. 10a shows the results of the CLIP using EGFP-AURKA, with increased signal upon EGFP-AURKA pulldown as well as a shift towards higher molecular weight upon RNase dilution (top panel). The bottom panel shows the similar pulldown efficiency in the four samples.

As can be seen in the **Supplementary Fig. 10a**, there was a strong signal specific for EGFP-AURKA, that was shifting to higher molecular weight upon RNase dilution. This was not visible in the EGFP control (lane 1, top panel) and also absent in the non-crosslinked control (lane 4, top panel). Accordingly, this indicated the specific binding of AURKA to RNA. This experiment has been performed in three biological replicates, which depicted reproducible results, available as Source Data files.

We have in addition performed a CLIP of AURKA after siRNA-mediated knockdown of AURKA (siRNA control versus siRNA against AURKA). While the siRNA treatment diminished the AURKA expression in the input lysate below the detection limit of the western blot (**Supplementary Fig. 10b**, also depicted below, lane 4), the residual amount of the protein was sufficient to be immunoprecipitated and give a well detectable signal in the IP (**Supplementary Fig. 10b**, lane 6). We have focused the CLIP to the samples with 1:50 dilution of the RNase, which showed the shift of the signal towards higher molecular weights. As seen in **Supplementary Fig. 10b** below, the smear towards the higher molecular weights was reduced upon AURKA siRNA. Together, this further hints to the specificity of the signal for AURKA interacting with RNA molecules.

This experiment was performed in three biological replicates, which resulted in reproducible results, available as Source Data files.

Supplementary Fig. 10b

Left panel: top, autoradiography indicating the direct binding of AURKA to RNA by iCLIP in a siRNA control sample as compared to an AURKA siRNA sample, where the signal is reduced (one representative image out of N=3 biological replicates is shown). Bottom, results of the corresponding IP showing the protein amounts in the IP. It is noted that even if AURKA appears to be strongly reduced in the input sample after AURKA siRNA treatment as compared to the control siRNA (as seen in c), there is still a residual AURKA signal in the IP after siRNA-mediated knockdown of AURKA (lane 2). One representative image out of N=3 biological replicates is shown.

Right panel: control IP of AURKA showing the amount of protein after siRNA-control and siRNA-mediated knockdown of AURKA. The signal for AURKA in the siRNA lysate (lane 2) is only visible upon long exposure time, resulting in a highly saturated signal in the siRNA AURKA lysate (lane 1). One representative image out of N=3 biological replicates is shown.

Together, these experiments offer three independent indications of the specificity of the AURKA CLIP signal: the absence of signal in the no crosslink control samples (see also **Figure 7a**), the absence of a signal in the EGFP pulldown control (**Supplementary Fig. 10a**) and the decrease of the signal upon AURKA knockdown (**Supplementary Fig. 10b**), so that we conclude that AURKA does directly bind to RNA. This is also strengthened by the experiments showing an increased *in vitro* kinase activity and a conformational stabilization of AURKA by RNA.

Minor points:

1. In Figure 4C, there appear to be two RNA-dependent peaks at fractions 17-18 and fractions 21-22. This should be mentioned in the text.

The text has been updated accordingly (see lines 299-300).

2. rRNAs account for approximately 45% of the iCLIP reads, but it is unclear whether rRNA reads are enriched in the library. Clarifying whether these data were obtained after removing unmappable reads and indicating the enrichment fold relative to input would improve the clarity of the figures.

We have now clarified that the proportion of rRNA reads is the proportion of rRNA reads in the libraries, *i.e.* before mapping to the genome and removal of unmappable reads (see lines 262-263 and **legend of figure 7c**). The procedure used for iCLIP sequencing does not require the sequencing of an input sample. Accordingly, it is not possible to indicate a relative fold enrichment of the rRNA reads. Also, the new RNA-seq data on total RNA were performed on RNA samples that were ribo-depleted, which would also not allow a proper comparison of the rRNA amount.

1) We have answered the minor points of the reviewers as followed:

Reviewer #1 (Remarks to the Author):

Summary

This is a revision of a previous manuscript that utilized the R-DeeP fractionation and mass spectrometry methodology to identify a catalog of RNA binding proteins during interphase and mitosis. The authors have performed a tremendous amount of work in this revision and addressed all the concerns that I raised. The new data included in Figure 2, S15c-d, and 8d-f are particularly valuable. I am now convinced that this is important contribution to the literature and is suitable for publication. I have two minor points that should be addressed prior to publication.

The authors thank Reviewer #1 for the positive evaluation of our manuscript.

Minor points

1. Line 309-311. Co-IP assay does not demonstrate a direct interaction between TPX2 and KIF1C. This interaction could be bridged by other proteins or RNA.

We have removed the word "direct" from the mentioned sentence. See line 311 of the manuscript.

2. Line 404-405. "in vitro kinase assay using purified AURKA in the absence or presence of total RNA for both, KIFC1 and TPX2." It is difficult to understand what was meant by this sentence.

The sentence was reformulated into "we performed an in vitro kinase assay to analyze the phosphorylation of KIFC1 and TPX2 by AURKA, using purified proteins, both in the absence or presence of total RNA. See lines 405-407 of the manuscript.

Reviewer #2 (Remarks to the Author):

Following the reviewers comments, the manuscript has been significantly improved and I have no further concerns. It is a nice study.

The authors thank Reviewer #2 for the positive evaluation of our manuscript.

Reviewer #3 (Remarks to the Author):

The authors have provided additional experimental data, which fully addressed my earlier concerns. I recommend the publication of this manuscript.

The authors thank Reviewer #3 for the positive evaluation of our manuscript.